

# Assessment of hydrothermal alteration on micro- and nanostructures of biocarbonates: quantitative statistical grain-area analysis of diagenetic overprint

Laura A. Casella[1*], Sixin He[1], Erika Griesshaber[1], Lourdes Fernández-Díaz[2], Elizabeth M. Harper[3],
Daniel J. Jackson[4], Andreas Ziegler[5], Vasileios Mavromatis[6,7], Martin Dietzel[7], Anton Eisenhauer[8],
Uwe Brand[9], and Wolfgang W. Schmahl[1]

[1]Department of Earth and Environmental Sciences and GeoBioCenter, Ludwig-Maximilians-Universität München, Munich, 80333, Germany
[2]Instituto de Geociencias, Universidad Complutense Madrid (UCM, CSIC), Madrid, 28040, Spain
[3]Department of Earth Sciences, University of Cambridge, Cambridge, CB2 3EQ, U. K.
[4]Department of Geobiology, Georg-August University of Göttingen, Göttingen, 37077, Germany
[5]Central Facility for Electron Microscopy, University of Ulm, Ulm, 89081, Germany
[6]Géosciences Environnement Toulouse (GET), CNRS, UMR 5563, Observatoire Midi-Pyrénées, 14 Av. E. Belin, 31400 Toulouse, France
[7]Institute of Applied Geosciences, Graz University of Technology, Rechbauerstr. 12, 8010 Graz, Austria
[8]GEOMAR-Helmholtz Centre for Ocean Research, Marine Biogeochemistry/Marine Geosystems, Kiel, Germany
[9]Department of Earth Sciences, Brock University, 1812 Sir Isaac Brock Way, St. Catharines, Ontario, L2S 3A1, Canada

*Correspondence to*: Laura A. Casella (Laura.Casella@lrz.uni-muenchen.de)

**Abstract.** The assessment of diagenetic overprint on microstructural and geochemical data gained from fossil archives is of fundamental importance for understanding palaeoenvironments. A correct reconstruction of past environmental dynamics is only possible when pristine skeletons are unequivocally distinguished from altered skeletal elements. Our previous studies (Casella et al., 2017) have shown that replacement of biogenic carbonate by inorganic calcite occurs via an interface coupled dissolution–reprecipitation mechanism. Furthermore, for a comprehensive assessment of alteration, structural changes have to be assessed on the nanoscale as well, which documents the replacement of pristine nanoparticulate calcite by diagenetic nanorhombohedral calcite (Casella et al., 2018a, b).

In the present contribution we investigated six different modern biogenic carbonate microstructures for their behaviour under hydrothermal alteration in order to assess their potential to withstand diagenetic overprint and to test the integrity of their preservation in the fossil record. For each microstructure (a) the evolution of biogenic aragonite and calcite replacement by inorganic calcite was examined, (b) distinct carbonate mineral formation steps on the micrometre scale were highlighted, (c) microstructural changes at different stages of alteration were explored, and (d) statistical analysis of differences in basic mineral unit dimensions in pristine and altered skeletons was performed. The latter analysis enables an unequivocal determination of the degree of diagenetic overprint and discloses information especially about low degrees of hydrothermal alteration.



# 1 Introduction

Biomineralised hard parts composed of calcium carbonate form the basis of studies of past climate dynamics and environmental change. However, the greatest challenge that all biological archives face lies in their capacity to retain original signatures, as alteration of them starts immediately upon death of the organism. Biopolymers decay, and inorganic minerals precipitate within as well as at the outer surfaces of the hard tissue (e.g., Patterson and Walter, 1994; Ku et al., 1999; Brand et al., 2004; Zazzo et al., 2004).

Despite ongoing and extensive research, carbonate diagenesis remains only partly understood. Many studies addressing the evolution of parameters which influence diagenetic alteration are discussed in only a qualitative manner (Brand and Veizer, 1980, 1981; Swart, 2015). In particular, deciphering the sequence of those processes with many steps of alteration and unknown intermediate stages poses one of the major problems in understanding carbonate diagenesis (Immenhauser et al., 2015a; Swart, 2015; Ullmann and Korte, 2015). Our previous studies on the shell of the modern bivalve *Arctica islandica* have shown that simulated diagenetic alteration discloses microstructural and geochemical features that are comparable to those found in fossils (Casella et al., 2017; Ritter et al., 2017). However, both studies covered only the hard tissue of one taxon. For a more comprehensive understanding of microstructural and chemical controls during diagenesis, hard tissues of other archives have to be thoroughly examined. Accordingly, we extended our studies onto hard tissues of other modern marine carbonate biomineralisers such as the bivalves, *A. islandica*, and *Mytilus edulis*, the coral *Porites sp.*, and the gastropod *Haliotis ovina*. With these organisms we cover both major calcium carbonate phases and, further to that present in the shell of *A. islandica*, five additional microstructures. When selecting model organisms for this study, care was taken to investigate those for which fossil counterparts are used for palaeoclimate and palaeoenvironmental reconstructions.

The bivalve *A. islandica* has been studied extensively in several scientific articles and fields (e.g., Ridgway and Richardson, 2011; Strahl et al., 2011; Wanamaker et al., 2011; Karney et al., 2012; Krause-Nehring et al., 2012; Ridgway et al., 2012; Butler et al., 2013; Schöne, 2013). The first occurrence of *A. islandica* in the Mediterranean Sea has a historical importance and was used until 2010 to mark the former Pliocene–Pleistocene boundary (e.g., Crippa and Raineri, 2015; Crippa et al., 2016). As long-lived organisms, stony corals attract great interest for the reconstruction of palaeoclimates derived from skeletal oxygen isotopic compositions and major element abundances, as these geochemical signals vary in response to changes in seawater temperature (e.g., Meibom et al., 2007). It is assumed that $\delta^{234}U$ in sea water has remained constant in the past, thus, the comparison between present-day and decay-corrected $\delta^{234}U$ in sea water and in coral skeletons is a major tool for the detection of diagenetically altered corals. $\delta^{234}U$ values of the latter are higher relative to present day sea water (Hamelin et al., 1991; Stirling et al., 1995; Delanghe et al., 2002), while pristine corals exhibit a $^{234}U/^{238}U$ activity ratio similar to modern sea water (Henderson et al., 1993; Blanchon et al., 2009). Shells of *M. edulis* and *H. ovina* represent new archives for studies of palaeo- and present environmental change. The work of Hahn et al. (2012, 2014) has shown that environmental reconstruction can be derived from microstructural information, as well as stable isotope and major element



data. The shells of *M. edulis* and *H. ovina* consist of two layers with distinct microstructures. In *H. ovina* the two layers are composed of aragonite, whereas the shell of *M.s edulis* consists of an outer calcite and inner aragonite layer.

To reliably identify low to moderate degrees of diagenetic overprint, we investigated the behaviour of biocarbonate skeletal microstructure during hydrothermal alteration. Laboratory-based hydrothermal alteration experiments were conducted for time spans between 1 and 35 days, at an alteration temperature of 175 °C and in the presence of Mg-rich fluid. Skeletons of two modern bivalves (*A. islandica* and *M. edulis*), one modern stony coral (*Porites sp.*) and one modern gastropod (*H. ovina)* were carried out*. With this selection of hard tissue we are able to investigate the influence, during alteration, of variations in mineral surface area, control by primary (inherent) and secondary (induced) porosity, the effect of biopolymer fabric and pattern of distribution within the skeleton, and the role of the size, form, and mode of organisation of the basic mineral unit.

We discuss differences between calcite and aragonite in biogenic to inorganic carbonate phase replacement kinetics, and illustrate differences in structure and porosity between original and product phases. Overprint highly affects basic mineral unit sizes in the alteration product, and we evaluate this characteristic for pristine and altered skeletons using statistics. Based on statistical grain area analysis, a new and reliable tool for the detection of diagenetic overprint in biological carbonate hard tissue is presented, and this tool is able to characterise low degrees of diagenetic alteration.

## 2 Materials and methods

### 2.1 Test materials

Shells of the modern bivalve *Arctica islandica* were collected from Loch Etive, Scotland (U.K). The shells are 8-10 cm in size and represent adult specimens. Pristine specimens of the scleractinian coral *Porites sp.* were collected at Moorea, French Polynesia (Rashid et al., 2014). Live specimens of the gastropod *Haliotis ovina* were collected from the reef flat of Heron Island, Queensland, Australia. All shell pieces used in this study were taken from the shell of one adult specimen with dimensions of approx. 8 x 6.5 cm. Shells of the modern common blue mussel, *Mytilus edulis*, were collected from 5-7 m depth in the subtidal of Menai Strait Wales, U.K.. Shell sizes varied from 5 to 6 cm and represent adult animals.

### 2.2 Methods applied

#### 2.2.1 Selective etching of organic matrix

In order to image the organic matrix in modern (reference) and hydrothermally altered shell samples, as well as the mineral reference (inorganic aragonite), shells or mineral pieces were mounted on 3 mm thick cylindrical aluminium rods using super glue. The samples were first cut using a Leica Ultracut ultramicrotome with glass knives to obtain plane surfaces. The cut pieces were then polished with a DiATOME diamond knife by stepwise removal of material in a series of 20 sections with successively decreasing thicknesses (90 nm, 70 nm, 40 nm, 20 nm, 10 nm and 5 nm, each step was repeated 15 times)





as reported in Fabritius et al. (2005). The polished samples were etched for 180 seconds using 0.1 M HEPES at pH = 6.5 containing 2.5 % glutaraldehyde as a fixation solution. The etching procedure was followed by dehydration in 100 % isopropyl three times for 10 minutes each, before specimens were critical point-dried. The dried samples were rotary coated with 3 nm platinum and imaged using a Hitachi S5200 Field-Emission Scanning Electron Microscope (FE-SEM) at 4 kV.

## 2.2.2 Microstructure and texture

For FE-SEM and Electron Backscatter Diffraction (EBSD) analyses 5 x 5 mm thick pieces were cut out of the shells and embedded in epoxy resin. The surface of the embedded samples was subjected to several sequential mechanical grinding and polishing steps down to a grain size of 1 μm. The final step was etch-polishing with colloidal alumina (particle size ~ 0.06 μm) in a vibratory polisher. Samples were coated with 4-6 nm of carbon for EBSD analysis, and with 15 nm for SEM visualisation. EBSD measurements were carried out on a Hitachi SU5000 field emission SEM, equipped with an Oxford EBSD detector. The SEM was operated at 20 kV and measurements were indexed with CHANNEL 5 HKL software (Schmidt and Olesen, 1989; Randle and Engler, 2000). Information obtained from EBSD measurements is presented as band contrast images, and colour-coded crystal orientation maps with corresponding pole figures.

The EBSD band contrast represents the signal strength of the EBSD-Kikuchi diffraction pattern and is displayed as a grey-scale component of EBSD scanning maps. The strength of the EBSD signal is high when a crystal is detected (bright), whereas it is weak or absent when a polymer such as organic matter is scanned (dark/black).

Co-orientation statistics are derived from pole figures obtained by EBSD scans and is given by the MUD (multiple of uniform (random) distribution) value. The MUD value measures crystal co-orientation (texture sharpness) in the scanned area, where a high MUD value indicates high crystal co-orientation and a low MUD value reflects a low to random co-orientation.

## 2.2.3 Grain area evaluation

Individual grains can be identified and various parameters measured with EBSD, such as grain area dimensions. A grain is defined as a region completely surrounded by boundaries across by which the misorientation angle relative to the neighbouring grains is larger than a critical value; the critical misorientation value. Griesshaber et al. (2013) determined empirically that a critical misorientation value of 2 ° best suits the microstructure of modern carbonate biological hard tissue. By using this value, individual basic mineral units (e.g., fibres, tablets, prisms, columns), also called as grains can be addressed and evaluated. For the relative frequency to grain area statistics, we use the critical misorientation value of 2 °, grain clusters with a class width of 0.2 μm, and corrected values for absolute distribution function/probability density ($F_x(x)$) to relative values.



### 2.2.4 Alteration experiments

Laboratory-based hydrothermal alteration experiments mimicked burial diagenetic conditions. In all experiments pieces of shells or skeletons up to 2 cm x 1 cm of modern *A. islandica*, modern *M. edulis*, modern *Porites sp.*, and modern *H. ovina* were placed inside a polytetrafluoroethylene (PTFE) vessel together with 10 mL of simulated burial fluid (100 mM NaCl + 10 mM $MgCl_2$ aqueous solution) and sealed with a PTFE lid. Each PTFE vessel was placed in a stainless steel autoclave, sealed and kept in the oven at a temperature of 175 °C for different periods of time ranging between 1 and 35 days. After the selected time period, the autoclave was removed from the oven, cooled down to room temperature and opened. Recovered solid material was dried at room temperature and prepared for XRD, EBSD and EDX measurements.

### 2.2.5 X-ray diffraction analysis

X-ray diffraction analysis of pristine and hydrothermally altered samples was performed with $Cu$-$K\alpha_1$-radiation in reflection geometry on a General Electric Inspection Technologies XRD3003 X-ray diffractometer with an incident-beam Ge111 focussing monochromator and a Meteor position-sensitive detector. The diffractograms underwent Rietveld analysis with the software package FULLPROF (Rodríguez-Caravajal, 2001) using the aragonite structure data of Jarosch and Heger (1986) and calcite structure data of Markgraf and Reeder (1985).

## 3 Results

### 3.1 Microstructural characteristics of *modern* bivalve, gastropod and coral skeletons

FE-SEM images shown in Figs. 1, A1 and A2 highlight characteristic basic mineral units and their assembly within the skeletons of the investigated species: the modern bivalves *Arctica islandica* and *Mytilus edulis*, the modern coral *Porites sp.*, and the modern gastropod *Haliotis ovina*. Skeletons of *A. islandica*, *H.s ovina*, and *Porites sp.* consist entirely of aragonite, whereas *M. edulis* contains both carbonate phases, calcite and aragonite.

The shell of *A. islandica* is comprised of an assemblage of irregularly-shaped and micrometre sized aragonitic basic mineral units (white stars in Fig. 1A), that are larger in the outer shell layer compared to basic mineral units of the inner shell layer (this study and Casella et al., 2017). An irregular network of thin biopolymer fibrils interconnects these basic mineral units (Casella et al., 2017). The skeleton of the modern stony coral *Porites sp.* consists of an assemblage of spherulites consisting of aragonitic needles and fibrils (white star in Fig. 1B). These grow radially outward from an organic template present at aragonite nucleation sites: the centres of calcification (white dots in Fig. 1B; Griesshaber et al., 2017). As skeletal growth proceeds, aragonite crystallites increase in size, and form thin fibres that are bundled into loosely co-oriented units (framed in white and yellow in Fig. A1A; Griesshaber et al., 2017). When sectioned in 2D, spherical, irregularly-shaped entities are obtained (yellow stars in Figs. A1B, A1C), which are cut-off from each other by cavities. The shell of the modern gastropod *H. ovina* consists of aragonite with two different microstructures (Figs. 1C, 1D, A2A): prisms and nacreous tablets



(nacre). Aragonite prisms form the outer shell layer (yellow stars in Figs. A2A, 1C), while aragonite nacreous tablets constitute the inner shell layer (white stars in Figs. A2A, Fig 1D). The prismatic units show a gradation in size that decreases towards the rim of the outer shell. Accordingly, large aragonitic prisms are within the central part of the shell, next to nacreous aragonite. Nacreous tablets in *H. ovina* are stacked and form columns (Fig. A2A). The shell of the modern bivalve

*M. edulis* contains arrays of highly co-oriented calcite fibres (yellow stars in Figs. 1E, A2B; Griesshaber et al., 2013) along the outer shell part, while the inner shell layer consists of nacreous aragonite (white star in Fig. A2B). Aragonitic tablets in *M. edulis* (white star in Fig. 1F) are grouped in a sheeted, 'brick-wall' arrangement (Fig. 1F; Griesshaber et al., 2013).

### 3.2 Microstructure and texture of *hydrothermally altered* bivalve, gastropod and coral skeletons

The shells and skeletal elements of modern *Arctica islandica*, *Porites sp.*, *Haliotis ovina* and *Mytilus edulis* were subjected
to laboratory-based hydrothermal alteration. Experiments were carried out at 175 °C in the presence of a Mg-rich fluid simulating burial water. Experiment durations varied between 1 and 35 days (Fig. A3).

The amount of newly-formed calcite was determined by Rietveld analysis of XRD data (Fig. A4). Diagrams of calcite content versus experimental time (Fig. 2) demonstrate the difference in replacement kinetics between biogenic calcium carbonates and inorganic calcite and highlight the profound influence of the biogenic microstructure on carbonate
replacement reactions. In hydrothermally altered aragonitic *A. islandica* shells new calcite formation starts after 4 days of alteration and progresses constantly. After 7 days of alteration most shell aragonite was replaced by calcite (Figs. 2A, A4A; Casella et al. 2017). In contrast, the hard tissue of *Porites sp.,* and of *H. ovina* respond differently to alteration. Replacement of their biogenic aragonite by newly-formed calcite is significantly slower compared to that occurring in the shell of *A. islandica*, such that after 35 days of alteration only 20 to 30% of biogenic aragonite is replaced by inorganic calcite (Figs.
2B, 2C, A4B, A4D). For all investigated microstructures, the amount of newly formed calcite is not a continuous function of time.

Microstructure and phase characterisation were carried out with EBSD. The results are presented as EBSD band contrast (Figs. A5 to A8A), colour-coded orientation maps (Figs. 3 to 5, A8B) and corresponding pole figures (Figs. 3 to 5). EBSD band contrast is a grey scale component that illustrates the strength of the diffracted signal for each measurement.
Thus, when mineral material is hit by the electron beam, the backscattered signal is high and light grey colours form the image. When an organic component is scanned, the backscattered diffraction signal is absent, and the band contrast measurement image is black. Carbonate mineral co-orientation strengths are given as MUD values (e.g., Casella et al., 2017, 2018a, 2018b). These are derived from pole density distributions and are quoted for each EBSD scan. Figures 3 to 5, and A5 to A8 show the differences in microstructure and texture between pristine and the most advanced stage of alteration carried
out in this study (35 days, at 175 °C in a Mg-rich fluid). At these conditions aragonite prisms in the shell of modern *A. islandica* (Fig. A5A) are quickly and almost completely replaced by inorganic calcite (Fig. A5B). In the modern shell, aragonite prisms are surrounded by a thin network of organic fibrils. These are easily destroyed with hydrothermal alteration, and space is created for fluid percolation and a pervasive and quickly progressing replacement of the biogenic aragonite by



inorganic calcite. Calcite nucleation and growth in *A. islandica* shells start after a dormant period of about 4 days (Fig A4A; Casella et al., 2017), however, once started, the replacement progresses readily to completion. In the outer shell layer the replacement of aragonite is completed with the development of large and randomly oriented calcite grains, while, in denser shell areas, patches of biogenic aragonite are preserved, containing features of the original biogenic microstructure and

texture.

In contrast, acicular aragonite in *Porites sp.* displays a different behaviour during alteration. Even after alteration of 35 days only a minor parts of the coral skeleton are replaced by calcite (Figs. 2B, 3B to 3E, A5D). Our results show that the alteration fluid enters the coral skeleton predominantly at centres of calcification (Figs. 3B, 3D, A5D). New calcite formation starts mainly at these sites and proceeds from there into the skeleton. As Fig. 3D demonstrates, even after

alteration for 35 days at 175 °C in the presence of a Mg-rich fluid, the acicular microstructure with its aragonite needles bundled into co-oriented units is still preserved. However, a decrease in MUD value from 41 in the pristine (Fig. 3A) to an MUD of 13 (Fig. 3E) in the altered shell is the only sign of alteration, as the decrease in MUD indicates overgrowth of new aragonite with a lower degree of crystallographic co-orientation. With progressively longer alteration large and randomly oriented calcite crystals develop in the coral skeleton (Figs. 3B, 3C, 3D, A5D). This calcite has high MUD values (Figs. 3D)

similar to single crystalline calcite precipitated from solution (Nindiyasari et al., 2015; Casella et al., 2017).

Figures 4B and A6B show that after 35 days of alteration in the presence of a Mg-rich fluid at 175 °C, the highly porous prismatic aragonite shell layer of modern *H. ovina* (Figs. 4A, A6A) is completely replaced by inorganic calcite. Aragonite prisms in the pristine shell are encased by a network of biopolymer fibrils which are readily destroyed by hydrothermal alteration. A significant amount of space becomes available for fluid infiltration, which results in extensive

overprint and a rapidly progressive replacement of the biogenic aragonite by inorganic calcite. In contrast, the nacreous shell layer of *H. ovina* is little affected. There is no major change between pristine and altered *H. ovina* nacre, neither in carbonate phase, nor in microstructure or in MUD value (Figs. 4C, D and A6C, D). However, it should be noted that even though there is a resemblance in basic mineral unit morphology and size, the existence of primary porosity, and the fabric of occluded biopolymers between the prismatic shell parts of *H. ovina* and *A. islandica*, the kinetics of carbonate phase replacement is

distinct for the two microstructures (Figs. 2A, 2C). While in *A. islandica* shell replacement between carbonate phases is rapid and extensive, it is slow and patchy in the prismatic shell layer of *H. ovina*. In the gastropod *H. ovina* it can be observed that prismatic shell areas are completely replaced by calcite, while in other shell regions some aragonite is still preserved and frames the newly-formed calcite grains (Fig. A10B). In addition, the difference between pristine and altered prismatic aragonite in *H. ovina* (compare pole figures and MUD values of Figs. 4A and 4D) is such that in the altered shell

the size of aragonitic prisms increases while the strength of aragonite co-orientation decreases. This was observed in the pole figures and the decreased MUD value (compare Fig. 4A with right hand part, framed in green with Fig. 4D).

The comparison of Figs. 5A to 5C and Figs. A7A to A7B and A8 demonstrates that alteration of *M. edulis* calcite fibres at 175 °C, in the presence of a Mg-rich fluid, results in severe distortion of the fibres. In the pristine shell each calcite fibre is wrapped in an organic sheath. These decompose during alteration and leave space for fluid permeation and inorganic





calcite reprecipitation. Crystal co-orientation strength for fibrous calcite decreases markedly, from a MUD value of 381 in pristine to 79 in altered shells. In contrast to the calcitic fibrous shell microstructure, and similar to *H. ovina* nacre, after 35 days of alteration, 175 °C and in the presence of a Mg-rich fluid there is no significant change in microstructure between pristine and altered *M. edulis* aragonite nacre (Figs. 5B, D, A7C, A7D). In altered *M. edulis* partial amalgamation of nacre

tablets was observed (yellow stars in Fig. A7D), and a slight decrease in aragonite crystal co-orientation strength (pristine nacre: MUD 129; altered nacre: MUD 105) was determined.

### 3.3 The dynamic evolution of hydrothermal alteration

Major changes in microstructure which develop during different alteration times are depicted in Figs. A9 to A11. For all investigated skeletons one of the first steps in the alteration process is an increase in basic mineral unit dimension relative to

that in the pristine skeleton. In the *Porites sp.* coral skeleton, individual spherulites grow together (white stars in Fig. A9B, A9C) and form large and compact entities. Even though the alteration fluid accessed the skeleton from all sides, calcite formation in *Porites sp.* starts within the skeleton and proceeds outward toward the outer perimeter of the hard tissue (Fig. A9D). An increase in mineral grain size with progressive alteration can also be observed for both microstructures that constitute the shells of *H. ovina* (Figs. A10) and *M. edulis* (Figs. A11). As the organic sheaths around the basic mineral units

decompose, space becomes available for new mineral formation. Aragonite prisms, calcite fibres, and nacreous tablets increase in size until they abut each other. In particular, the nacreous microstructure, irrespective of its specific arrangement into columns or sheets, and the calcite fibres form compact entities in response to alteration. In addition to an increase in fibre dimension, *M. edulis* calcite fibre morphology becomes highly distorted with progressively longer alteration. Even though the prisms of the prismatic shell layer in *H. ovina* also amalgamated, due to their slightly rounded and irregular

morphology voids get entrapped. The resulting structure becomes more compact than the pristine nacre, but not as compact as possible if smaller pores had formed within the tablets.

A further characteristic caused by hydrothermal alteration is the significant rise in porosity within individual basic mineral units (Fig. 6). Even though the latter grow together at their perimeters (Fig. 7) a multitude of nanopores develop within them due to decomposition of biopolymer fibrils, which were present in the pristine hard tissue (e.g., Griesshaber et

al., 2013; Casella et al., 2018a, 2018b). In contrast, as Fig. 8 shows, the inorganic calcite that forms from the altered biogenic aragonite is almost devoid of pores. The patches of pores that are visible within the calcite (white arrows in Fig. 8) are all residues of the incorporated altered biogenic prismatic aragonite. Our results indicate that major features of the mesoscale original microstructure are retained even at advanced stages of alteration (Fig. A12). In the shell of *H. ovina,* for instance, where prismatic aragonite is almost entirely replaced by calcite (Fig. A12), the original gradation in the basic mineral unit

size towards the rim of the outer shell layer is still present. Large newly formed calcite crystals (white stars in Fig. A12B) are within the central part of the shell next to nacreous aragonite and decrease in size towards outer shell areas (Fig. A12B) – as it is the case in the pristine aragonitic shell before alteration.



Our results highlight that among all investigated microstructures, the nacreous microstructures were most resistant to hydrothermal alteration for 35 days at 175 °C in a Mg-rich fluid, irrespective of tablet thickness or their mode of assembly (columns or sheets). It could be observed that replacement of biogenic nacreous aragonite by inorganic calcite takes place in stages with various microstructural and chemical intermediates. These are described in detail for *H. ovina* nacre, as

illustrated in Figs. 9-11 and A13-A15. Alteration of bivalve and gastropod nacre starts with the decomposition of organic biopolymers, which is followed by tablet amalgamation and the generation of increased porosity within the tablets. Ongoing alteration destroys the tablet assembly (blue stars in Figs. 9A, 9B) up to the complete obliteration of the nacreous structure (yellow stars in Figs. 9A, 9B, 10A, 10B). However, as the phase map in Fig. 9E shows a phase replacement of biogenic aragonite by inorganic calcite has not yet occurred at this stage of overprint. Thus, when altered, the microstructure is

destroyed first; replacement of one carbonate phase by another occurs subsequently (Fig. 9). During alteration in a Mg-rich fluid, a Mg-rich rim was always present at the phase replacement front, between the newly formed calcite and the highly overprinted nacreous aragonite (white arrows in Figs. 9A, 9D, Fig. A14, white arrows in Fig. A15A). Based on Mg-contents, in addition to the 'final' calcite, two high-Mg-calcite phases can be distinguished (Figs. 10, 11, A15), which segregate between the 'final' calcite (calcite with a low Mg-contents) and the overprinted aragonite that was not yet replaced by calcite

(Figs. 11, A15). The last step in the replacement of biogenic nacreous aragonite by inorganic calcite is the formation of low-Mg calcite, the 'final' calcite, which in the final stage of alteration constitutes the overprinted hard tissue. Despite the change from one carbonate phase into another, the newly formed calcite retained much of the original mesoscale morphology of the basic mineral units inherited from the pristine biogenic skeleton.

## 4 Discussion

Biomineralised tissue provides the bulk of fossil material which is used for geochemical analysis. As all fossil archives are overprinted to some degree, it is of major importance to identify those which are subject to minor and moderate degrees of overprint, as (1) these are the materials that still contain mostly primary information, and (2) the detection of extensive overprint does not pose a problem as that microstructure is either highly distorted or completely destroyed. The latter two characteristics are easily identified, while in contrast, microstructures with a low to moderate degree of overprint are difficult

to recognise and to detect. Accordingly, important questions which arise in this context are: what are the intermediate steps of alteration and diagenetic overprint? What is destroyed first, the original skeletal microstructure or the original mineralogical phase and, with this the geochemical information stored by the biogenic archive? In general, what determines the preservation potential of a fossil archive?

### 4.1 The process of overprint

Diagenetic overprinting of biogenic carbonates encompasses morphological and chemical changes that take place during post-mortem alteration. Fluids act as catalysts for the alteration reactions at fluid-rock contacts and allow the overprint



reactions to proceed at a rapid rate (Brand, 1994). This response is in contrast to solid-state alteration in dry systems, where overprint kinetics is much slower. Brown et al. (1962) have shown that replacement of aragonite by calcite at Earth surface pressure and temperature conditions is 10 orders of magnitude faster in the presence of water compared to dry conditions. Accordingly, with the death of the organism and burial in sediments biomineralised hard tissues become subject to

diagenetic overprint, to solvent mediated phase replacement (Cardew and Davey, 1985), and the coupled dissolution of the original material and the precipitation of a new product(s) (Putnis, 2002, 2009).

It has been shown for inorganic systems that coupled dissolution-reprecipitation is highly influenced by the availability of interfaces, the reactivity of the involved surfaces, and the extent and topological characteristics of the original and newly formed porosity (Putnis, 2002, 2009; Arvidson and Morse, 2014; Ruiz-Agudo et al., 2014). It is demonstrated for

rocks and minerals that a coupling of the two (sub)reactions takes place when the rate of dissolution of the original phase and the rate of crystallisation of the product is almost equal. This has the effect that coupled dissolution-reprecipitation of mineral replacement proceeds with preservation of the external shape of the primary mineral, and leads to formation of pseudomorphs (Xia et al., 2009a; Quian et al., 2010). If the coupling between dissolution and recrystallisation is balanced, delicate microtextural features are well preserved, such as twin boundaries (Xia et al., 2009b) or exsolution lamellae (Altree-

Williams et al., 2015).

It has been further demonstrated for inorganic materials that microstructural elements such as grain boundaries are of key importance for the overprint process. At the first stages of alteration, these form pathways for fluid infiltration and percolation through the material and ensure a pervasive replacement of the original mineral (Eschmann et al., 2014; Jonas et al., 2014). In inorganic systems, mass transfer along grain boundaries is an order of magnitude faster than through the

porosity which is generated as a result of the mineral replacement reaction itself (Eschmann et al., 2014; Jonas et al., 2015). Even though, in inorganic systems an interconnected pore system is also developed with progressive alteration (Putnis, 2002, 2009; Pollok et al., 2011; Ruiz-Agudo et al., 2014; Altree-Williams et al., 2015). In fact, the formation of porosity is a requirement for the progress of the replacement reaction itself, as it is the pore system which allows for the continuous communication between the bulk aqueous phase and the primary and secondary phases at the reaction front (Putnis, 2002,

2009; Etschmann et al., 2014). Pore formation also takes place as a direct consequence of the mineral replacement process itself, in cases when the molar volume change involved in the reaction is negative. A further source of porosity development during mineral replacement relates to the difference in solubility between the primary and secondary phases (Pollock et al., 2011). Porosity is generated when the primary phase is more soluble than the secondary phase as a small amount of the latter precipitates after dissolution of the former. In the case of carbonates, even though the solubility of biogenic aragonite is

higher than the solubility of inorganic calcite, the solubility difference is not large enough to compensate the positive volume change in the dissolution-reprecipitation reaction. A positive molar volume change of only 8.12 % is associated with the replacement of aragonite by calcite (Perdikouri et al., 2011, 2013).

Perdikouri et al. (2011) investigated the replacement of inorganic aragonite by inorganic calcite. The authors immersed inorganic aragonite in pure water and in solutions which contained calcium and carbonate, with the solutions



being saturated with respect to calcite and undersaturated with respect to aragonite. In experiments which were carried out in the presence of water, a replacement was not observable, even after an entire month, unless the solution temperature was equal or higher than 180 °C. Even at elevated temperatures there was only a narrow rim of aragonite replaced by some calcite overgrowth. The newly formed calcite was devoid of pores, hence there was no communication between the bulk

aqueous phase and the phases at the reaction front, and thus, the overgrowth sealed the aragonite and prevented progressive replacement. However, by using aqueous solutions containing calcium and carbonate Perdikouri et al. (2011) obtained different results. When the composition of the solution was *stoichiometric*, comparable results were obtained to the experiment with water: little replacement was observed and the formation of a non-porous calcite overgrowth. In contrast, in the presence of a *non-stoichiometric* solution, the amount of calcite overgrowth was still very small, however, a high degree

of replacement was achieved; an effect that was even more increased by the absence of calcium in the solution. Thus, the experiments of Perdikouri et al. (2011) demonstrate the importance of porosity and porosity generation for the progress of dissolution-reprecipitation reactions and allude to at least one fundamental difference between biologic and inorganic hard materials. In the absence of primary porosity and/or secondary porosity that should have been generated at early stages of alteration is attributed to the positive molar volume change involved in the aragonite by calcite replacement. The only

porosity that might be generated in inorganic systems will arise from the minor difference in solubility between aragonite and calcite. As the solubility products of the two main carbonate phases are similar, little porosity formation takes place, and consequently, the replacement of inorganic aragonite by inorganic calcite occurs at a slow rate and is significantly less pervasive as it is in the case of biogenic aragonite.

Biological hard tissues are hierarchically organised and are composite materials where at all scale levels there is an

20 interlinkage of biopolymers with minerals. The degradation of these biopolymers, being occluded within and between the basic mineral units of the hard tissue provides the necessary network of interconnected porosity (Figs. 6, 7, 8, A9, A10, A17). Even more, the porosity network not only facilitates alteration, it drives and accelerates it as it allows for pervasive circulation of the alteration fluids within the skeleton. Our results show, that for biological carbonate tissue the presence of primary (inherent) and secondary (induced) porosity together with the characteristics of the porosity network determines the

25 kinetics and extent of the alteration. Furthermore, the transient character of porosity additionally influences mineral replacement reactions, apart from porosity generation, porosity closure and porosity coarsening in biological material are widespread phenomena. These modify the geometry of the porosity network, increase its tortuosity, reduce its permeability, and thereby affect the mass transfer at the interface between the bulk solution and the original mineral phase and hinder physicochemical re-equilibration.

Porosity characteristics are different for the different microstructures investigated in this study (Fig. 1). Primary porosities are present in the shell of *Arctica islandica* and in the prismatic shell layer of *Haliotis ovina*. It is important to note that the skeleton of the coral *Porites sp.* is compact. However, the coral skeleton has a particularly high surface area as the skeleton consists of various combinations of vertical and transverse elements, with most of these being developed as thin lamellae. Basic mineral units which comprise these skeletal elements consist of irregularly organised clusters of closely





packed aragonitic needles. The centres of calcification are the primary pores in the skeleton of *Porites sp.*, however, these are in general not interconnected, and thus, do not facilitate transfer of solutes to and away from the reaction front to a large extent. Stacks of calcite fibres in *Mytilus edulis* and the nacreous tablet arrangements in *M. edulis* and *H. ovina* are the most compact microstructures investigated in this study. These materials lack primary porosities. Nonetheless, when altered, the extent of alteration-induced secondary porosity is high in the nacreous tablets, as the occluded intra-tablet membranes and inter-tablet fibrils are decomposed and create space for fluid circulation.

## 4.2 The effect of microstructure on alteration

A still unsolved problem in palaeoenvironmental reconstruction is the assessment of the extent of diagenetic overprint that compromises the fidelity of geochemical proxies. One strategy is to use numerical approaches for the quantification of the extent of diagenetic alteration that are based on the comparison of element to Ca ratios and associated partition coefficients, and the comparison between isotope compositions of the pore fluid and the precipitate (Regenberg et al. 2007 and references therein). In a previous study (Casella et al., 2017), we reported experimental data for *Arctica islandica* shell material for the replacement reaction of biogenic aragonite by inorganic calcite. In the present study, we extend our previous work with the investigation of additional (mainly aragonitic) carbonate skeletons, and thus, other mineral fabrics. One of the major aims of this study is the reliable identification of the first stages of alteration and the attempt to qualitatively assess diagenetic alteration based on microstructural reorganisation. For these targets, we apply statistical grain area evaluation and develop this approach as a qualitative tool for the detection of moderate diagenetic overprint.

Figures 12 and A16 show relative frequency and grain area (basic mineral unit in the case of biological hard tissues) diagrams for pristine and the most altered (alteration for 35 days, at 175 °C, in Mg-rich fluid) skeleton equivalents for six microstructures. Grain area data is obtained from EBSD measurements. A grain is defined by a misorientation angle relative to neighbouring grains that is larger than a critical value, the critical misorientation value (see Chapter 2.2.3). Griesshaber et al. (2013) determined empirically that a critical misorientation value of 2 ° best suits the microstructure of modern carbonate biological hard tissues to differentiate between individual basic mineral units (e.g. fibres, tablets, prisms, columns). Thus, we adopt a critical misorientation value of 2 ° to define a grain. Thus, adjacent grains are recognised as two individual grains when one unit is tilted relative to the adjacent unit by more than 2 °.

The compilation in Fig. 12 clearly demonstrates the influence of the biogenic microstructure to withstand or to yield to alteration. The relation between log (frequency) versus log (grain area) is linear for *A. islandica*, *M. edulis* calcite and *Porites sp*. aragonite, clearly an indication of fractal distribution for the microstructures of these skeletons.

The least difference in grain area change between pristine and most altered states was observed for *A. islandica* aragonite (Fig. 12A), while the most significant difference occurred for *M. edulis* fibrous calcite (Fig. 12E). For *Porites sp.* acicular aragonite and *H. ovina* prismatic and nacreous aragonite, we find a perceivable, however, only a small difference in grain-area size between pristine and the most altered states. For *M. edulis* nacre the majority of grain area data overlap. However, for this microstructure some large grains formed in the altered shell as well (Fig. A16).



As described in the results section, subsequent to the destruction of organic sheaths, membranes and fibrils, the amalgamation of basic mineral units is the next and highly drastic step in the overprint process. Inorganic mineral precipitation starts in cavities between the basic mineral units and in voids within them (e.g., Figs. 7, A17; Casella et al., 2018a, 2018b). It is important to note that this occurs prior to carbonate phase replacement, and thus, prior to inorganic calcite formation. With EBSD we not only measure patterns of crystal orientation but determine the mineralogical phases of the hard tissue. At this early stage of alteration crystallites that are deposited between the basic mineral units retain the phase of the host crystal and often even the crystallographic information of the mineral in the pristine skeleton. Thus, in aragonitic biogenic microstructures, inorganic aragonite will precipitate, while in calcitic biogenic microstructures inorganic calcite will form. Syntactic nucleation of a secondary phase that has the same mineralogical nature as the primary phase is prompted by the reduction of the energy barrier associated to heterogeneous nucleation in comparison to homogenous nucleation from a bulk aqueous solution. This barrier is further reduced as a result of a perfect match between the crystal lattice of the original and secondary phase. This energy barrier reduction explains the preference of inorganic aragonite formation on biogenic aragonite at the first stages of the alteration process, rather than the more stable inorganic calcite.

Due to its composite nature, biogenic aragonite is more soluble than inorganic aragonite and even more soluble than inorganic calcite. Thus, an aqueous solution in equilibrium with biogenic aragonite is supersaturated with respect to both, inorganic aragonite and inorganic calcite. This supersaturation is higher with respect to calcite, and as calcite nucleation on aragonite can be epitactic, the much better matching through the interface makes it more likely that nucleation and growth of inorganic aragonite occurs on biogenic aragonite. Hence, even though calcite is the more stable phase at Earth surface pressure and temperature conditions, free energies and solubilities of the two carbonate phases are close enough that the lower energy barrier associated with epitactic nucleation kinetically favours the formation of new aragonite on the surface of the pre-existing aragonite (Fernandez-Diaz et al., 2009; Roncal-Herrero et al., 2017; Cuesta Mayorga et al., 2018). This has been also observed in nature. Hover et al. (2001) report early diagenetic overprint of foraminifera and green algae skeletal hard tissues and demonstrate that the overprint mechanism is the coupled process of dissolution and reprecipitation. The authors find thin overgrowths on the mineral units of the original hard tissues and show that the precipitated material is largely similar in composition and structure to that of the host crystallites.

Accordingly, aspect ratios of the basic mineral units change as their original morphologies get distorted (Figs. 7, A8, A17) and compaction of the hard tissue is the result (e.g., nacre tablets). However, even though already altered, at this early stage of alteration the gross microstructure of the shell or skeleton is not modified to a large degree. We observe that alteration occurs in two stages: (1) related to the original carbonate phase of the hard tissue *overgrowth* and nucleation of inorganic aragonite or calcite in voids and pores without a major destruction of the original microstructure, and (2) phase replacement, new formation with distortion of the original microstructure up to its complete destruction. These processes involve the constant rearrangement of porosity, which in this case is driven by the free energy reduction associated with the increase in the volume/surface ratio of the basic mineral units.





We observed the above described features for all investigated microstructures (Figs. 12B to 12F) except for the prismatic aragonitic microstructure of the shell of the bivalve *A. islandica* (Fig. 12A). Aragonitic prisms in *A. islandica* shell are small and are embedded in a network of biopolymer fibrils (Casella et al., 2017). The thin fibrils are easily destroyed when altered and leave behind a network of voids and cavities which facilitate fluid infiltration and permeation through the

shell. The large number of small basic mineral units gives rise to exceedingly large surface areas where the fluid can get into contact with the mineral. Carbonate phase alteration kinetics in *A. islandica* shell is sluggish at first, however, when the nucleation barrier is overcome and the alteration process is started, it proceeds very rapidly (Figs. 2A, A4A; Casella et al., 2017). Thus, overgrowth of inorganic aragonite in voids and basic mineral unit amalgamation might well be masked by the almost instantaneous replacement of biogenic aragonite by inorganic calcite in the microstructure of *A. islandica* shells. The

high volume of interconnected porosity in *A. islandica* not only explains that alteration becomes vital after only a short time in contact with diagenetic fluids, the topological characteristics of porosity facilitates the coupling between the rate of aragonite dissolution and calcite reprecipitation. This in turn, explains the little difference in mineral grain area found in the hard tissue of *A. islandica* between the pristine and the most altered states.

      In contrast, *M. edulis* calcite shows the most significant difference in grain area between the pristine and the most

overprinted states (Figs. 12E). When altered, the morphology of calcite fibres was distorted (Fig. A8A); fibre amalgamation was substantial and led to the formation of large and highly irregularly-shaped mineral units (Fig. A8B). In the pristine state, calcite co-orientation strength is high in *M. edulis* calcite, a single-crystal-like distribution of c- and a*-axes is present (Figs. 6 and 7 in Schmahl et al., 2012). Hence, many neighbouring calcite fibres are highly co-aligned, a circumstance that favours the amalgamation of similarly oriented fibres (Fig. A8B). The nacreous shell layer in *M. edulis* was little affected by

alteration (Fig. 12F, Fig. A16A, A16B), even though nacre tablet amalgamation was well perceivable. The nacreous shell part grows into a compact entity and becomes sealed and protected against fluid infiltration. This explains the observation of remnants of nacreous shell areas surrounded by calcite (Brand, 1994) as well as the increased prevalence of the nacreous shell layer of *M. edulis* relative to calcitic shell layers in seashore sediments.

      Nacre in *H. ovina* behaves slightly differently when hydrothermally altered (Figs. 12D, A16A, A16C). In *H. ovina*

nacreous tablets are assembled in columns, and tablet dimensions are smaller than those present in *M. edulis*. As for both, *M. edulis* and *H. ovina*, nacreous tablets are encased by organic sheaths, compared to *M. edulis* nacre, nacre in *H. ovina* has a larger organic-mineral interface and mineral surface area per volume fraction of shell. Nacreous tablet amalgamation and compaction of the nacreous shell layer occurs in the shell of *H. ovina* as well. In contrast to *M. edulis*, *H. ovina* nacre exhibits a distinct increase in grain size in the altered hard tissue. Due to the larger interface and surface area in *H. ovina*

nacre alteration fluids infiltrate the shell more profusely, and dissolution/reprecipitation occurs to a higher extent. Hence, overprint becomes more significant and evident. The same argument holds for prismatic aragonite found in *H. ovina* (Fig. 12C) and acicular aragonite in *Porites sp.* (Fig. 12B), where prior to replacement of biogenic aragonite by inorganic calcite, basic mineral units increase in size in the altered skeleton. It is important to note that this size increase is accompanied in *H. ovina* and *Porites sp.* by partial closure of the porosity, and the newly formed calcite is completely devoid of pores (Figs. 8,



10, 11, A5B, A10B). The partial closure of pores explains the low degrees of replacement that is reached by these hard tissues even after long alteration periods.

Our study clearly shows that of the investigated aragonite microstructures the nacreous tablets are the most resistant to replacement by calcite, irrespective of the assembly pattern of the tablets in columns or sheets. Porosity closure and basic

mineral unit (nacre tablet), amalgamation recasts at first completely the original microstructure, however, with the preservation of the original phase (Figs. 9A, A17A, A17B). Hence, even though nacreous aragonite is still preserved as aragonite, it is an overprinted aragonite that, most probably, holds little of the original microstructural or geochemical signature. With increasing alteration, the 'remoulded' aragonite finally becomes replaced by inorganic calcite. In general, we find that in our alteration experiments the microstructural signature is lost first, prior to a complete loss of the original

mineralogical phase while the geochemical information is retained in the mineral. When alteration takes place in a Mg-rich fluid, it can be observed that at the original material-product interface, in addition to the 'final' inorganic, low-Mg calcite, two other calcite phases are present. These can be distinguished by their Mg-content (Figs. 9A, 11). We clearly see an evolution in fluid composition after hydrothermal alteration, an evolution in cation-anion exchange between the alteration fluid, the overprinted original and the newly-formed carbonate products.

**4.3 Implications for preservation of carbonate skeletons in the fossil record**

Several studies have shown that in modern cold and warm water environments aragonite dissolution takes place during burial diagenesis (e.g., Cherns et al., 2008 and references therein). It has been further demonstrated that in Palaeozoic marine faunae taxa with calcitic skeletons prevail. This is an indication of preferential loss of aragonitic shells and skeletons due to dissolution during diagenetic overprint (e.g., Wright et al., 2003; James et al., 2005). In addition to preferential carbonate

phase preservation, experimental studies document that the microstructure of the biogenic skeleton influences fossil preservation (e.g., Harper, 1998, 2000; Kidwell, 2005), leading to a possibly distorted notion of palaeoecological and evolutionary patterns. Accordingly, laboratory-based hydrothermal alteration experiments accounting for microstructural as well as mineral phase variability offer important insights into the fate of carbonate hard tissues a) during shallow burial early dissolution, and b) surviving dissolution and preservation in the fossil record. Do we see resemblances between the

microstructural, chemical outcome of our alteration results and microstructural and geochemical features of fossilised hard tissues?

It is remarkable, that even though our experiments lasted only 35 days and were carried out at a single temperature and performed in the presence of only one type of alteration fluid there is much overlap between our experimental results and of carbonates which underwent diagenesis. Several decades ago Friedman (1964) and Land (1967) reported on the early

diagenesis of skeletal carbonates and carbonate sediments exposed to marine waters. The biological carbonates retained their original mineralogical and textural characteristics. Biogenic aragonite was dissolved for the reprecipitation of low-Mg calcite, with high-Mg calcite being an intermediate phase. Mg is removed from high-Mg calcite to yield low-Mg calcite, and, on a micrometer scale, without textural change (Friedman, 1964). Land (1967) observed that skeletal aragonite is altered





much quicker relative to non-skeletal aragonite grains. Brand (1989) investigated the replacement of biogenic aragonite by calcite in fossil molluscs (Boggy Formation, Oklahoma, U.S.A.) for an assessment of the degree of diagenetic overprint and the possible detection of the least-altered shells. With screening the mineralogy, microstructure and chemical composition it was detected that primary nautiloid aragonite is gradually replaced by diagenetic low-Mg calcite. During initial stages of

alteration nacreous tablets fused to larger units (Brand, 1989). With further alteration amalgamated nacreous aragonite was replaced by fine- or coarse grained low-Mg calcite. Brand (1989) noted that the original aragonite determined the elemental and isotopic composition of the calcite in the diagenetically altered shells. Furthermore, the author reported that grain size and surface area play an important role for the process of overprint. Diagenetically overprinted aragonitic corals were investigated by Sorauf (1980) and Tomiak et al. (2016). The authors observed that during early diagenesis, subsequent to

decomposition of organic matrices, aragonitic units formed through fusion of pristine skeletal elements. Pore space became filled, prior to burial, with aragonite needles growing syntaxially on existing biogenic aragonite. Subsequent submarine diagenesis leads to recrystallisation of fibrous aragonite to intermediate, micritic high-Mg calcite. Tomiak et al. (2016) and Regenberg et al. (2007) observed formation of new mineral overgrowth during early diagenesis of coral aragonite and planktonic foraminifera calcite. The latter retained, at first, and corresponded to the carbonate phase of the original pristine

skeleton. Wardlaw et al. (1978), Sandberg and Hudson (1983), and Martin et al. (1986) describe the influence of skeletal porosity as conduits for alteration fluids during diagenesis. As the replacement of aragonite by calcite is driven by the greater solubility of aragonite relative to that of calcite, the diagenetic pore fluid is undersaturated with respect to aragonite while it is supersaturated with respect to calcite (Maliva et al., 2000) during the replacement reaction. Hendry et al. (1995) proposed on the basis of supersaturation variation a 'two-water diagenetic system' with a slow moving (at the dissolution-

repreciptation front) and a relatively fast moving (bulk pore water) alteration fluid.

In summary, some major steps of alteration could be detected in our experiments: decomposition of biopolymers, secondary porosity formation, amalgamation of mineral units, and chemical evolution of the alteration fluid. These were also observed in nature. Our experiments, which lasted only for a short time compared to geologic time scales, show major and drastic steps of alteration taking place at very initial time periods of the overprint process.

**5 Conclusions**

Biogenic carbonate hard tissue form the basis of studies of past climate and environmental change. However, the greatest challenge that all biological proxies face lies in their capacity to retain their pristine signatures. With death of the organism, diagenetic overprinting starts immediately during which the original, biogenic signals are replaced by inorganic features. We investigated the behaviour of six biogenic carbonate samples and their associated microstructures at different degrees of

hydrothermal alteration in order to evaluate their capacity to withstand alteration and thereby estimate their ability to be preserved in the fossil record. The main conclusions are:



1. Alteration of biogenic aragonite to inorganic calcite is fastest in hard tissues which contain primary porosity and are composed of irregularly shaped basic mineral units embedded in a network of biopolymer fibrils. The latter are easily destroyed and provide, together with primary pores, ample space for extensive fluid infiltration into and percolation through the hard tissue. This mode of overprint is observed for the prismatic shell layer of the gastropod *Haliotis ovina* and for the shell of the bivalve *Arctica islandica*. Overprinting of these hard tissues is fast and completed with the formation of irregularly shaped and randomly oriented calcite units.

2. The slowest alteration kinetics can be observed when biogenic nacreous aragonite is replaced by inorganic calcite, irrespective of the mode of assembly of nacre tablets. Alteration proceeds in four subsequent stages: (a) decomposition of biopolymers and formation of secondary porosity, (b) lateral and longitudinal amalgamation of nacre tablets, (c) formation of a compact zone within the hard tissue with an entirely erased microstructure: The latter occurred at the alteration front. However, the original bioaragonite phase is still retained, and (d) replaced by inorganic calcite.

3. The acicular microstructure of the stony coral *Porites sp.* is highly resistant to alteration. During alteration aragonite needles fuse and form a compact aragonitic fabric, still retaining some morphological aspects of the pristine microstructure. Replacement of biogenic aragonite by inorganic calcite starts within the coral skeleton at centres of calcification and proceeds from the latter inward into the hard tissue.

4. For the investigated hard tissue we observe first the destruction of the microstructure and second, the replacement by newly formed calcite.

5. Alteration in a fluid enriched in Mg results in the development of a high-Mg seam between the altered, compact aragonite and the newly formed calcite. Progressively decreasing Mg-concentrations allow for clearly trace the chemical evolution of the alteration fluid at the biogenic aragonite –inorganic calcite interface.

6. Statistical evaluation of differences in grain area of pristine and altered skeletal equivalents demonstrates an increase in grain area within the altered hard tissues relative to that found for the pristine skeleton. Hence, even though at very early stages of alteration the original phase is retained, overprint starts with the formation of overgrowths. This is most pronounced in the calcitic shell layer of *Mytilus edulis* and is least for the grains which constitute the shell of *Arctica islandica*. Thus, in the case of aragonitic tissue the survival of biogenic aragonite cannot be used as a distinct indicator for pristine elemental and isotope signals. Statistical evaluation of grain area (basic mineral unit) values is a promising new tool for the estimation of the degree of diagenetic overprint.

**Acknowledgements**

We thank the German Research Council (DFG) for financial support in the context of the collaborative research initiative CHARON (DFG Forschergruppe 1644, Grant Agreement Number SCHM 930/11-1).



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



**Figures and Figure captions**

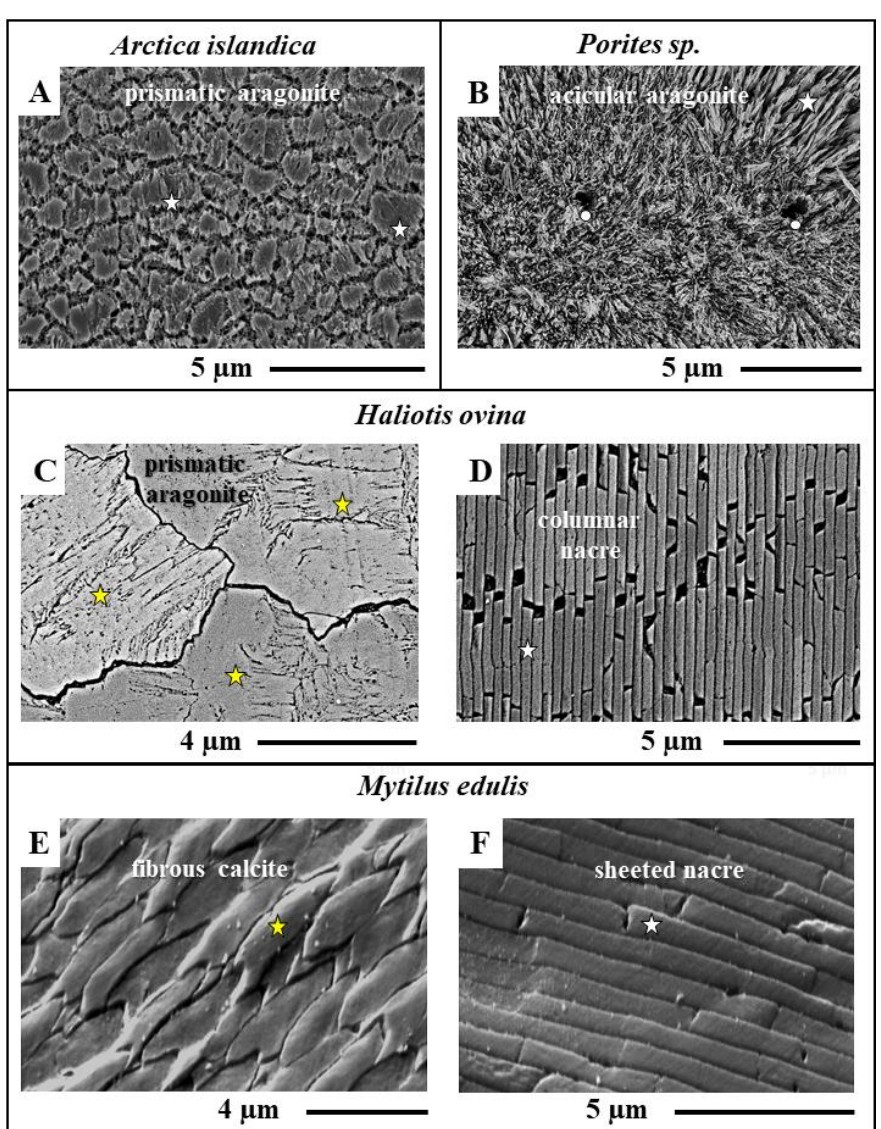

**Figure 1: SEM micrographs showing the characteristic microstructures of skeletons of the modern specimens of (A) bivalve**
5    *Arctica islandica***, (B) scleractinian coral** *Porites sp.***, (C, D) gastropod** *Haliotis ovina***, and (E, F) bivalve** *Mytilus edulis***. The shell of**
*A. islandica* **consists of an assemblage of irregularly shaped and sized aragonitic basic mineral units (white stars in (A)) which are**
**embedded in a network of biopolymer fibrils (this study and Casella et al., 2017). The acicular aragonitic skeleton of the modern**
**coral** *Porites sp.* **(white star in (B)) is composed of differently sized spherulites consisting of fibrils and needles. These grow**
**outward from an organic template which lines the mineral nucleation sites, the centres of calcification (white dots in (B)). Shells of**
10   **the gastropod** *H. ovina* **and the bivalve** *M. edulis* **comprise two distinct carbonate layers. The shell of** *H. ovina* **consists of**
**irregularly shaped and sized prisms (yellow stars in (C)) next to a nacreous shell layer with nacre tablets assembled as columns**
**(white star in (D)). The outer shell layer in** *M. edulis* **is formed by stacks of calcite fibres (yellow star in (E)), while the inner shell**
**layer is nacreous with nacre tablets arranged in a 'brick wall fashion' (white star in (F)).**





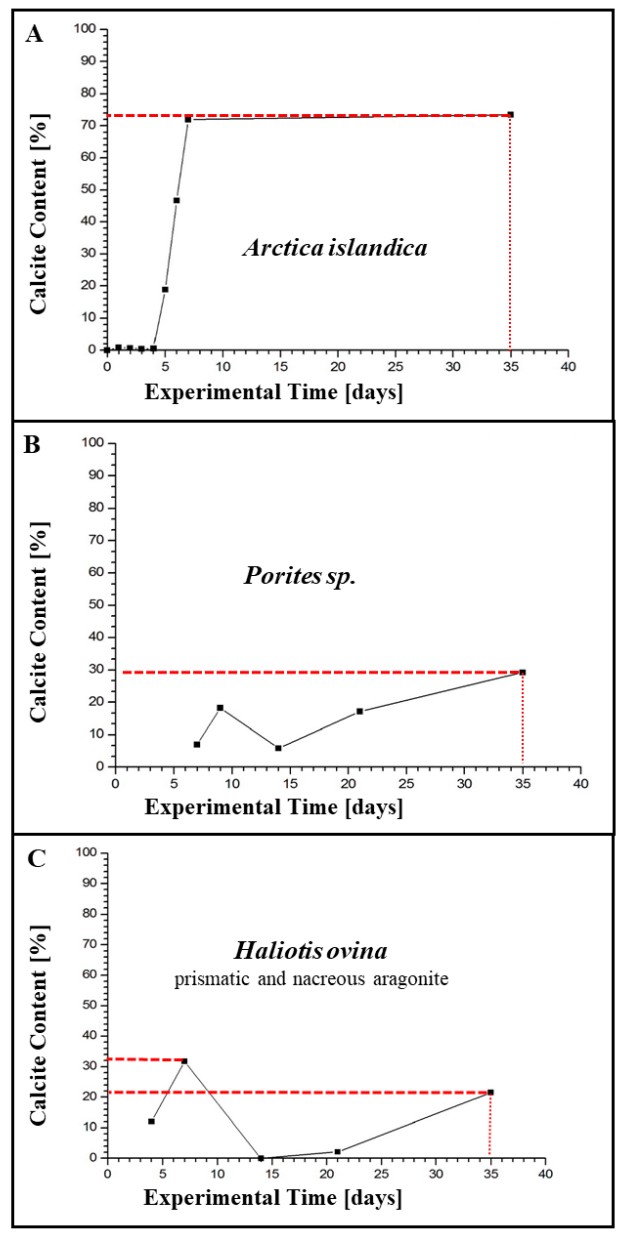

**Figure 2: Newly formed inorganic calcite content versus alteration time plots calculated from Rietveld analyses of XRD data obtained for hard tissues of (A)** *Arctica islandica***, (B)** *Porites sp.* **and (C)** *Haliotis ovina***. Red dashed lines indicate the percentage of newly formed calcite at 35 days of alteration as well as maximum contents of inorganic calcite for each investigated species. Differences in newly formed calcite contents among the three species clearly highlight the influence of the different microstructures on the replacement kinetics of biogenic carbonate by inorganic calcite.**



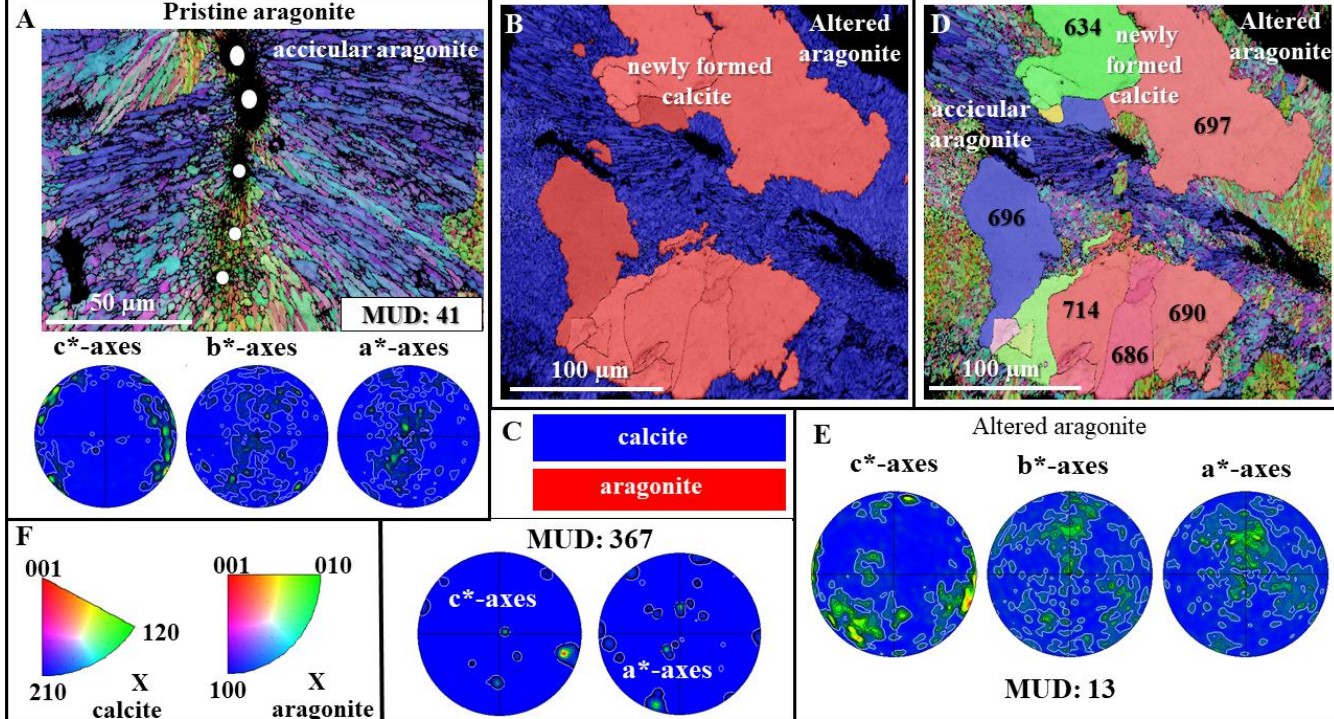

**Figure 3:** EBSD colour-coded orientation and phase maps with corresponding pole figures which depict the microstructure, texture and pattern of biogenic and inorganic carbonate phase distribution in pristine (A) and in hydrothermally altered (B, C, D, E) skeletal elements of the scleractinian coral *Porites sp.*. Alteration lasted for 35 days and was carried out at 175 °C in a Mg-rich fluid simulating burial water (100 mM NaCl + 10 mM MgCl$_2$ aqueous solution). EBSD colour codes are given in (F). The strength of crystal co-orientation is expressed with MUD values and is given at each EBSD measurement. MUD values for newly formed calcites (D) are written into the EBSD map and are given for most newly formed calcite crystals. Even though crystal co-orientation strength is moderate in the modern coral specimen (MUD: 41 in (A)), it decreases significantly in the altered coral skeleton (MUD: 13 in (D)). Co-orientation strength in newly formed calcite is exceedingly high, as high as that of calcite grown from solution (D).




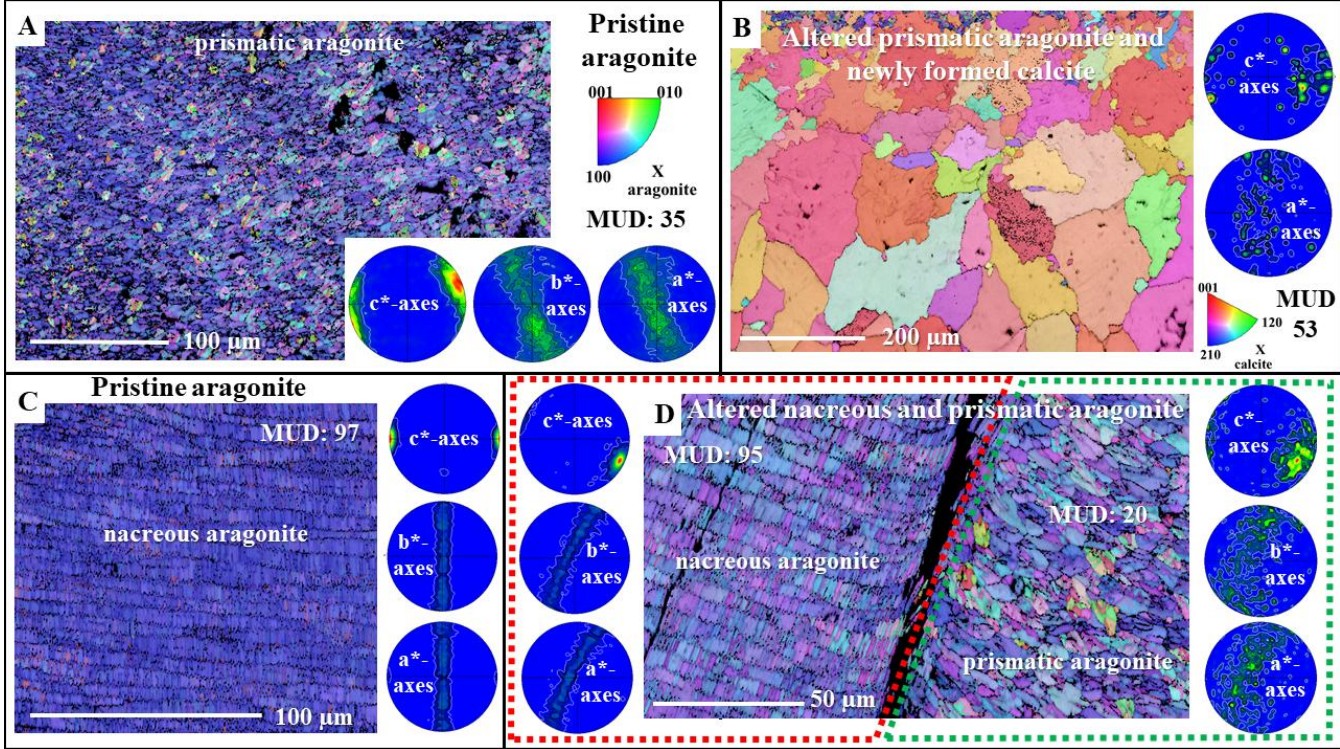

**Figure 4: EBSD colour-coded orientation and phase maps with corresponding pole figures show the microstructure, texture and pattern of biogenic and inorganic carbonate phase distribution in pristine (A, C) and hydrothermally altered (B, D)** *Haliotis ovina* **shells. Alteration lasted for 35 days and was carried out at 175 °C in a Mg-rich solution. Crystal co-orientation strengths, expressed with MUD values, are given at each EBSD map. Alteration for 35 days induces the replacement of large parts of prismatic biogenic aragonite (A) by inorganic calcite (B). However, in shell layers where replacement has not yet taken place, aragonitic prisms amalgamate and MUD values decrease (right-hand side of EBSD map shown in (D) framed with a green dashed line; compare to EBSD map, pole figures and MUD value shown in (A)). The nacreous part of the shell is little affected by alteration. Even when altered for 35 days the columnar microstructure is still well preserved (D). The MUD value of altered nacre (left-hand side of EBSD map shown in (D), framed in red) is very similar to that obtained for pristine nacre (C).**



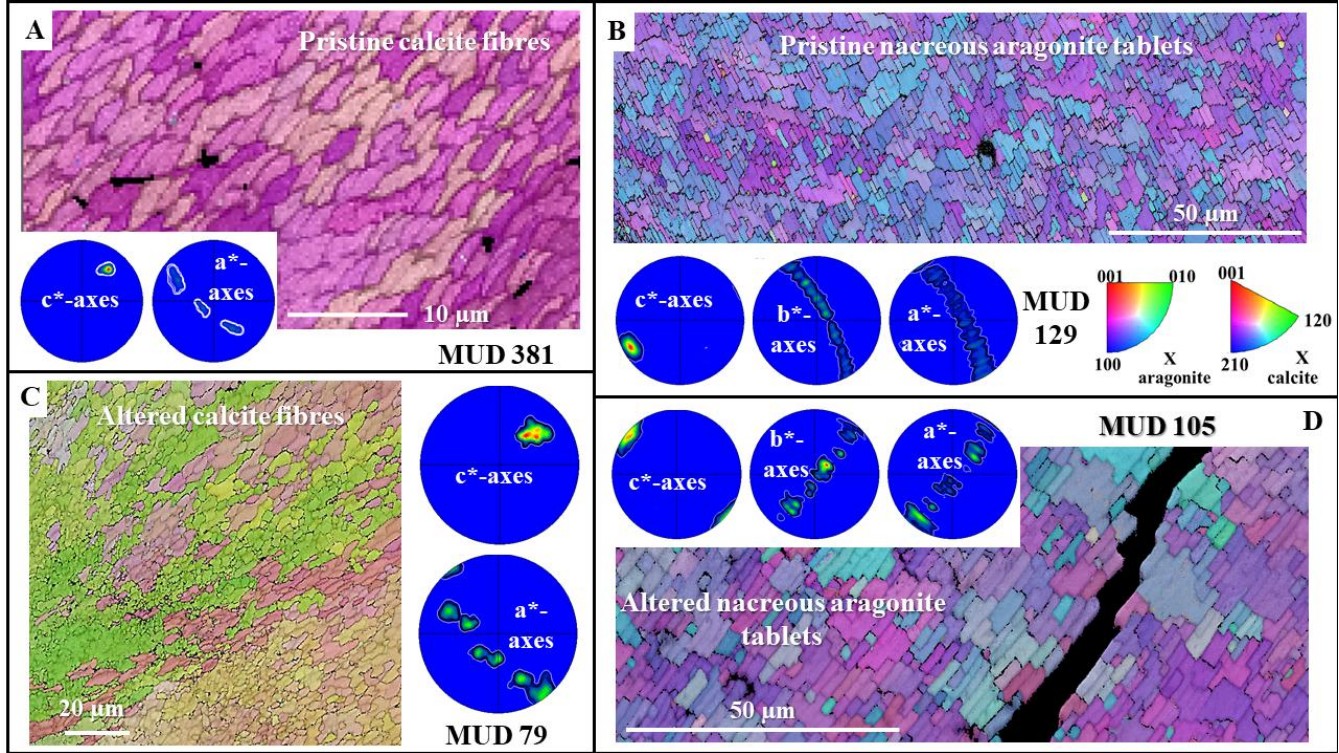

**Figure 5: Colour-coded EBSD orientation maps with corresponding pole figures depict differences in microstructure and texture between pristine (A, B) and hydrothermally altered (C, D) *Mytilus edulis* shells. Alteration lasted for 35 days and was carried out**
5   **at 175 °C in a Mg-rich fluid. The EBSD colour code used is shown in (B); crystal co-orientation strengths, expressed with MUD values, are given on each EBSD map. Hydrothermal alteration induces a significant change in pristine *M. edulis* calcite fibres (compare maps (A) and (C)). The strength of calcite co-orientation decreases from an MUD of 381 in the pristine (A) to a MUD of 79 in the altered shell (C), respectively. In the overprinted sample, morphology of calcite fibres is highly distorted due to profound fibre amalgamation. In contrast, nacre in *M. edulis* was little affected by the applied hydrothermal alteration conditions (D); a**
10   **slight decrease in MUD and sporadic tablet amalgamation can be observed, otherwise tablet morphology is not distorted.**




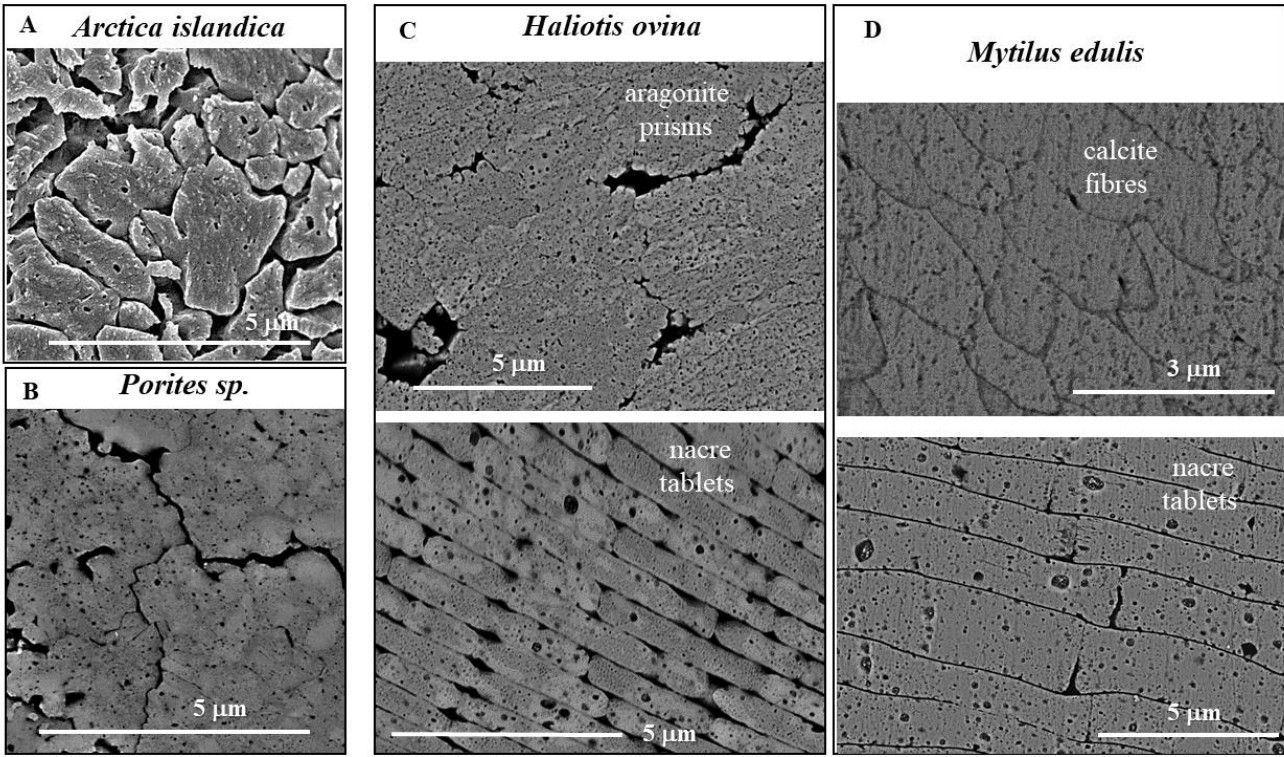

**Figure 6: Distortion of basic mineral unit morphologies, basic mineral unit amalgamation, and development of porosity in hydrothermally altered shells and skeletons. (A)** *Arctica islandica*, **(B)** *Porites sp.*, **(C)** *Haliotis ovina*, and **(D)** *Mytilus edulis*. **Hard tissue material was altered for 35 days at 175 °C in the presence of a Mg-enriched fluid.**



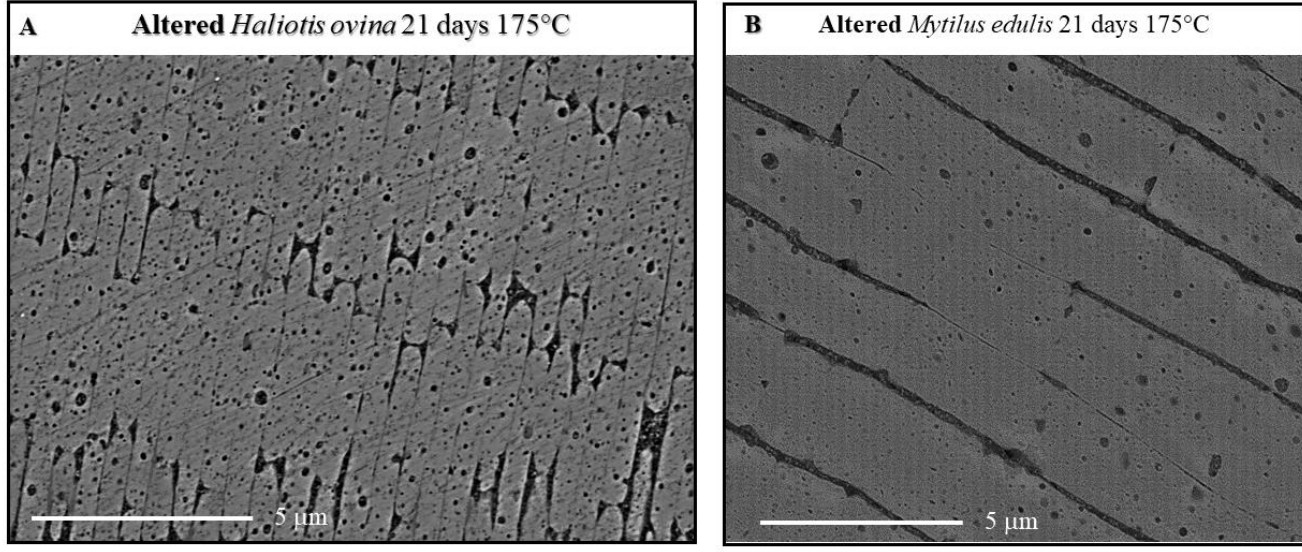

**Figure 7: Nacre tablet amalgamation in hydrothermally altered *Haliotis ovina* (A), and in *Mytilus edulis* (B) shells. Hard tissue material was altered for 21 days at 175 °C in the presence of an alteration fluid enriched in Mg.**



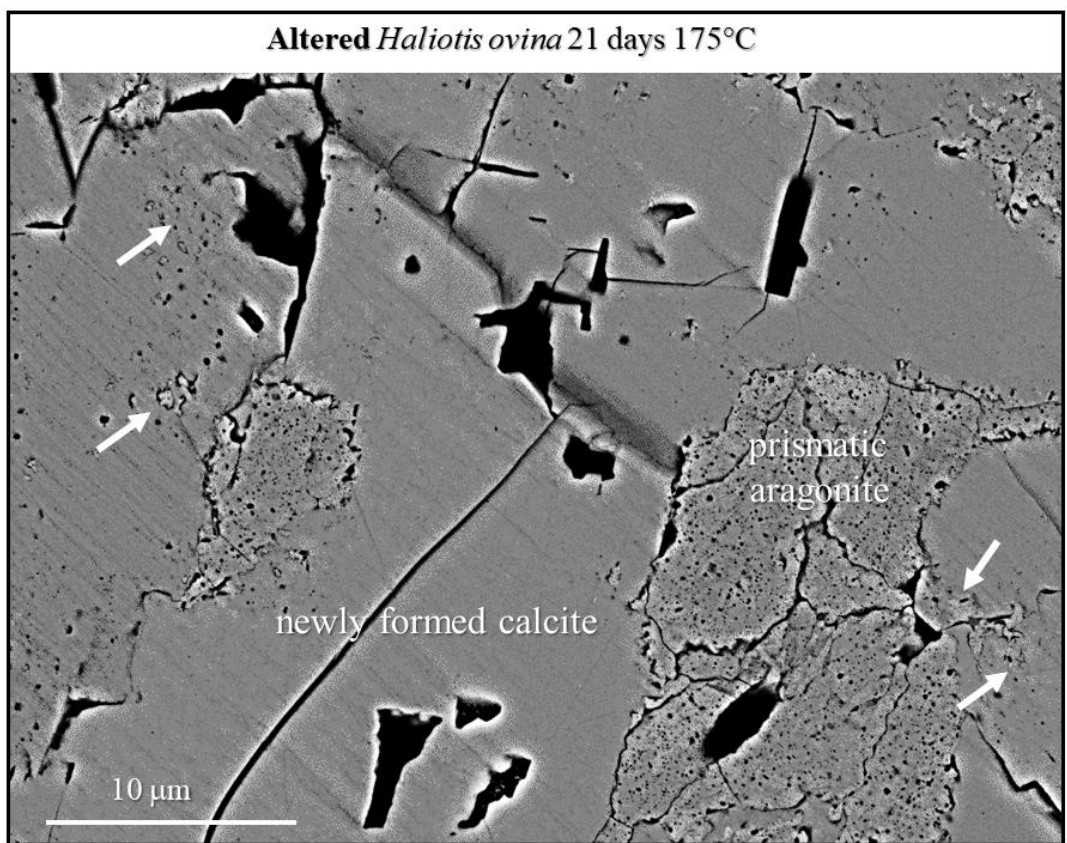

**Figure 8: Distinctness in porosity between hydrothermally-altered biogenic prismatic aragonite and newly formed inorganic calcite. White arrows point to the aragonite interspersed with calcite; the aragonite is not yet fully consumed and replaced by calcite.**



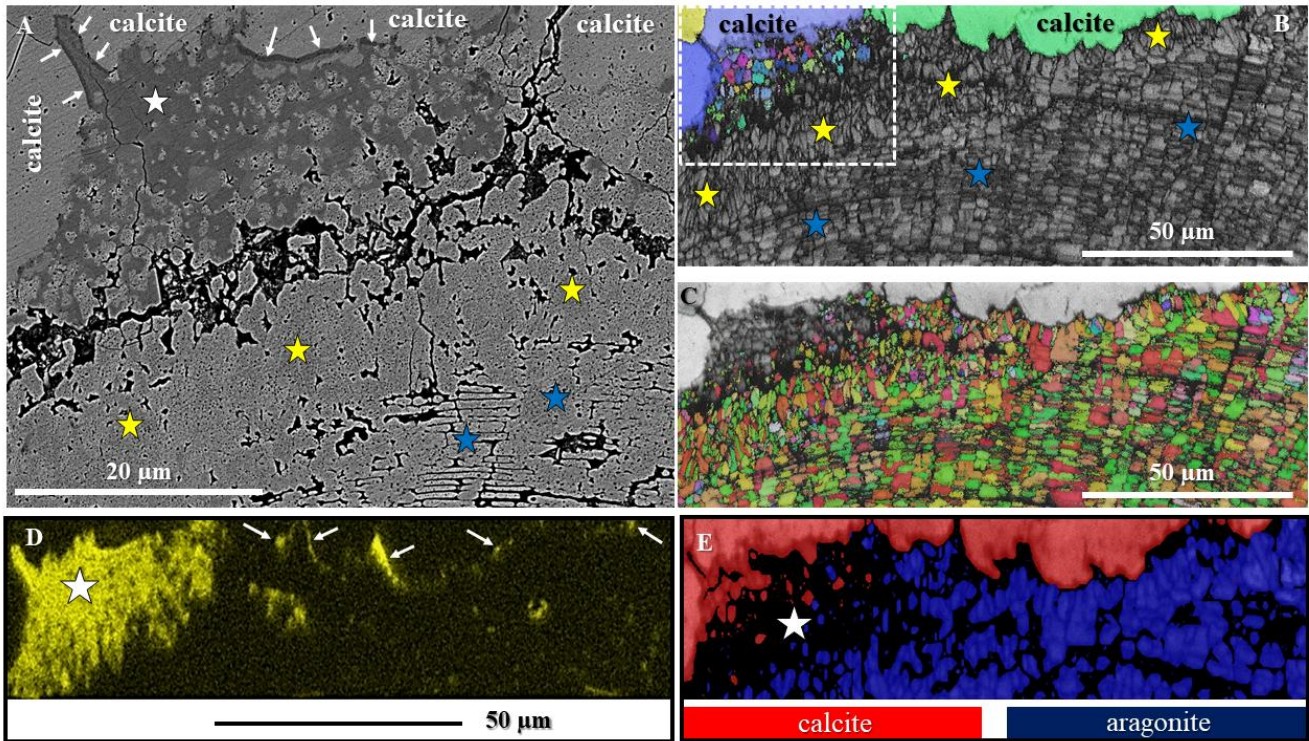

**Figure 9: Microstructural and chemical stages in the replacement process of biogenic nacreous aragonite by inorganic calcite.** *Haliotis ovina* shell material was subjected to hydrothermal alteration for 35 days at 175 °C in a Mg-enriched hydrothermal fluid.

**(A)** SEM image depicting the replacement front between nacreous aragonite and newly formed calcite. Blue stars in (A): nacre tablets forming columns; some traces of the original microstructure can be still observed. Yellow stars in (A): a formerly nacreous shell layer, but, at this stage of alteration, the nacreous microstructure is completely erased. White arrows, white star in (A): high-Mg intercalation between the newly formed calcite and the overprinted, formerly biogenic, aragonite (yellow stars in (A)). **(E)**: Phase map derived from EBSD showing the newly formed calcite (red) and biogenic aragonite (blue). Note that even though the

tablet microstructure cannot be discerned any more, the original mineralogical phase (aragonite) is still preserved. The white star in (E) marks the region where high-Mg calcite intercalation is located, which, in the presence of a Mg-rich fluid, is always present at the replacement front between inorganic calcite and biogenic aragonite. **(D)**: EDX map showing the enrichment in Mg at the transition front in yellow. **(B, C)**: EBSD band contrast shown in grey, and colour-coded orientation maps, respectively depicting traces of columnar aragonite (blue stars in (B)) and overprinted aragonite (yellow stars in (B)). Coloured in (B): newly formed

calcite. Shell layer which is marked with a white dashed rectangle in (B) is shown enlarged in (A). **(C)**: Colour-coded EBSD map of aragonite; in light grey: newly formed calcite, in dark grey: rim containing high-Mg calcite.



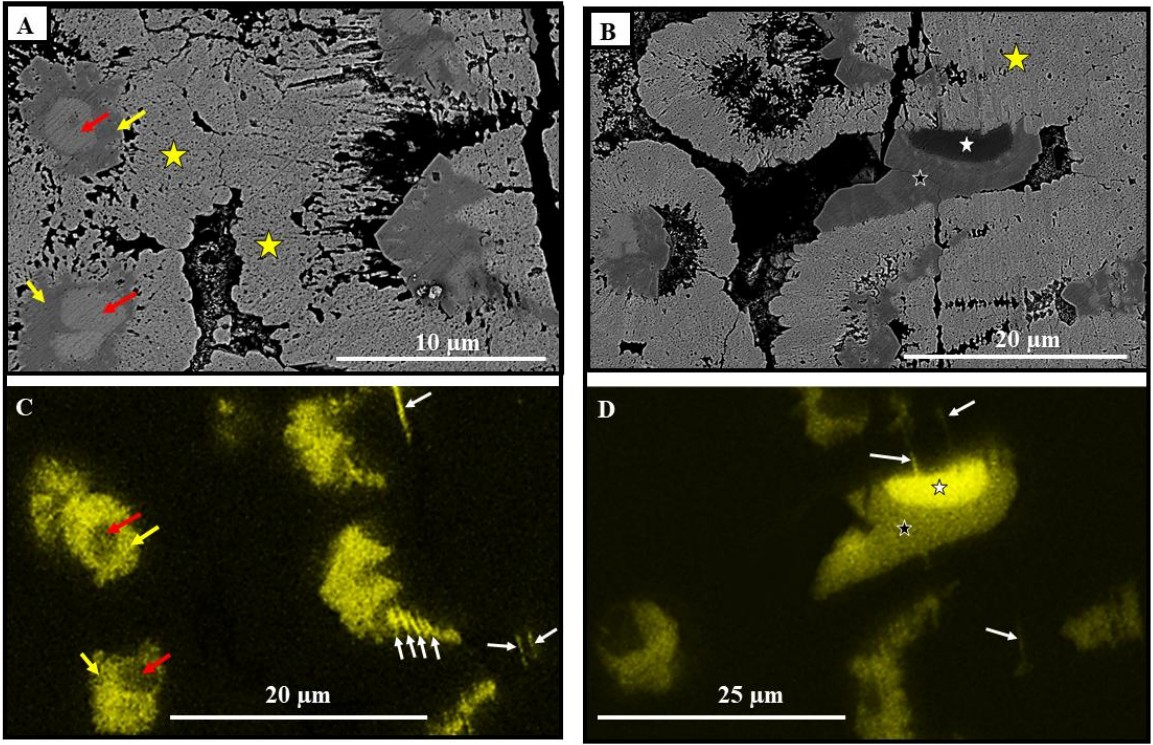

**Figure 10: Replacement of biogenic aragonitic nacre by inorganic calcite. (A): SEM image showing the nucleation and growth of inorganic calcite within shell aragonite. Note the residual aragonite (red arrows in (A and C)) replacing calcite crystals (yellow arrows in (A and C)). Yellow stars in (A): faint traces of nacre columns. (B) SEM image depicting the formation of high-Mg calcite crystals (white stars) within the overprinted, originally nacreous shell layer (yellow star in (B)). Tablet assemblages of columns (yellow star in (B)) are still perceivable. (C, D) EDX map showing (yellow) the presence of calcite crystals with high-Mg content (white and yellow arrows in (C), stars and white arrows in (D)). Red arrows in (C): traces of occluded aragonite distinguished by a low Mg content. White arrows in (C, D): high-Mg streaks that form in cavities between nacreous tablets subsequent to the decomposition of biopolymer membranes around the tablets.**





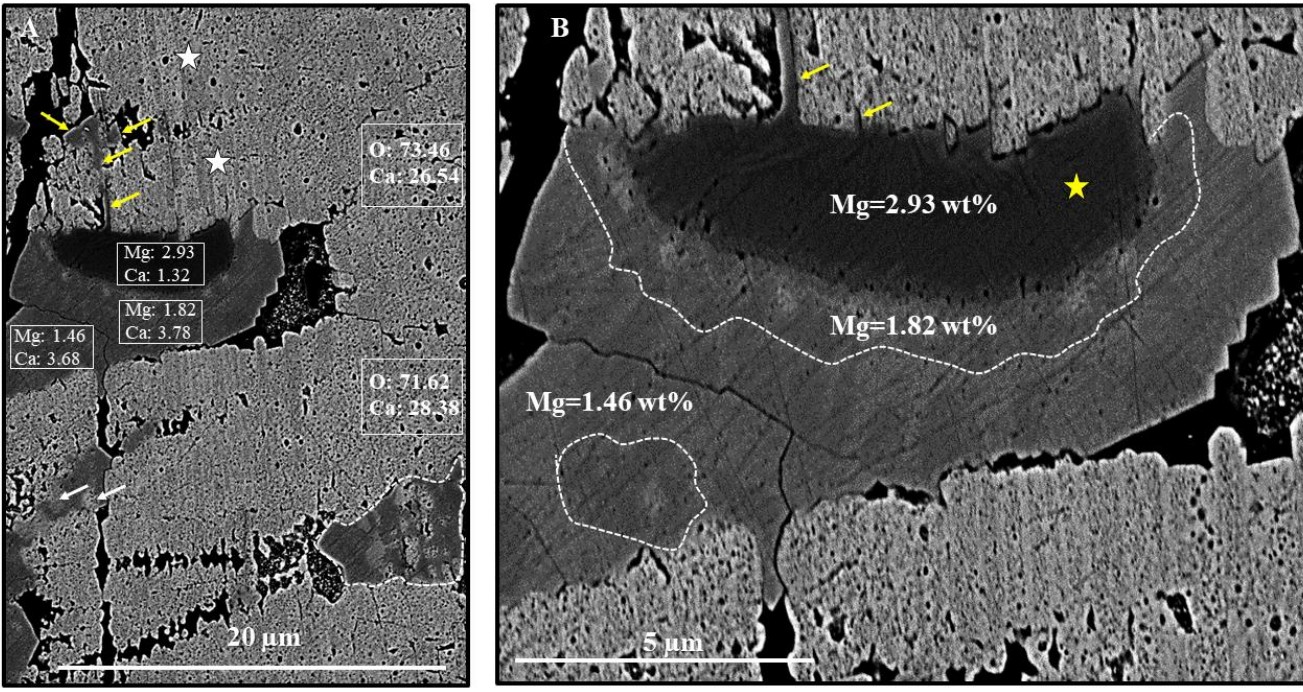

**Figure 11: Enlargement of image shown in Figures 10B and 10D. Mg, Ca, O concentration variation in newly formed calcite and overprinted, formerly biogenic aragonite. The columnar assembly of tablets around the calcite is still perceivable (white stars in (A), (B)). Yellow arrows in (A, B) point to the deposition of high-Mg calcite which fills voids and cavities between former nacreous tablets.**





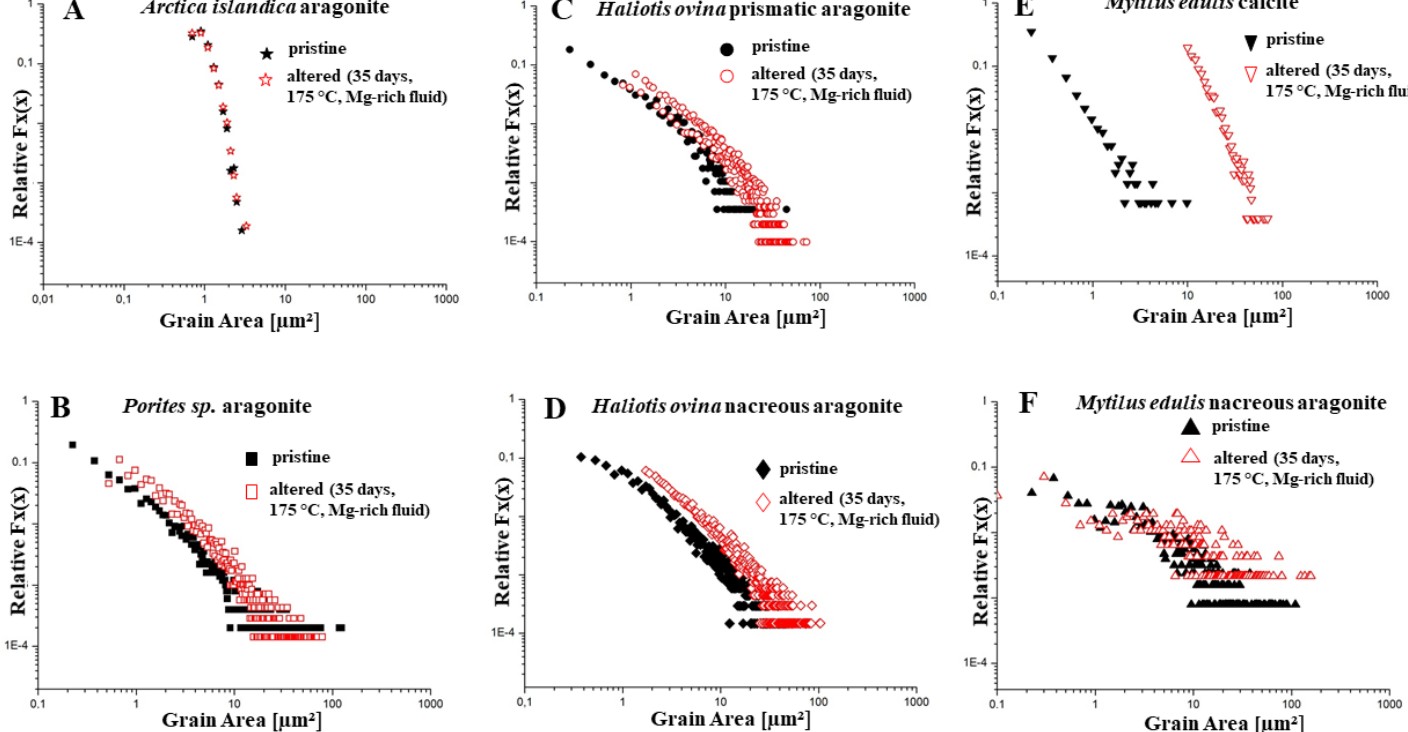

**Figure 12: Relative frequency vs. grain area diagrams for pristine (black) and most altered (red: 35 days, 175 °C, Mg-rich fluid) stages. (A):** *Arctica islandica***, (B):** *Porites sp.***, (C, D):** *Haliotis ovina***, and (E, F):** *Mytilus edulis***. Mineral grain area increases with progressive hydrothermal alteration. The least difference in mineral grain area between pristine and most altered stages is present for the microstructure which forms the shell of *A. islandica* (A), while the most significant difference was observed for *M. edulis* (E) calcite. For all other investigated microstructures (B, C, D, F) mineral grain area increases with alteration, prior to inorganic calcite formation.**



## Appendix

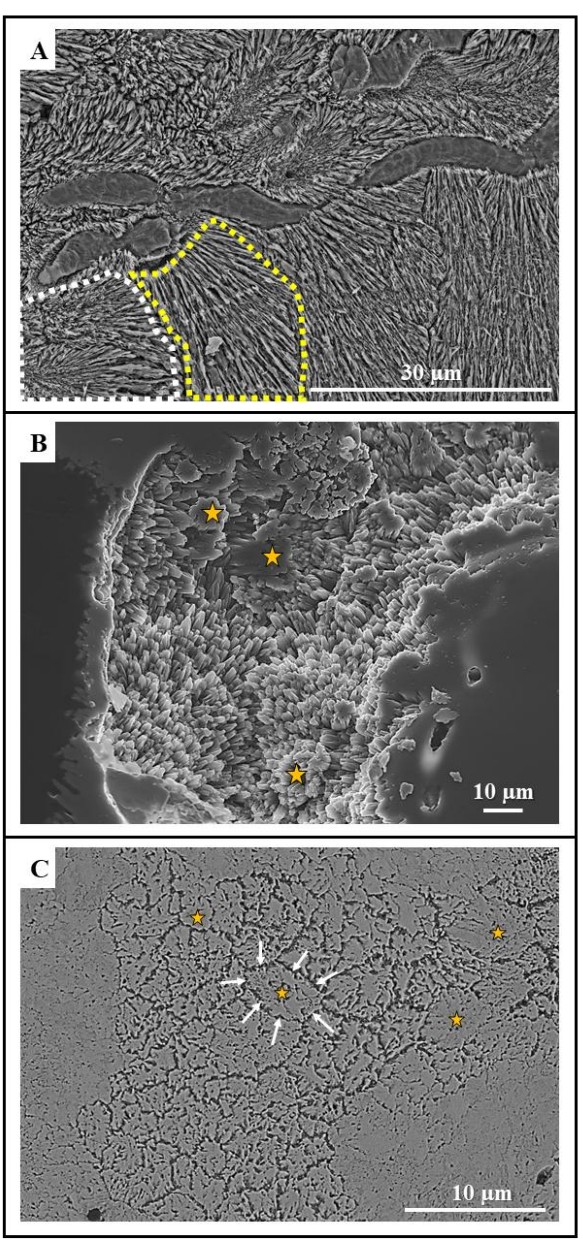

**Figure A1: SEM micrographs showing the acicular microstructure of the modern scleractinian coral *Porites sp.* (A): differently sized and oriented spherulites constitute the skeleton of *Porites sp.*. When fractured in 2D, differently oriented individual basic mineral units consisting of diverging aragonite needles emerge (encircled with white and yellow dashed lines in A1A). (B) Fracture surface image with top view onto bundles of aragonite needles. When needles show some co-alignment, subunits with a closer packing of aragonite needles develop (yellow stars in A1B). On flat, 2D surfaces (A1C) these co-aligned needles form irregularly shaped units with roundish morphology (yellow stars in A1C), that are separated from each other by a multitude of cavities (e.g., white arrows in A1C).**



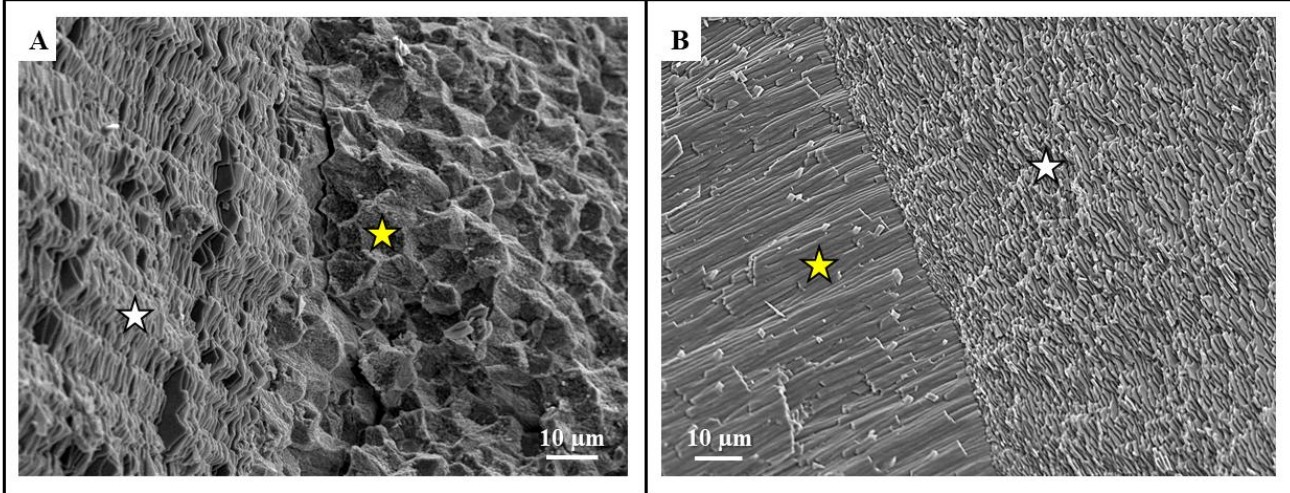

**Figure A2: SEM micrographs of fracture surfaces showing the microstructures of modern shells of (A) gastropod *Haliotis ovina*, and (B) bivalve *Mytilus edulis*. White stars in (A) and (B) indicate the columnar and brick-and-mortar nacre in *H. ovina* and *M. edulis*, respectively, whereas yellow stars in (A) and (B) point to aragonitic prisms in *H. ovina* and calcite fibres in *M. edulis*, respectively.**







**Figure A3: Schematic time line illustrating hydrothermal alteration times. (A)** *Arctica islandica*, **(B)** *Haliotis ovina*, **(C)** *Mytilus edulis*, **and (D)** *Porites sp.*. **Alteration time varied between one and 35 days.**



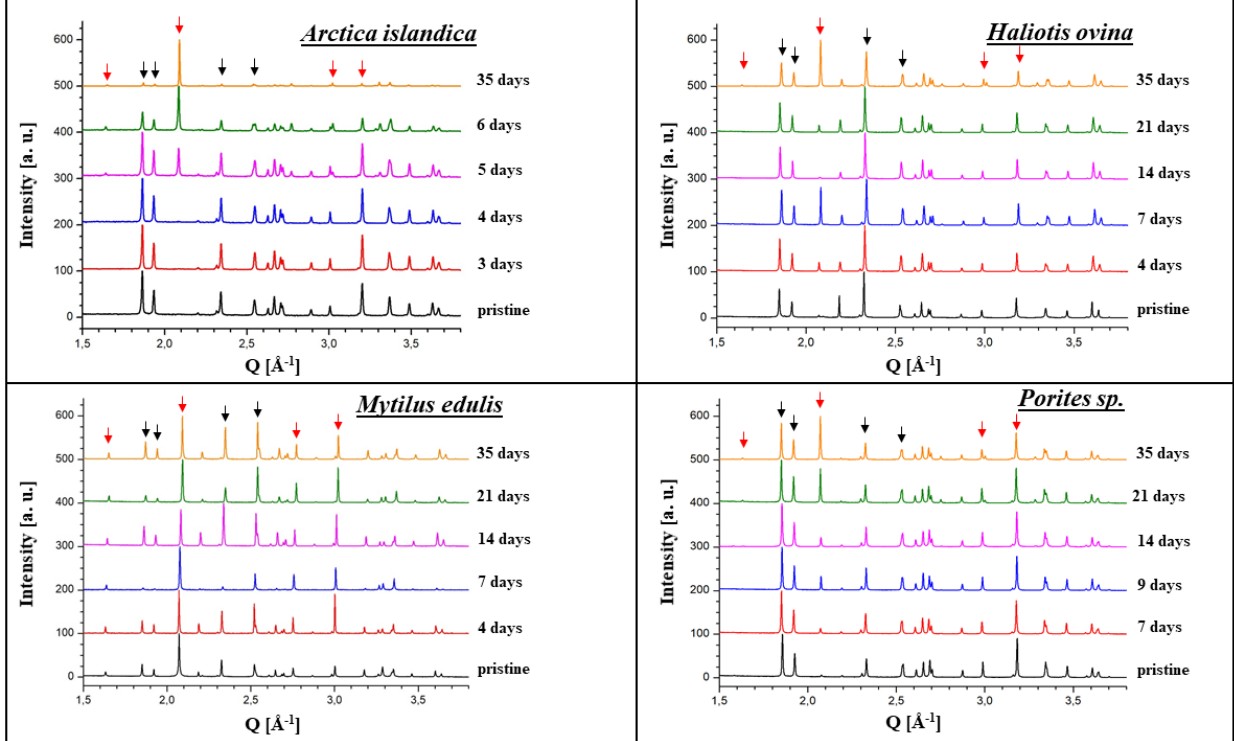

**Figure A4: Selected X-ray diffractograms for pristine and hydrothermally altered (A)** *Arctica islandica***, (B)** *Haliotis ovina***, (C)** *Mytilus edulis***, and (D)** *Porites sp.* **specimens (red arrows: calcite, black arrows: aragonite). Alteration was performed at 175 °C in a Mg-rich fluid simulating burial alteration (100 mM NaCl + 10 mM MgCl$_2$ aqueous solution) and was carried out in a time range between one and 35 days.**




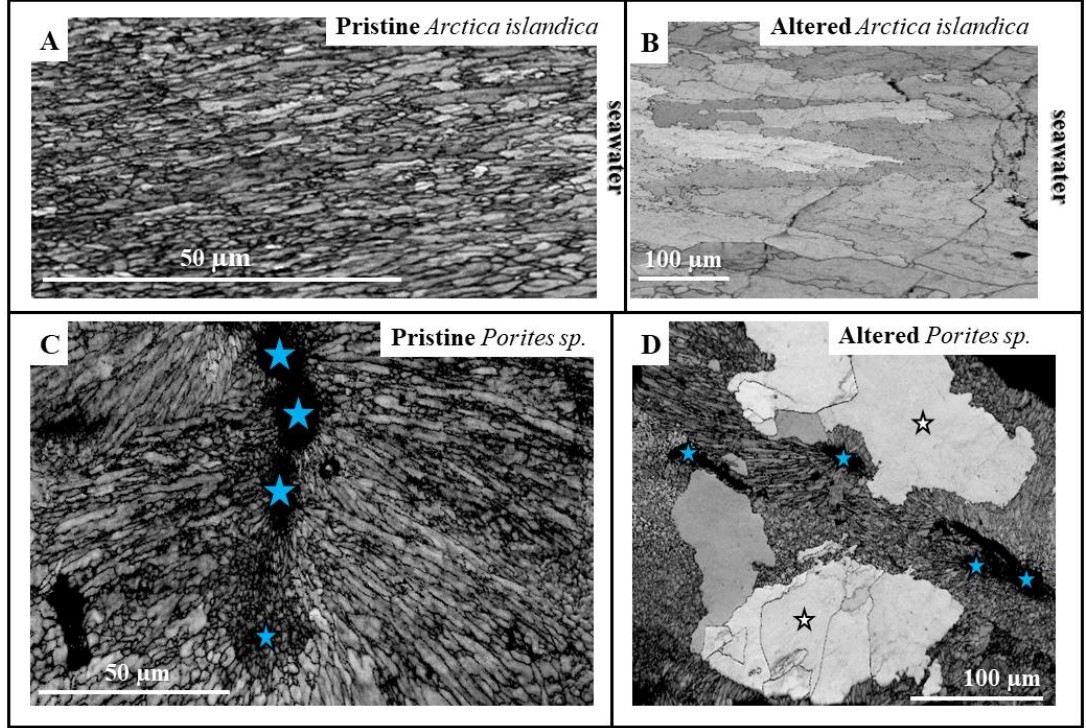

**Figure A5: EBSD band contrast measurements illustrating the difference in microstructure between the pristine and the most altered (A, B): *Arctica islandica* shells, and (B, C): *Porites sp.* skeletons. Hydrothermal alteration lasted for 35 days and was carried out at 175 °C in a fluid simulating burial diagenesis (100 mM NaCl + 10 mM MgCl₂ aqueous solution). (A) The microstructure of the inner shell layer of pristine *A. islandica* consists of small round to elongated aragonitic basic mineral units. (B) Hydrothermal alteration for 35 days induces the replacement of biogenic aragonite by inorganic calcite comprising large calcite crystals. (C) Aragonite needles growing outward from centres of calcification (blue stars (C)) are distinctive features of the microstructure of pristine *Porites sp.* skeletons. When altered for 35 days large calcite crystals (white stars in (D)) develop and grow outward from the centres of calcification (blue stars in (D)). Even after 35 days of alteration, relicts of the original biogenic, acicular coral microstructure are still present and surround the newly formed calcite (D).**



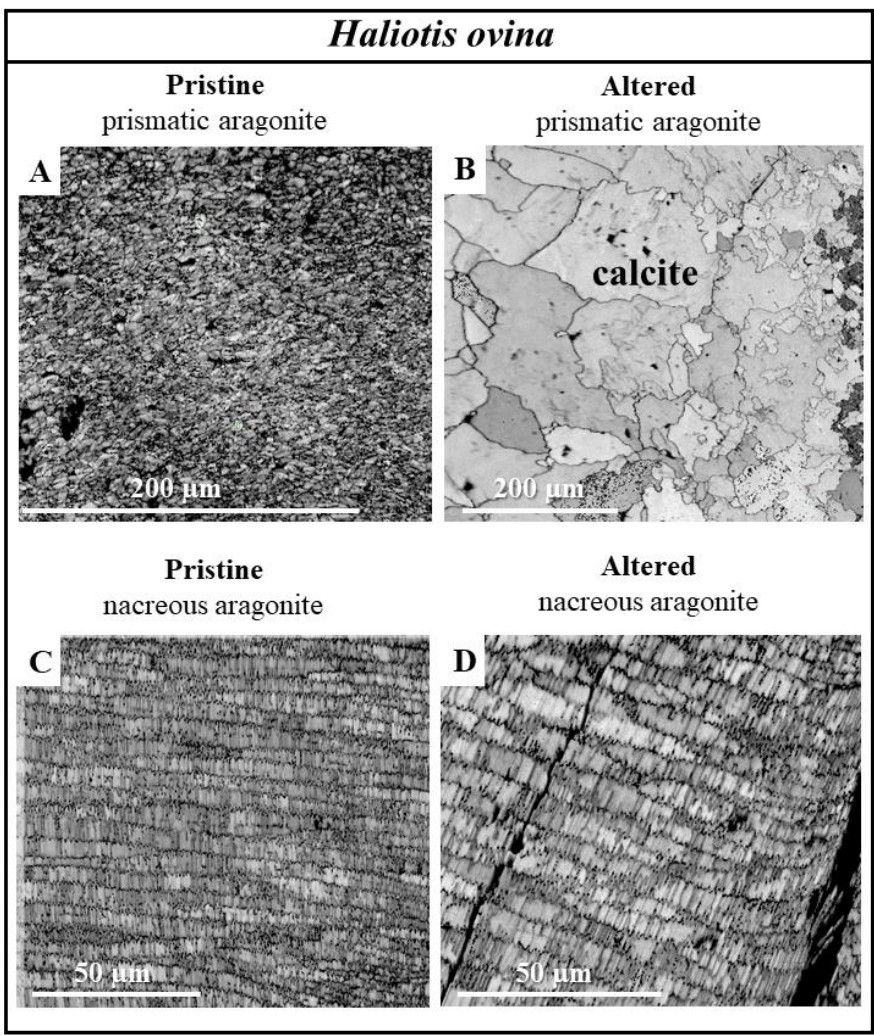

**Figure A6: EBSD band contrast measurements illustrating the difference in microstructure between pristine and hydrothermally altered shells of the gastropod *Haliotis ovina*. Alteration occurred at 175 °C in Mg-rich fluid (100 mM NaCl + 10 mM MgCl$_2$ aqueous solution) and lasted for 35 days. (A) Prismatic aragonite comprising the pristine outer shell layer. (B) After 35 days of alteration calcite crystals, which increase in size towards the centre of the hydrothermally altered shell, form. (C) Columnar nacre in the pristine shell, and (D) in the hydrothermally altered specimen. Nacre is highly persistent through the alteration conditions applied in our experiments. The original microstructural features are well retained, even after 35 days of alteration.**




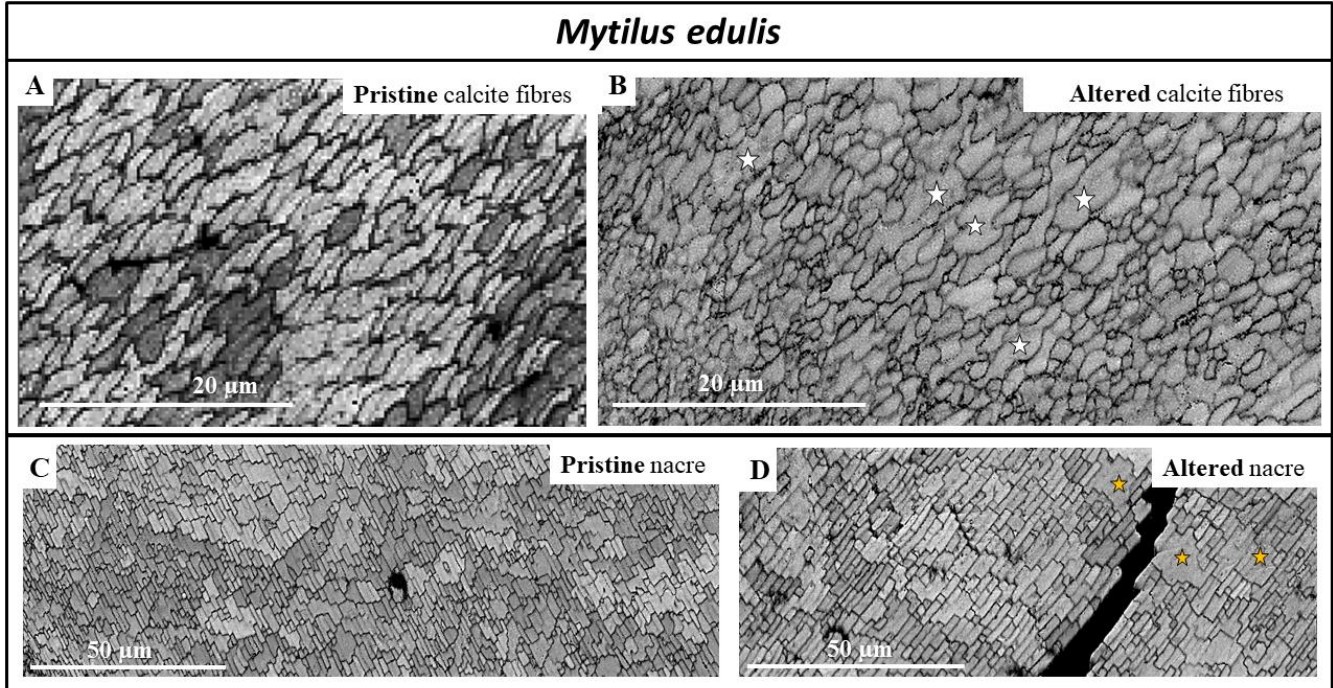

**Figure A7: EBSD band contrast measurements illustrating the microstructures in pristine and hydrothermally altered *Mytilus edulis* shell. Alteration lasted for 35 days at 175 °C and was carried out in a fluid simulating burial diagenesis (100 mM NaCl + 10 mM MgCl$_2$ aqueous solution). (A, B) Pristine and overprinted calcite fibres. (C, D) Pristine and overprinted nacre tablets. The nacre is assembled in a brick-and-mortar arrangement. Even though the nacreous microstructure is very little affected by alteration, some amalgamation of nacre tablets (yellow stars in (D)) is perceivable in the altered sample.**



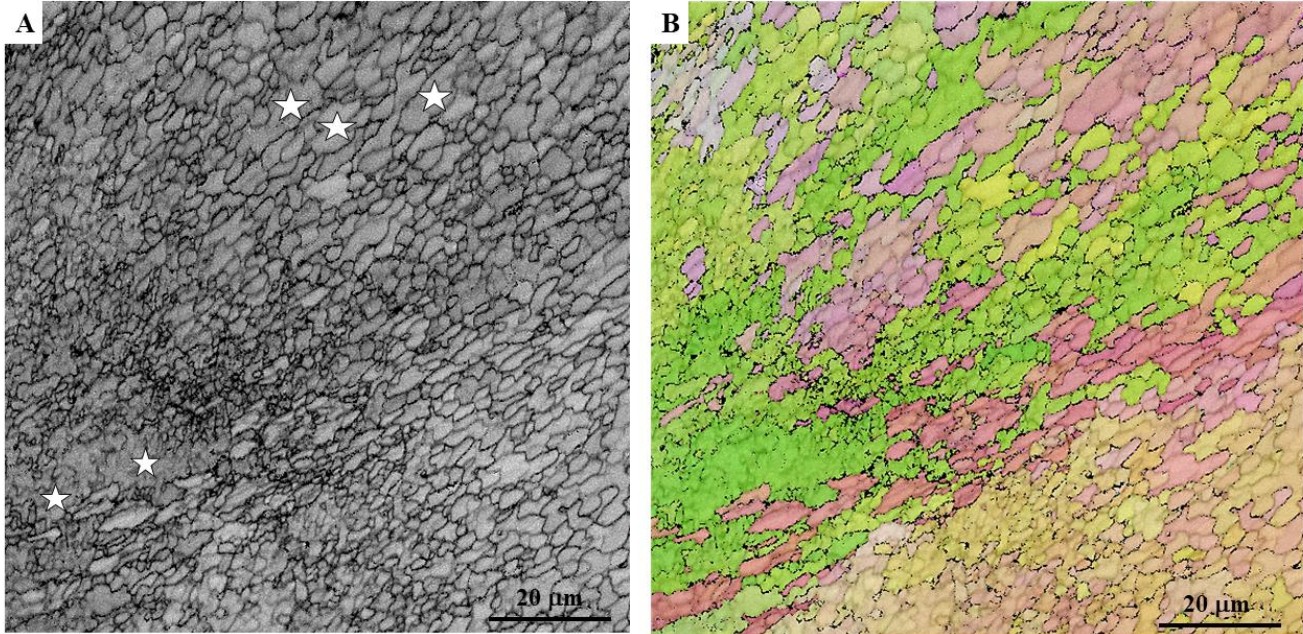

**Figure A8: EBSD band contrast (A) and colour-coded orientation maps of hydrothermally altered (35 day at 175 °C in the presence of Mg-rich burial water)** *Mytilus edulis* **calcite fibres. Significant distortion of fibre morphology and amalgamation into irregularly shaped and sized units can be observed (white stars in A8A).**





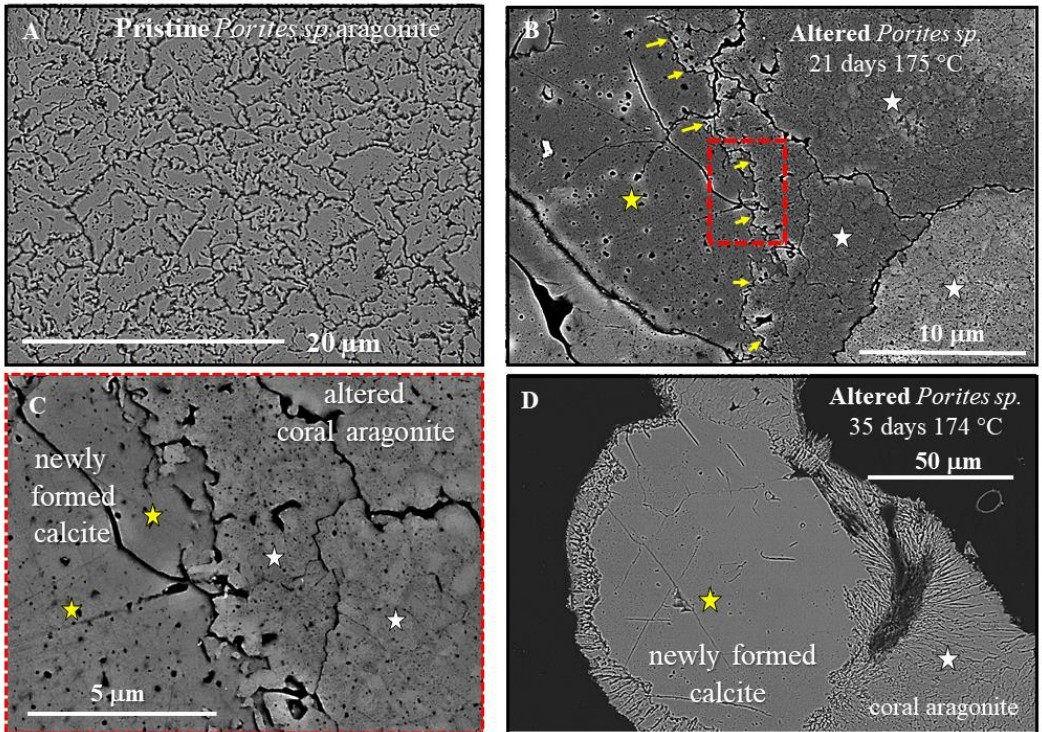

**Figure A9: Basic mineral units in the pristine and altered coral skeleton. (A): SEM image of pristine *Porites sp.* showing irregularly shaped, roundish aragonite entities separated from each other by cavities. (B, C): SEM images of altered *Porites sp.*; white stars: overprinted aragonite; yellow stars: overprinted aragonite now replaced by calcite. Yellow arrows in (B) point to the aragonite-calcite border. Red dashed rectangle in (B) indicates the skeletal region which is shown with a zoom-in in (B, C). Note the amalgamation of basic mineral units in the overprinted, but still aragonitic skeleton. (D): *Porites sp.* skeleton altered for 35 days. Large calcite crystal (yellow star in (D)) extending towards the rim of the skeleton framed by coral aragonite (white star in (D)).**







**Figure A10: SEM images of pristine and altered *Haliotis ovina* prismatic (A, B) and nacreous (C, D) aragonite. In comparison to the pristine microstructures, amalgamation of basic mineral units is one of the major characteristics of both microstructures in the altered shell. (B): New calcite formation (yellow stars in (B)) is significant in the prismatic shell layer, while it is absent in the nacreous shell layer (D). Note the compactness of the nacreous microstructure due to tablet amalgamation in (D).**





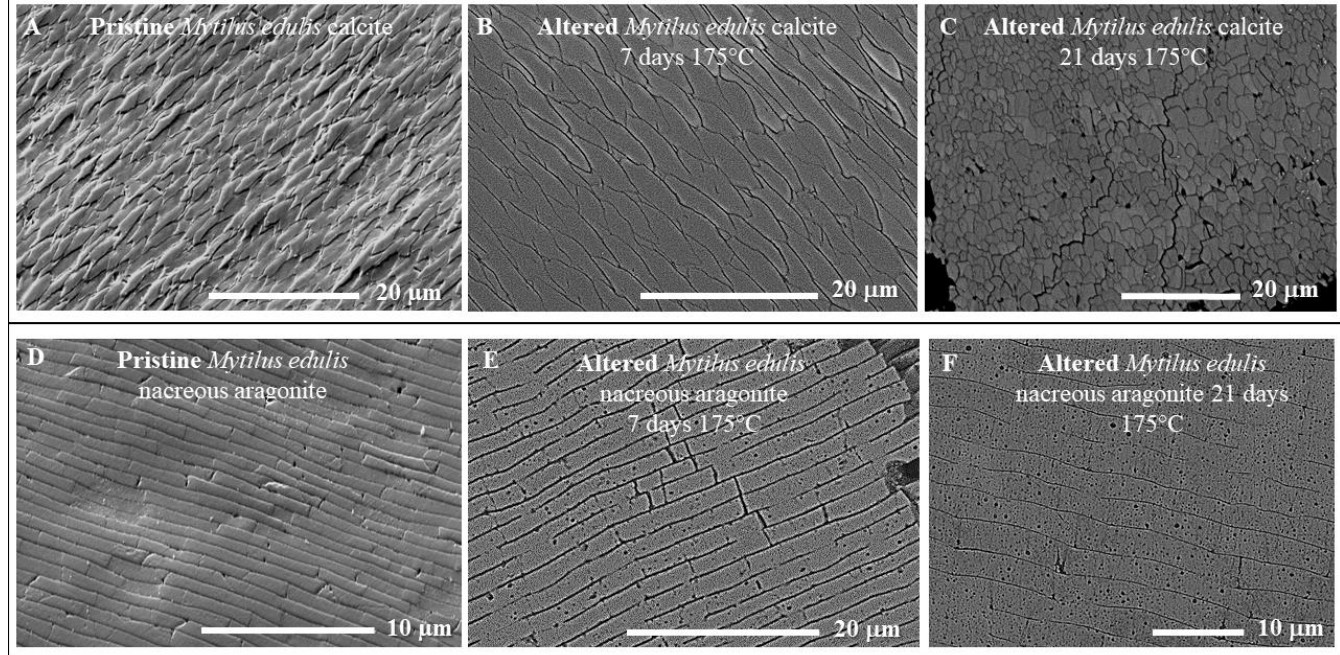

**Figure A11: SEM images depicting microstructural characteristics of pristine and altered *Mytilus edulis* shell calcite and aragonite. (A, D): cross-sections of pristine calcite fibres (A), and nacre tablets (D). (B, C): altered calcite fibres with the clear distortion of fibre morphology (C) after 21 days of alteration. (E, F): Nacre tablets altered for 7 and 21 days. After 7 days of alteration the development of porosity is already evident within nacre tablets (E). This porosity increases significantly with progressive alteration (F) in addition to fibre amalgamation.**



**Figure A12: Retention of some characteristic features of the original biogenic microstructure with progressive alteration. In pristine *Haliotis ovina* there is a gradation in basic mineral unit size, such that large basic mineral units are in the central part of the shell next to the nacre (white stars in (B)). These decrease in size towards the outer rim of the skeleton. (A): EBSD band contrast and colour-coded orientation image showing newly formed calcite in colour and nacreous aragonite in grey. (B): EBSD band contrast and colour-coded orientation image showing newly formed calcite in grey, nacreous and prismatic aragonite in colour.**





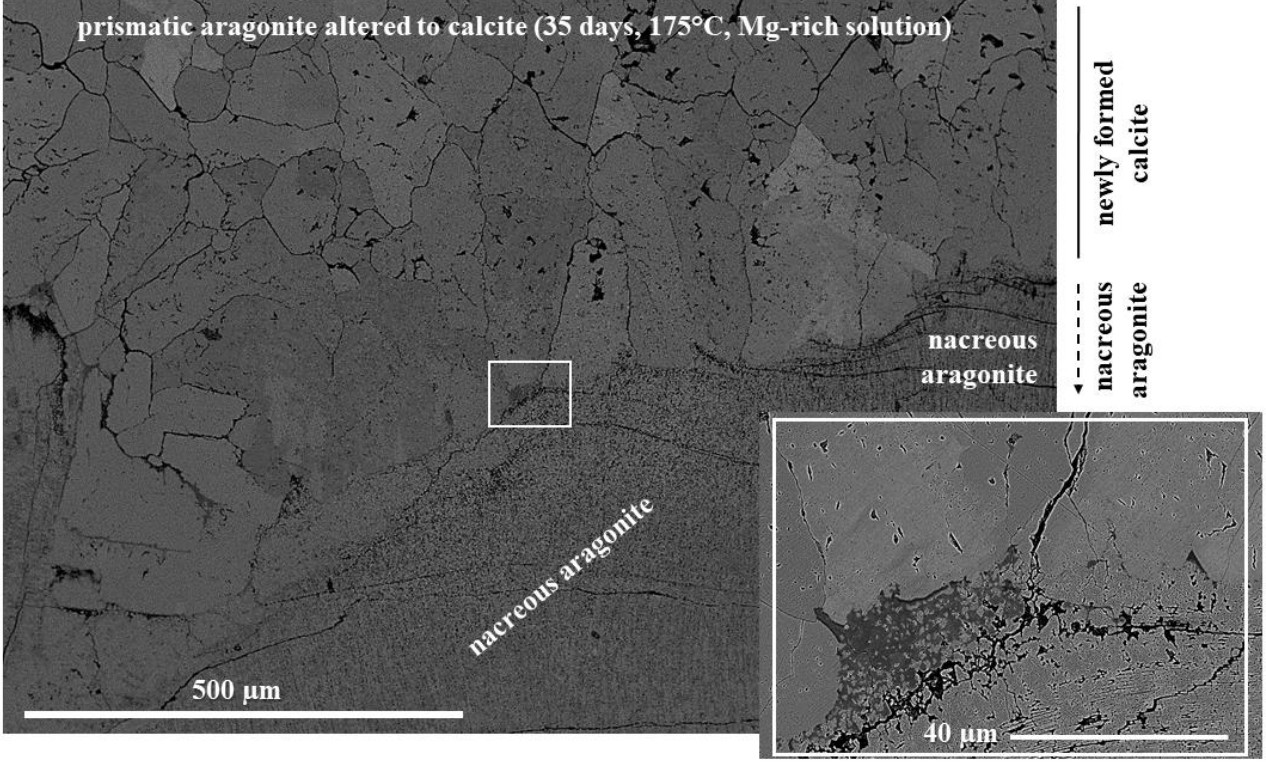

**Figure A13: SEM image showing an overall view of a cross-section through the shell of *Haliotis ovina* which was altered for 35 days at 175 °C in Mg-rich solution. The white rectangle indicates the shell area where the insert of Fig. 9 and Fig. A13 zooms into.**



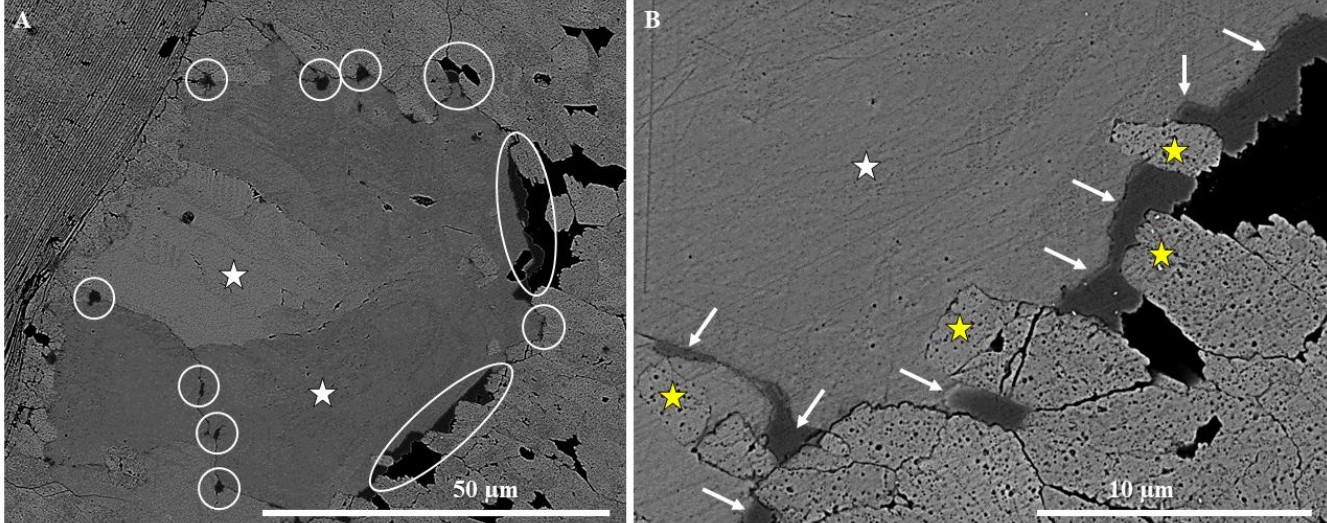

**Figure A14:** Shell segment of *Haliotis ovina* altered for 14 days at 175 °C in the presence of a Mg-rich solution. Large newly formed calcite units grow from prismatic aragonite and are present within the shell next to the nacre (white stars in (A) and (B)). These are seamed by patches of a high-Mg carbonate phase (encircled in (A), indicated with white arrows in (B)), mainly located between the newly formed calcite and the overprinted prismatic aragonite. The newly formed calcite is framed by altered, not yet by calcite replaced prismatic aragonite (yellow stars in (B)).



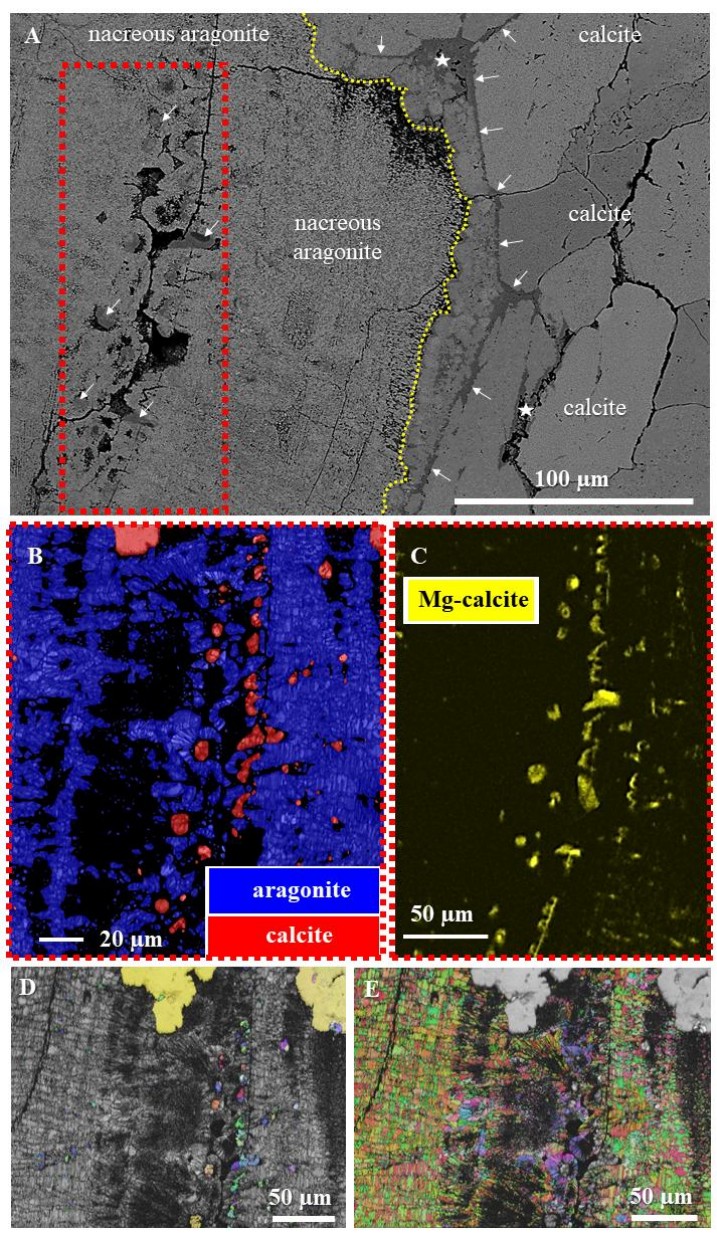

**Figure A15: Contact between newly formed calcite and overprinted nacreous aragonite in hydrothermally altered *Haliotis ovina*.** Alteration occurred for 14 days at 175 °C in the presence of a Mg-rich solution. (A): SEM image showing an overview. Accumulations of high-Mg calcite within calcite can be observed (white stars and white arrows in (A) at the alteration front to nacreous aragonite. Red dashed rectangle in (A) indicates the shell areas shown in (B) and (C). (B): Carbonate phase determination derived from EBSD. (C): Distribution pattern of high-Mg calcite determined with EDX. (D, E): EBSD band contrast (grey scale) and orientation maps (in colour). (D): Band contrast map giving an overview in grey of the aragonitic microstructure which is overlain in colour by the distribution pattern of calcite. (E): Aragonite distribution and mode of orientation in colour overlain by the distribution of calcite in grey.



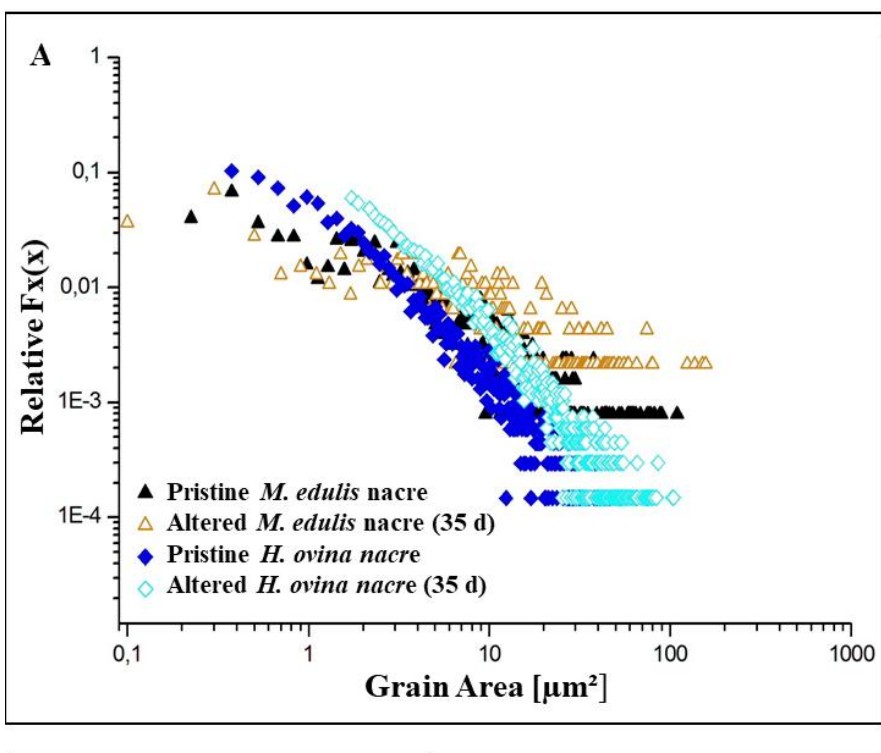

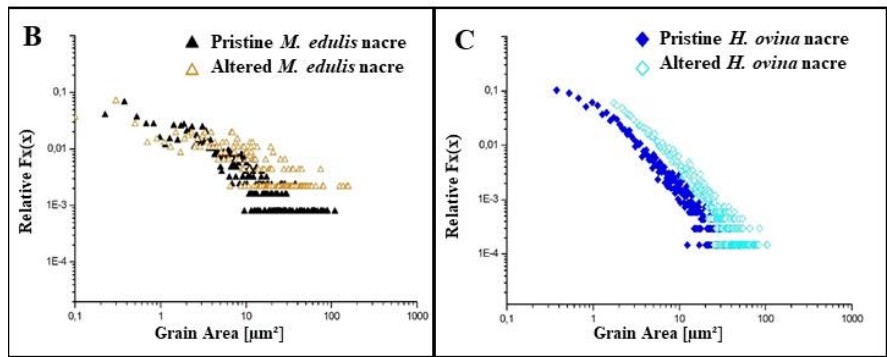

**Figure A16: Relative frequency vs. mineral grain area diagrams for pristine and most altered *Mytilus edulis* and *Haliotis ovina* nacreous aragonite, respectively (A, B, C). For both nacreous structures basic mineral unit growth (tablet amalgamation) can be observed after alteration. This feature is most pronounced for tablet dimensions and mode of assembly in *Haliotis ovina* nacre. Alteration occurred for 35 day in the presence of Mg-enriched fluid at 175 °C.**





**Figure A17: SEM images showing the distortion of the nacreous microstructure prior to phase replacement. (A, B):** *Haliotis ovina* **nacre, (C, D):** *Mytilus edulis* **nacre. White arrows in (A) point to high-Mg calcite 'spots' at the replacement front between newly formed calcite and overprinted shell aragonite. White stars in (A) point to areas at the phase replacement front where traces of the original microstructure (tablets, columns) can be still observed. (B): Overprinted aragonite in three different microstructures: amalgamated nacre tablets (yellow star), over-worked, formerly tabular aragonite (blue stars), amalgamated aragonite prisms (white stars). (C, D):** *Mytilus edulis***: strong tablet amalgamation (white stars in (D)), tablet distortion (white arrows in (C)) and compaction of the nacreous microstructure.**