# Peer review of "Hydrothermal alteration of aragonitic biocarbonates: assessment of micro- and nanostructural dissolution-reprecipitation and constraints of diagenetic overprint from quantitative statistical grain-area analysis"

_Biogeosciences, 2018_

## Referee Comment (RC1) · Anonymous Referee #1 · 25 Jul 2018

The manuscript "Assessment of hydrothermal alteration on micro- and nanostructures of biocarbonates: quantitative statistical grain-area analysis of diagenetic overprint" by Casella et al. represents a substantial contribution to scientific progress in the field of biomineralization and addresses a very important scientific question, the alteration of biogenic hard tissues, which is within the scope of Biogeosciences. The applied methods are valid and clearly outlined and the interpretation and conclusions are strongly supported by the results. The references are appropriate. The conclusions are fundamental as the authors prove the different steps which ultimately lead to calcite replacement of biogenic carbonates, the possible occurrence of overprinted aragonite and importance of grain size, intergrain surfaces and porosity in controling timing and extent of alteration.

However, the overall presentation is not very clear and the language is not always fluent and precise, so I think that the manuscript would benefit of moderate revisions, as discussed below.

General comments

In the Introduction, the authors should describe in more details the mineralogy of selected material (i.e. anticipate what it is written at p. 5).

In paragraph 3.1, the authors should describe in greater details the microstructural characteristics of modern bivalve, gastropod and coral skeletons, which at the moment is only briefly addressed. For instance, A. islandica is known to have an outer homogenous/crossed lamellar/crossed acicular layer, an inner fine complex crossed lamellar layer and an irregular simple prismatic pallial myostracum. The brief description reported in 3.1 does not adequately inform the reader about the fabric and does not correspond to what subsequently written at p. 6 line 30 (aragonite prisms, but the microstructure of A. islandica is not prismatic see Dunca et 2009; Schone et al 2013).

I do not think that the microstructure of M. edulis can be described as consisting of calcite fibres. What shown in Fig. A2B are calcite prisms not fibers. Other figures may be more questionale, but the microstructures of M. edulis is foliated and prismatic (see for instance Brom & Szopa 2016; Carter et al. 2013). Eventually it is described as fibrous prismatic (Brom & Szopa 2016), a term which I do not agree with, but which is used (Carter et al. 1990) and it is distinct from the typical fibrous fabric of brachiopods.

An important issue is the time of decay of organic sheaths around the basic mineral units, which is not clearly indicated but just discussed as short.

Some important concepts (porosity) are not described enough clearly.

In paragraph 4.3, the authors should add the stratigraphic age of the described fossil material in order to support their conclusions.

In the conclusions, the authors should report and give more enphasis to the important statement: "even though nacreous aragonite is still preserved as aragonite, it is an overprinted aragonite that, most probably, holds little of the original microstructural or geochemical signature".

Technical corrections

p. 1 line 34: sentence unclear

p. 2 line 9-10: long and complex sentence

p. 2 line 16 (and below in the text): sp. not italics

p. 2 line 25-30: I would describe before all the molluscs and only after the corals or viceversa.

p.3 line 2: correct M s edulis to M edulis test material: it would be better to indicate the dimension for the size (length, width, height?)

p. 4, line 24: the critical misorientation value. Sentence not finished

p.5 line 19: correct H s ovina to H ovina

p. 5 line 20: add the type of fabric for A. islandica.

p. 6 line 5: I do not think that the shell of M. edulis can be described as consisting of fibres, but prisms. Please check carefully also in the literature (Carter et al. 1990).

p. 6 line 13-14: explain better this statement. The examples that follows are not strictly related to it.

p. 6 line 29: How long does it take for organic fibrils to be destroyed? What is the relationship between this processdecay and the "dormant" interval reported at p. 7?

p. 6 line 30 and p. 14: the microstructure of A islandica is not prismatic (Dunca et 2009; Schone et al 2013)

p. 7 line 25: again it is very important for this statement that the microstructure of the two taxa is described in great details, which is not at the moment.

p. 8 line 13: "for both microstructures", is it true also for both mineralogical phases in M edulis?

p. 8 line 22-25: the description of the "rise in porosity" is very important but it is not described enough clearly. It should be stated more clearly that pores are present in the biogenic carbonates.

p. 9 line 12-14. Sentence not clear.

p. 12, lines 3-4 and p. 15, line 3-4. Prismatic and nacre microstructures are among the shell microstructures, the ones having the higher amount of organic content, more than the homogeneous/fine complex crossed lamellar fabric in A. islandica. Having a high organic content they should have also a high primary porosity. Or is it a matter of pore size?

p. 12, line 11 Regenberg et al. 2007, comma missing after et al.

p. 12, line 29-35. This part is not very clear and not very well fitted into the paragraph. Also should not it be placed in the results?

p. 14 line 5-13. Very important process, to be described more clearly. It is nor clear why "Carbonate phase alteration kinetics in A. islandica shell is sluggish at first" and why porosity "explains the little difference in mineral grain area".

p. 14 line 22-23. "the increased prevalence of the nacreous shell layer of M. edulis relative to calcitic shell layers in seashore sediments". This statement should be better explained and supported.

p. 15 line 5-6: sentence not clear

p. 15, "It has been further demonstrated that in Palaeozoic marine faunae taxa with calcitic skeletons prevail". The authors have to add fossil before marine fauna

p. 16 line 26: tissue forms or tissues form

p. 17 line 10-13: sentence long and not clear

p. 17 line 26: "Thus, in the case of aragonitic tissue the survival of biogenic aragonite" better to correct into "Thus, in the case of aragonitic tissue the survival of biogenic aragonite"?

References: Crippa & Raineri (2015) is in the text but it missing from the ref list

––––––––––––––––––––––––––––––––

---

## Referee Comment (RC2) · Anonymous Referee #2 · 26 Jul 2018

The authors present an interesting study of the process of alteration of six selected microstructures when submitted to the action of hydrothermal fluids. They assess the process of transformation of either biogenic aragonite or calcite into inorganic calcite with time. Although the simulated diagenetic alteration conditions are probably a small portion of the whole range existing in ambient conditions, the study is highly meritorious. Particularly, the application of EBSD analysis provides a wealth of information. Although I am not expert on diagenesis, I would be surprised if such a picture of the progression of a simulated diagenetic alteration ever existed.

[Figure]

The study is significant in its field, and the conclusions are backed by data. It is technically very sound. The statistics is consistent. The Ms is well written and profusely illustrated. The selected microstructures are representative of the range present in molluscs.

I am only concerned about the dissonance between the references in text and in the reference list. Many in-text references are not found at the end and the other way round (see at the end). This has to be amended before publication.

Minor comments: Abstract.- I do not know if the journal allows for the presence of references within the abstract. Please check. Page 2, line 8.- "Despite ongoing and extensive research", do you mean 'previous' extensive research. Page 3, line 2.- delete "s" after "M." Page 3, line 8.- 'were carried out' should not be italicized; "tissue" should be plural. Page 5, line 13.- Correct spelling is Carvajal. Page 5, lines 25-26.- "consists", "consisting"; avoid redundancy. Page 5, line 28.- "When sectioned in 2D", section, by definition, provides a 2D view; 'in 2D' can be deleted. Page 7, line 7.- "sp.", here and elsewhere, should not be italicized. Page 7. line 31.- "compare Fig. 4A with right hand part, framed in green with Fig. 4D", I would suggest 'compare Fig. 4A with Fig. 4D, right hand part, framed in green'. Page 9, line 8.- Comma after "shows". Page 10, line 4.- Comma after "sediments". Page 10, line 20.- Correct spelling is 'Etschmann'. Jonas et al. 2015 appears as 2017 in References. Page 11, lines 13-14.- The sentence contained there is either incomplete or the initial "In" needs to be removed. Page 13, line 21.- Correct spelling is "Fernández-Díaz". Page 14, lines 22-23.- "as well as the increased prevalence of the nacreous shell layer of M. edulis relative to calcitic shell layers in seashore sediments." Do the experimental conditions really simulate the diagenetic conditions in nearshore sediments? If so, please provide additional data and/or references. Page 15, line 3.- Replace "the nacreous tablets are" by 'nacre is'; the microstructure is nacre. Page 15, line 20.- Delete "of". Page 15, line 31.- "Biogenic aragonite was dissolved for the reprecipitation of low-Mg calcite", this is odd; please rewrite. Page 16, line 9.- Sorauf (1980) appears as 1981 in References. Page 16, line

28.- Comma after "immediately". Page 17, line 11.- 'replacement' instead of "replaced". Page 18, line 14.- "Their" should be lower case. Page 19, line 33.- "tuberculate" should be 'tuberculata'. Page 21, line 13.- "cabonates", 'carbonates'.

THE following in-text references have not been found in the References list (per order of appearance): - Patterson and Walter, 1994 - Ku et al. 1999 - Brad et al. 2004 - Zazzo et al. 2004 - Immerhauser et al. 2015Âł - Ridgway and Richardson 2011 - Krause-Nehring et al. 2012 - Rodgway et al. 2012 - Crippa and Raineri 2015 - Blanchon et al. 2009 - Hahn et al. 2012 - Nidiyasari et al. 2015 - Brown et al. 1962 - Cardew and Davey 1985 - Regenberg et al. 2007 - Hover et al. 2001 - Cherns et al. 2008 - Wright et al. 2003 - James et al. 2005 - Harper 1998 - Harper 2000 - Kidwell 2005 - Land 1967 THE following references in the References list have not been found in the main text: - Addadi et al. 2006 - Allison et al. 2007 - Altree-Williams et al. 2017 - Barthelat and Spinosa 2007 - Bathurst 1975 - Böhm et al. 2006 - Brahmi et al. 2012 - Brocas et al. 2013 - Butler et al. 2009 - Cartwright and Checa 2007 - Checa et al. 2006 - Checa et al. 2009 - Checa et al. 2011 - Cohen et al. 2001 - Currey et al. 2001 - Dauphin et al. 1989 - Elliot et al. 2003 - Gries et al. 2009 - Grossman et al. 1993 - Heiss 1994 - Hipler et al. 2009 - Hubbard et al. 1990 - Jackson et al. 1988 - Korte et al. 2005 - Levi-Kalisman et al. 2001 - Li et al. 2006 - Marchitto et al. 2000 - Marin and Luquet 2004 - Mayer 2005 - McGregor and Gagan 2002 - Metzler et al. 2007 - Morton 2011 - Nudelman et al. 2006 - Nudelman et al. 2008 - Oeschger and Storey 1993 - Parkinson et al. 2005 - Putnis et al. 2005 - Raffi 1986 - Richardson 2001 - Rüggeberg et al. 2008 - Sanchez et al. 2005 - Schöne et al. 2004 - Schöne et al. 2005a - Schöne et al. 2005b - Schöne and Surge 2012 - Taylor 1976 - Wanamaker et al. 2008 - Wang and Gupta 2011 - Wang et al. 2011 Figure A16.- Panel A is simply the sum of B and C. It can be simplified.

---

## Referee Comment (RC3) · Anonymous Referee #3 · 1 Aug 2018

Through a series of well planned experiments followed by analyses at the micrometer and nanometer scale, the authors have documented the effect of medium-temperature hydrothermal solutions on the calcium carbonate shells of six different species displaying five different types of microstructural units. They have demonstrated the (expected) importance of the amount of time the materials spend in the hydrothermal solutions on the degree of shell alteration. What may be less expected, however, are the documented differences in degree of alteration under the same experimental conditions, but in different species and in different microstructures. The authors focused on observ-

able structural changes, but also investigated some aspects of compositional changes due to interaction between the calcite/aragonite shells and the Mg-Na-Cl solutions at 175 C. Changes in grain size and orientation within selected areas of the shells were monitored through EBSD and FE-SEM. Changes in grain size were found to be the most sensitive indicators of hydrothermal alteration. The paper is replete with excellent photomicrographs and EBSD color-coded images showing grain orientation and phase identification. The results of this extensive work have important implications for our ability to recognize which biocarbonate fossil materials are most pristine and therefore most appropriate for chemical and isotopic analysis that may be used to reconstruct the details of past environments.

In summary, this is a very good paper that provides well-documented, quantitative experimental observations on the species-specific, microstructure-specific aqueous alteration of biocarbonate shells. I believe that this work marks a large step forward in the understanding of the preservation/fossilization process of shells. As the authors point out, their work also sheds light on how differential alteration effects can account for differential preservation of shell material. The implications of this result alone are extremely important for the interpretation of the fossil record. The results of this study also will guide the evaluation of fossil materials for isotopic and geochemical analysis with the goal of environmental reconstruction.

The current document requires some minor re-writing, as well as the addition of materials indicated in the comments below and on the attached annotated copy of the manuscript. I believe this paper will be of strong interest to the readership of this journal.

There are some improvements that could be made in the paper. Immediately below are more general suggestions. Below them are more specific comments, questions, and suggestions keyed to identified lines in the manuscript. A scanned copy of the annotated manuscript is also provided with additional detailed suggestions.

The abstract is not appropriate as written. The first paragraph provides too much background information that instead should be in the introduction. The abstract, however, does not include necessary details about the experimental procedure and the types of analyses performed.

This paper covers a huge amount of information on several morphologically different parts of the shells of six species, undergoing alteration for up to 35 days. The authors have included some figures and tables to guide the reader through this information so that the discussion and conclusions will be convincing. Any additional techniques to help the reader keep track of, organize, and compare these data/observations would further enhance the impact of this work.

As much as I appreciate the level of detail presented on the individual samples and their respective microstructures, I suggest that the authors take a fresh look at the paper to determine if they reasonably can condense it further. At times, the density of information approaches overwhelming.

A distinction that should be addressed in the paper is the difference intended by the authors between the terms "overprinted" and "replaced." This is an important distinction especially for (future?) work on chemical and isotopic signatures that may or may not be transferred to the "new" solids formed after hydrothermal alteration.

The tracking of Mg and its importance in the alteration products are not well explained; nor is the addition of only that one element to the NaCl hydrothermal solution. Given the so-called "Mg-poisoning" of calcite, some more discussion in needed. Does all the Mg that occurs in the secondary Mg-calcites come from that introduced by the hydrothermal solutions, or is there Mg in the pristine aragonite?

Question that is more appropriate for on-line discussion:

What should be the next steps in experimental and analytical studies to evaluate how well texturally altered samples (as shown by changes in grain-area evaluation) of biogenic carbonate hard tissues retain their chemical (incl. trace-element) and isotopic signatures?

Attached here is a scanned copy of the annotated manuscript, which has numerous detailed suggested changes. In the right-hand margins are circled numbers that are linked to comments and questions below.

Point 1, page 2: The information in the first paragraph of the abstract would be better placed in the introduction. One usually avoids including reference citations in an abstract.

Point 2, page 2: The second paragraph in the abstract is a good overview of the results and implications of the work. However, more detail is needed on the experimental procedure and analytical techniques used.

Point 3, page 4, line 111: The term "basic mineral unit" should be defined here, where it is first introduced, or the placement of the definition should be stated here.

Point 4, page 4, lines 115-117: The parts of this statement that are not currently in the abstract would be very useful there.

Point 5, page 6, lines 169-171: It would seem that the authors wish to provide an overview for or give a "heads-up" to the reader regarding the upcoming experiments and their results. However, this sentence is very compact and uses several not-yet-defined terms, making it unsuccessful in this inferred purpose.

Point 6, page 6, lines 175-176: The term "burial fluid" should be explained and its chosen composition justified, especially given the importance of the Mg component, as revealed by the experimental results.

Point 7, page 9, line 232: Although the term "microstructure" was defined earlier, it would be useful to remind the reader how that term is being applied in this paper. It is not a general term, but rather a stand-in for several types of mineralogical-structural morphologies focused on in the paper. Perhaps the parenthetical interjection "(e.g., calcite fibers, prismatic aragonite, nacreous aragonite)" would be useful here.

Point 8, page 10, lines 261-262: It would helpful to the reader to be told how long these intervals of time are that give rise to the changes observed.

Point 9a, page 10, line 278: Does "other shell regions" refer to other parts of the prismatic area or to regions other than in the prismatic area?

Point 9b, page 13, lines 355-356: The excellent question is posed here of what determines the preservation potential of a fossil archive. This question should be re-stated and succinctly answered toward the end of the paper.

Point 10, page 15, lines 408-410: This is an important discussion of a complicated set of experiments. It is difficult for the uninitiated reader to understand what the focal points are. Perhaps some additional details about the original experiments would clarify the discussion.

Point 11, page 16, lines 446-447: I agree with this statement. At some point in the paper, though, it should be acknowledged that the present work does not directly address the issue of the geochemical fidelity of the recognizably/quantifiably altered carbonates.

Point 12, pages 20 and 21, lines 563 and 566: There seems to be a self-contradiction between these two lines with regard to the (lack of) fidelity in the geochemical signature.

Please also note the supplement to this comment:
https://www.biogeosciences-discuss.net/bg-2018-249/bg-2018-249-RC3-supplement.pdf

———————————————————

[Figure]

**Supplement:**

**Review due Aug. 3**

| 1                                                                    | Assessment of hydrothermal alteration on micro- and nanostructures of                                                                                                                                                                                                                                                                                                                                                                                                                                                                                                                                                                                                                                                                                                                                                                                                                                                                                                                                                                                                                                                                              |
|----------------------------------------------------------------------|----------------------------------------------------------------------------------------------------------------------------------------------------------------------------------------------------------------------------------------------------------------------------------------------------------------------------------------------------------------------------------------------------------------------------------------------------------------------------------------------------------------------------------------------------------------------------------------------------------------------------------------------------------------------------------------------------------------------------------------------------------------------------------------------------------------------------------------------------------------------------------------------------------------------------------------------------------------------------------------------------------------------------------------------------------------------------------------------------------------------------------------------------|
| 2                                                                    | biocarbonates: quantitative statistical grain-area analysis of diagenetic overprint                                                                                                                                                                                                                                                                                                                                                                                                                                                                                                                                                                                                                                                                                                                                                                                                                                                                                                                                                                                                                                                                |
| 3
                                                     | Laura A. Casella 1* Sixin He 1 , Erika Griesshaber 1 Lourdes Fernández-Díaz 2 Elizabeth M. Harper 3 Daniel I                                                                                                                                                                                                                                                                                                                                                                                                                                                                                                                                                                                                                                                                                                                                                                                                                                                                                                                                                                                |
| 5                                                                    | Jackson 4 Andreas Ziegler 5 Vasileios Mavromatis 6,7 Martin Dietzel 7 Anton Fisenbauer 8 Uwe Brand 9 and                                                                                                                                                                                                                                                                                                                                                                                                                                                                                                                                                                                                                                                                                                                                                                                                                                                                                                                                                                         |
| 5                                                                    | Wolfgang W Schmahl 1                                                                                                                                                                                                                                                                                                                                                                                                                                                                                                                                                                                                                                                                                                                                                                                                                                                                                                                                                                                                                                                                                                                    |
| 0
                                                     | Wongang W. Seminan                                                                                                                                                                                                                                                                                                                                                                                                                                                                                                                                                                                                                                                                                                                                                                                                                                                                                                                                                                                                                                                                                                                                 |
| 8
|  <li>1Department of Earth and Environmental Sciences and GeoBioCenter, Ludwig-Maximilians-Universität München, Munich, 80333, Germany</li> <li>2Instituto de Geociencias, Universidad Complutense Madrid (UCM, CSIC), Madrid, 28040, Spain</li> <li>3Department of Earth Sciences, University of Cambridge, Cambridge, CB2 3EQ, U. K.</li> <li>4Department of Geobiology, Georg-August University of Göttingen, Göttingen, 37077, Germany</li> <li>5Central Facility for Electron Microscopy, University of Ulm, Ulm, 89081, Germany</li> <li>6Géosciences Environnement Toulouse (GET), CNRS, UMR 5563, Observatoire Midi-Pyrénées, 14 Av.</li> <li>E. Belin, 31400 Toulouse, France</li> <li>7Institute of Applied Geosciences, Graz University of Technology, Rechbauerstr. 12, 8010 Graz, Austria</li> <li>8GEOMAR-Helmholtz Centre for Ocean Research, Marine Biogeochemistry/Marine Geosystems, Kiel, Germany</li> <li>9Department of Earth Sciences, Brock University, 1812 Sir Isaac Brock Way, St. Catharines, Ontario,</li>  |
| 20
                                             | L2S 3A1, Canada                                                                                                                                                                                                                                                                                                                                                                                                                                                                                                                                                                                                                                                                                                                                                                                                                                                                                                                                                                                                                                                                                                                                    |
| 23
| *Corresponding author: Laura Antonella Casella
Ludwig-Maximilians-Universität München
Department of Earth and Environmental Sciences
Theresienstr. 41
80333 Munich, Germany
Tel.: +49 89 2180-4354
eMail: Laura.Casella@lrz.uni-muenchen.de                                                                                                                                                                                                                                                                                                                                                                                                                                                                                                                                                                                                                                                                                                                                                                                                                                                                                      |
| 35                                                                   | Key words: Biogenic calcite, biogenic aragonite, coupled dissolution-reprecipitation, primary and secondary                                                                                                                                                                                                                                                                                                                                                                                                                                                                                                                                                                                                                                                                                                                                                                                                                                                                                                                                                                                                                                        |
| 36                                                                   | porosity, mineral replacement kinetics, diagenesis                                                                                                                                                                                                                                                                                                                                                                                                                                                                                                                                                                                                                                                                                                                                                                                                                                                                                                                                                                                                                                                                                                 |
| 37                                                                   |                                                                                                                                                                                                                                                                                                                                                                                                                                                                                                                                                                                                                                                                                                                                                                                                                                                                                                                                                                                                                                                                                                                                                    |
| 38                                                                   |                                                                                                                                                                                                                                                                                                                                                                                                                                                                                                                                                                                                                                                                                                                                                                                                                                                                                                                                                                                                                                                                                                                                                    |
| 39                                                                   |                                                                                                                                                                                                                                                                                                                                                                                                                                                                                                                                                                                                                                                                                                                                                                                                                                                                                                                                                                                                                                                                                                                                                    |
| 40                                                                   |                                                                                                                                                                                                                                                                                                                                                                                                                                                                                                                                                                                                                                                                                                                                                                                                                                                                                                                                                                                                                                                                                                                                                    |
| 41
                                                   |                                                                                                                                                                                                                                                                                                                                                                                                                                                                                                                                                                                                                                                                                                                                                                                                                                                                                                                                                                                                                                                                                                                                                    |
| 42
                                                   |                                                                                                                                                                                                                                                                                                                                                                                                                                                                                                                                                                                                                                                                                                                                                                                                                                                                                                                                                                                                                                                                                                                                                    |
| 43
                                                   |                                                                                                                                                                                                                                                                                                                                                                                                                                                                                                                                                                                                                                                                                                                                                                                                                                                                                                                                                                                                                                                                                                                                                    |
| 77                                                                   |                                                                                                                                                                                                                                                                                                                                                                                                                                                                                                                                                                                                                                                                                                                                                                                                                                                                                                                                                                                                                                                                                                                                                    |

**45 Abstract**

The assessment of diagenetic overprint on microstructural and geochemical data gained from fossil archives is of fundamental importance for understanding palaeoenvironments. A correct reconstruction of 47 past environmental dynamics is only possible when pristine skeletons are unequivocally distinguished from 48 altered skeletal elements. Our previous studies (Casella et al. 2017) have shown that replacement of biogenic 49 carbonate by inorganic calcite occurs via an interface-coupled dissolution-reprecipitation mechanism. Our  $\mathcal{A}$ 50 51 studies have further shown that, for a comprehensive assessment of alteration, structural changes have to be 52 assessed on the nanoscale as well, which documents the replacement of pristine nanoparticulate calcite by diagenetic(nandrhombohedral calcite (Casella et al. 2018a, b). 53

In the present contribution we investigated six different modern biogenic carbonate microstructures for 54 their behaviour under hydrothermal alteration in order to assess their potential to withstand diagenetic 55 overprinting and to test the integrity of their preservation in the fossil record. For each microstructure we: 56 (a) examined the evolution of biogenic aragonite and biocalcite replacement by inorganic calcite, (b) 57 highlighted distinct carbonate mineral formation steps on the micrometre scale, (c) explored microstructural 58 changes at different stages of alteration, and (d) completed our studies with a statistical analysis of differences 59 60 in basic mineral unit dimensions in pristine and altered skeletons. The latter process enables an unequivocal determination of the degree of diagenetic overprint and discloses information especially about low degrees 61 of hydrothermal alteration. 62

**63 1 Introduction**

Biomineralised hard parts composed of calcium carbonate form the basis of studies of past climate 64 dynamics and environmental change. However, the greatest challenge that all biological archives face lies in 65 their capacity to retain original signatures, as alteration of them starts immediately upon death of the organism. 66 Biopolymers decay, and inorganic minerals precipitate within as well as at the outer surfaces of the hard tissue 67 (e.g., Patterson and Walter, 1994, Ku et al., 1999, Brand et al., 2004, Zazzo et al., 2004). 68 In Despite ongoing and extensive research, carbonate diagenesis remains only partly understood. Many 69 the latter that studies addressing the evolution of parameters which influence diagenetic alteration, are discussed in only a 70 qualitative manner (Brand and Veizer, 1980, 1981; Swart, 2015). In particular, deciphering the sequence of 71

V

those processes with many steps of alteration and unknown intermediate stages poses one of the major 73 problems in understanding carbonate diagenesis (Immenhauser et al., 2015a; Swart, 2015; Ullmann and Korte, 2015). Our previous studies on the shell of the modern bivalve Arctica islandica have shown that 74 experiment-based diagenetic alteration discloses microstructural and geochemical features that are 75 comparable to those found in fossils (Casella et al., 2017; Ritter et al., 2017). However, both studies covered 76 only the hard tissue of one taxon. For a more comprehensive understanding of microstructural and chemical 77 78 controls during diagenesis, the hard tissues of other archives have to be thoroughly examined. Accordingly, we extended our studies *to* hard tissues of other modern marine carbonate biomineralisers such as the 79 bivalves, A. islandica, and Mytilus edulis, the coral, Porites sp., and the gastropod, Haliotis ovina. With these 80 organisms we cover both major calcium carbonate phases and, further to that present in the shell of A. 81 islandica, five additional microstructures. When selecting model organisms for this study, care was taken to 82 investigate those for which fossil counterparts are used for palaeoclimate and palaeoenvironmental 83 84 reconstructions.

The bivalve Arctica islandica has been studied extensively in several scientific articles and fields 85 (e.g., Strahl et al., 2011; Ridgway and Richardson, 2011; Wanamaker et al., 2011; Ridgway et al., 2012; 86 Krause-Nehring et al., 2012; Karney 
[revised manuscript text omitted]
, in the prismatic 276 that shell layer of Haliotis ovina it is slow and patchy. In Haliotis ovina we find prismatic shell areas which are 277 completely replaced by calcite, while in other shell regions some aragonite is still preserved and frames the 伯 278 newly-formed calcite grains (Fig. A10B). In addition, the difference between pristine and altered prismatic 279 aragonite in Haliotis ovina (compare pole figures and MUD values of Figs. 4A and 4D) is such that in the 280 altered shell the size of aragonitic prisms increases, while the strength of aragonite co-orientation decreases. 281 282 This was observed in the pole figures and the decreased MUD value (compare Fig. 4A with right hand part, Fig. 4A cf. Fig. 4B Fig. 4A cf. Fig. 4D (right side) 283 framed in green, with Fig. 4D).  $\int T$  the comparison of Figs. 5A to 5C and Figs. A7A to A7B and A8 demonstrates that alteration of 284 Mytilus edulis calcite fibres at 175 °C, in the presence of a Mg-rich fluid, highly distorts the shape of the 285 fibres. In the pristine shell each calcite fibre is wrapped in an organic sheath. These decompose during 286 alteration and leave space for fluid permeation and inorganic calcite precipitation. (Crystal co-orientation 287

strength for fibrous calcite decreases markedly, from a MUD value of 381 in pristine to 79 in altered shells. In contrast to the shell part with the fibrous calcite microstructure, and similar to *Haliotis ovina* nacre, after 35 days of alteration,  $(175 \, ^{\circ}
[revised manuscript text omitted]
 are obtained from EBSD 460 See section measurements (ste for the definition of a grain in carbonate biological hard tissues, in Chapter 2.2.3: Grain 461 area evaluation for the determination of alteration). A grain is defined through misorientation angle relative 462 at in angle to neighbouring grains that is larger than a critical value, the critical misorientation value. Griesshaber et al. 463 (2013) determined empirically that a critical misorientation value of 2° best suits the microstructure of modern 464 carbonate biological hard tissues to differentiate between individual basic mineral units (e.g, fibres, tablets, 465 prisms, columns). Thus, we adopt a critical misorientation value of 2° to define a grain. They, dijacent grains 466 467 are recognized as two individual grains when one unit is tilted relative to the adjacent unit by more than 2°. The compilation in Fig. 12 clearly demonstrates the influence of the biogenic microstructure  $\#_0 n h e$ 468 ability To 469 A withstand grass sigle to alteration. The relation by y from 
[revised manuscript text omitted]
 the demonstrated that in Palaeozoic marine faunas faxa with calcitic skeletons prevail, this being an indication of 575 576 the preferential loss of aragonitic shells and skeletons, due to dissolution during diagenetic overprint (e.g., Wright et al. 2003, James et al. 2005). In addition to preferential carbonate phase preservation, experimental 577 studies document that the microstructure of the biogenic skeleton influences fossil preservation (e.g., Harper 578 1998, 2000; Kidwell 2005), leading to a possibly distorted notion of paleoecological and evolutionary 579 patterns. Accordingly, laboratory-based hydrothermal alteration experiments accounting for microstructural 580 as well as mineral phase variability offer important insights into the fate of carbonate hard tissues during a) 581 (shallow burial)early dissolution, and b) surviving dissolution and preservation in the fossil record. Do we see 582 resemblances between the microstructural, chemical outcome of our alteration results and microstructural and 583 geochemical features of fossilized hard tissues? 584 It is remarkable, that even though our experiments lasted only 35 days, were carried out at single 585 weve temperature and performed in the presence of only one type of alteration fluid there is much overlap between 586 Those products our experimental results and of carbonates that underwent diagenesis. Several decades ago Friedman (1964) 587

- and Land (1967) reported on the early diagenesis of skeletal carbonates and carbonate sediments exposed to 588 2 indicating that
- marine waters, the biological carbonates retained their original mineralogical and textural characteristics. 589 > They found that
- Brogenic aragonite was dissolved for the reprecipitation of low-Mg calcite, with high-Mg calcite being an 590 intermediate phase. Mg  $\frac{1}{6}$  removed from high-Mg calcite to yield low-Mg calcite, and, on a micrometer scale,
- 591

without textural change (Friedman 1964). Land (1967) observed that skeletal aragonite is altered much 593 quicker, relative to non-skeletal aragonite grains. Brand (1989) investigated the biogenic aragonite to calcite 594 transformation in fossil molluses (Boggy Formation, Oklahoma, USA) for an assessment of the degree of diagenetic overprint and the possible detection of the least-altered shells. When screening the mineralogy, 595 indicated microstructure and chemical composition detected that primary nautiloid aragonite is gradually 596 WAA replaced by diagenetic low-Mg calcite. During initial stages of alteration pacreous tablets fused to larger units 597 598 (Brand 1989). With further alteration amalgamated nacreous aragonite was replaced by fine- or coarse grained Jow-Mg calcite. Brand (1989) noted that the original aragonite determined the elemental and isotopic 599 composition of the calcite in the diagenetically altered shells. Brand (1989) further reports that grain size and 600 surface area play an important role for the process of overprint, Diagenetically overprinted aragonitic corals 601 were investigated by Sorauf (1980) and Tomiak et al. (2016). The authors observed that during early 602 Verb tense should be diagenesis, subsequent to organic matrix decomposition, aragonitic units formed through fusion of pristine 603 skeletal elements. Pore space become filled, prior to burial, with aragonite needles growing syntaxially on 604 lea existing biogenic aragonite. Subsequent submarine diagenesis leads to recrystallization of fibrous aragonite 605 21510 In 606 m to intermediate, micritic high-Mg calcite. Tomiak et al. (2016) and Regenberg et al. (2007) find at early Found diagenesis of coral aragonite and planktonic foraminifera calcite formation of new mineral overgrowth, with 607 the latter retaining, at first, the carbonate phase of the original pristine skeleton. Wardlaw et al. (1978), 608 Sandberg and Hudson (1983) and Martin et al. (1986) describe the influence of skeletal porosity as conduits 609 for alteration fluids during diagenesis. As the transformation of aragonite to calcite is driven by the greater 610 solubility of aragonite relative to that of calcite, at carbonate phase transformation the diagenetic pore fluid is 611 612 undersaturated with respect to aragonite while in saturated with respect to calcite (Maliva et al. 2000). differences in degree of 613 Hendry et al. (1995) proposed on the basis of supersaturation variation a 'two-water diagenetic system' with 614 a slow moving (at the dissolution-reprecipitation front) and a relatively fast moving (bulk pore water) 615 alteration fluid. In summary, some major steps of alteration may may be observed in our experiments (decomposition of 616 biopolymers, secondary porosity formation, amalgamation of mineral units, chemical evolution of the 617 alteration fluid) were also observed in nature. As, our experiments, that lasted only for a short time compared 618

> shown by to geologic time scales, show major and drastic steps of alteration take place at very initial time periods of alteration take place at very initial time periods of alteration take place at very initial time periods of alteration take place at very initial time periods of alteration take place at very initial time periods of alteration take place at very initial time periods of alteration take place at very initial time periods of alteration take place at very initial time periods of alteration take place at very initial time periods of alteration take place at very initial time periods of alteration take place at very initial time periods of alteration take place at very initial time periods of alteration take place at very initial time periods of alteration take place at very initial time periods of alteration take place at very initial time periods of alteration take place at very initial time periods of alteration take place at very initial time periods of alteration take place at very initial time periods of alteration take place at very initial time periods of alteration take place at very initial time periods of alteration take place at very initial time periods of alteration take place at very initial time periods of alteration take place at very initial time periods of alteration take place at very initial time periods of alteration take place at very initial time periods of alteration take place at very initial time periods of alteration take place at very initial time periods of alteration take place at very initial time periods of alteration take place at very initial take place at ver

**622 5 Conclusions**

Biogenic carbonate hard tissue5 form the basis of studies of past climate and environmental change. However, the greatest challenge that all biological proxies face lies in their capacity to retain their pristine signatures. With death of the organism, diagenetic overprinting starts immediately during which the original, biogenic signals are replaced by inorganic features. We investigated the behaviour of six biogenic carbonate samples and their associated microstructures at different degrees of hydrothermal alteration in order to evaluate their capacity to withstand alteration and thereby estimate their ability to be preserved in the fossil record. The main conclusions are:

good overview as in the abstract

1. Alteration of biogenic aragonite to inorganic calcite is fastest in hard tissues that contain primary porosity and are composed of irregularly shaped basic mineral units embedded in a network of biopolymer fibrils. The latter are easily destroyed and provide, together with primary pores, ample space for extensive fluid infiltration into and percolation through the hard tissue. This mode of overprint is observed for the prismatic shell layer of the gastropod *Haliotis ovina* and for the shell of the bivalve *Arctica islandica*. Overprinting of these hard tissues is fast and completed with the formation of irregularly shaped and randomly oriented calcite units.

2. The slowest alteration kinetics can be observed when biogenic nacreous aragonite is replaced by
inorganic calcite, irrespective of the mode of assembly of nacre tablets. Alteration takes proceeds in
four subsequent stages: (a) decomposition of biopolymers and formation of secondary porosity, (b)
lateral and longitudinal amalgamation of nacre tablets, (c) at the alteration front formation of a
compact zone within the hard tissue where the original microstructure is entirely erased according to the original bioaragonite phase is still retained, and (d) replacement by inorganic calcite.

3. The acicular microstructure of the stony coral *Porites sp.* is highly resistant to alteration. With 645 alteration aragonite needles fuse and form a compact aragonitic fabric, still retaining some

|     | , Replacement of                                                                                                       |
|-----|------------------------------------------------------------------------------------------------------------------------|
| 646 | morphological aspects of the pristine microstructure. Biological aragonite to inorganic calcite                        |
| 647 | replacement starts within the coral skeleton at the centers of calcification and proceeds from the latter              |
| 648 | inward into the hard tissue.                                                                                           |
| 649 | 4. For the investigated hard tissues we observe first the destruction of the microstructure and, second,               |
| 650 | the replacement by newly formed calcite.                                                                               |
| 651 | 5. Atteration in a fluid enriched in Mg, a high-Mg seam develops between the altered, compact aragonite                |
| 652 | and the newly formed calcite. With the progressive decrease of Mg concentration we can clearly trace                   |
| 653 | the chemical evolution of the alteration fluid at the biogenic aragonite to calcite interface.                         |
| 654 | 6. Statistical evaluation of differences in grain area size of pristine and altered skeletal equivalents 11 |
| 655 | demonstrates an increase in grain area within the altered hard tissues relative to that in the pristine                |
| 656 | skeleton. Hence, even though at the very early stages of alteration the original phase is retained,                    |
| 657 | overprint starts with the formation of overgrowths. This is most pronounced in the calcitic shell layer                |
| 658 | of Mytilus edulis and is least for the grains that constitute the shell of Arctica islandica. Thus, in the             |
| 659 | case of aragonitic tissue the survival of biological aragonite cannot be used as a $\frac{fellable indicator fo}{for}$ |
| 660 | pristine elemental and isotope signals. Statistical evaluation of grain area (basic mineral unit) values               |
| 661 | is a promising new tool for the estimation of the degree of diagenetic overprinting.                                   |

**662 7 Acknowledgements**

We thank the German Research Council (DFG) for financial support in the context of the 664 collaborative research initiative CHARON (DFG Forschergruppe 1644, Grant Agreement 665 Number SCHM 930/11-1).

**1036 Figures and captions**

---

## Author Comment (AC1) · 15 Oct 2018

**Reviewer 1**

| Comments of the reviewer | Reviewed manuscript | Author comments / revised manuscript |
|---|---|---|
| "The manuscript "Assessment of hydrothermal alteration on micro- and nanostructures of biocarbonates: quantitative statistical grain-area analysis of diagenetic overprint" by Casella et al. represents a substantial contribution to scientific progress in the field of biomineralization and addresses a very important scientific question, the alteration of biogenic hard tissues, which is within the scope of Biogeosciences. The applied methods are valid and clearly outlined and the interpretation and conclusions are strongly supported by the results. The references are appropriate. The conclusions are fundamental as the authors prove the different steps which ultimately lead to calcite re placement of biogenic carbonates, the possible occurrence of overprinted aragonite and importance of grain size, intergrain surfaces and porosity in controlling timing and extent of alteration. | | |
| However, the overall presentation is not very clear and the language is not always fluent and precise, so I think that the manuscript would benefit of moderate revisions, as discussed below." | | We accounted for the suggestions of reviewer 1 and rephrased many sections of the manuscript, improved fluency and organization of the text. In addition, the revised version of the manuscript was corrected by two native speaking co-authors (U. Brand and E. M. Harper). |
| "**General comments** In the Introduction, the authors should describe in more details the mineralogy of selected material (i.e. anticipate what it is written at p. 5)." | | The mineralogy, microstructural characteristics and biopolymer content is now described in greater detail for each selected species. See the results section: chapter 3.1 Microstructural |

| | | characteristics of modern bivalve, gastropod and coral skeletons. |
|---|---|---|
| "In paragraph 3.1, the authors should describe in greater details the microstructural characteristics of modern bivalve, gastropod and coral skeletons, which at the moment is only briefly addressed. For instance, A. islandica is known to have an outer homogenous/crossed lamellar/crossed acicular layer, an inner fine complex crossed lamellar layer and an irregular simple prismatic pallial myostracum. The brief description reported in 3.1 does not adequately inform the reader about the fabric and does not correspond to what subsequently written at p. 6 line 30 (aragonite prisms, but the microstructure of A. islandica is not prismatic see Dunca et 2009; Schone et al 2013)." | | The microstructure of the shell of *Arctica islandica* is described in the results section (chapter 3.1) in greater detail, according to the suggestion of Reviewer 1. |
| "I do not think that the microstructure of M. edulis can be described as consisting of calcite fibres. What shown in Fig. A2B are calcite prisms not fibers. Other figures may be more questionale, but the microstructures of M. edulis is foliated and prismatic (see for instance Brom & Szopa 2016; Carter et al. 2013). Eventually it is described as fibrous prismatic (Brom & Szopa 2016), a term which I do not agree with, but which is used (Carter et al. 1990) and it is distinct from the typical fibrous fabric of brachiopods." | | To our opinion the mineral units that compose the calcitic shell layer of the bivalve *Mytilus edulis* are fibres and are NOT prisms (e.g. Griesshaber et al. 2013, Acta Biomaterialia). The calcitic fibres in *Mytilus edulis* have a roundish outer morphology and can be few hundred micrometers long. Prisms are significantly shorter, thicker and are bounded at their sides by four to six planes. In order to be called a fibre mineral units in other carbonate biological hard tissues do not need to have the morphology of brachiopod fibres. We definitely want to keep to the term fibre for the mineral units in the calcitic shell layer of *Mytilus edulis*. |
| "An important issue is the time of decay of organic sheaths around the basic mineral units, which is | | Another manuscript focussing on organic contents in pristine and altered hard tissues is currently in |

| | | |
|---|---|---|
| not clearly indicated but just discussed as short." | | preparation. |
| "In paragraph 4.3, the authors should add the stratigraphic age of the described fossil material in order to support their conclusions." | | Paragraph 4.3 describes similarities between microstructural features that we observe in our hydrothermally altered specimens and microstrcutural/geochemical characteristics that we find in diagenetically overprinted fossil samples. For each example that we describe we state clearly a reference, where additional details such as stratigraphic age, sedimentological context, lithologies are stated. Our intention with paragraph 4.3 is to show that some microstructural features that we observe in our altered skeletons can also be observed in fossil samples. The intention of the paragraph is clearly stated at ist beginning. |
| "In the conclusions, the authors should report and give more enphasis to the important statement: "even though nacreous aragonite is still preserved as aragonite, it is an overprinted aragonite that, most probably, holds little of the original microstructural or geochemical signature"." | | This is corrected according to the suggestion of the reviewer. We added an additional point in the conclusions. |
| "Technical corrections
p. 1 line 34: sentence unclear" | "...The latter analysis enables an unequivocal determination of the degree of diagenetic overprint and discloses information especially about low degrees of hydrothermal alteration...." | "...The used statistical analysis derived from EBSD measurements enables an unequivocal determination of the degree of diagenetic overprint of biogenic carbonates, and discloses information especially on low degrees of hydrothermal alteration...." |
| "p. 2 line 9-10: long and complex sentence" | "...In particular, deciphering the sequence of those processes with many steps of alteration and unknown intermediate stages poses one of the major problems in understanding carbonate diagenesis (Immenhauser et al., 2015a; Swart, 2015; Ullmann and Korte, 2015)...." | "...In particular, deciphering the sequence of diagenetic evolution poses one of the major problems in understanding carbonate diagenesis (Immenhauser et al., 2015a; Swart, 2015; Ullmann and Korte, 2015)...." |

| | | |
|---|---|---|
| "p. 2 line 16 (and below in the text): sp. not italics" | "...*Porites sp*...." | "...*Porites* sp...." |
| "p. 2 line 25-30: I would describe before all the molluscs and only after the corals or viceversa." | "...As long-lived organisms, stony corals attract great interest for the reconstruction of palaeoclimates derived from skeletal oxygen isotopic compositions and major element abundances, as these geochemical signals vary in response to changes in seawater temperature (e.g., Meibom et al., 2007). It is assumed that $\delta^{234}U$ in sea water has remained constant in the past, thus, the comparison between present-day and decay-corrected $\delta^{234}U$ in sea water and in coral skeletons is a major tool for the detection of diagenetically altered corals. $\delta^{234}U$ values of the latter are higher relative to present day sea water (Hamelin et al., 1991; Stirling et al., 1995; Delanghe et al., 2002), while pristine corals exhibit a $^{234}U/^{238}U$ activity ratio similar to modern sea water (Henderson et al., 1993; Blanchon et al., 2009)...." | The order of the described specimens is based on their mineralogy and not on their animal class. To avoid repetitive descriptions of similar microstructures and to keep the manuscript as short as possible we keep to this order.

Order:
- *Arctica islandica* – aragonite
- *Porites* sp. – aragonite
- *Haliotis ovina* – aragonite (prisms & nacre)
- *Mytilus edulis* – calcite (fibres) & aragonite (nacre) |
| "p.3 line 2: correct M s edulis to M edulis test material: it would be better to indicate the dimension for the size (length, width, height?)" | "...In *H. ovina* the two layers are composed of aragonite, whereas the shell of *M.s edulis* consists of an outer calcite and inner aragonite layer...." | Changed accordingly

Dimensions of used specimens are given in subchapter 2.1 (Test materials) |
| "p. 4, line 24: the critical misorientation value. Sentence not finished" | "...A grain is defined as a region completely surrounded by boundaries across by which the misorientation angle relative to the neighbouring grains is larger than a critical value; the critical misorientation value...." | Wrong punctuation
"...A grain is defined as a region completely surrounded by boundaries across by which the misorientation angle relative to the neighbouring grains is larger than a critical value, the critical misorientation value...." |
| "p.5 line 19: correct H s ovina to H ovina" | "...Skeletons of *A. islandica*, *H.s ovina*, and *Porites* sp. consist entirely of aragonite, whereas *M. edulis* contains both carbonate phases, calcite and aragonite...." | Changed accordingly |
| "p. 5 line 20: add the type of fabric for A. | "...The shell of *A. islandica* is comprised of an | The fabric is given within this sentence: |

| islandica." | assemblage of irregularly-shaped and micrometre sized aragonitic basic mineral units (white stars in Fig. 1A), that are larger in the outer shell layer compared to basic mineral units of the inner shell layer (this study and Casella et al., 2017)...." | "...assemblage of irregularly-shaped and micrometre sized aragonitic basic mineral units..." |
|---|---|---|
| "p. 6 line 5: I do not think that the shell of M. edulis can be described as consisting of fibres, but prisms. Please check carefully also in the literature (Carter et al. 1990)." | | We do not agree with the comment of the referee and follow the definition for the calcite microstructure found in *M. edulis* as is described by Griesshaber et al. (2013) and Checa et al. (2014) in detail. Reference added.

Checa, A.G., Pina, C.M., Osuna-Mascaró, A.J., Rodrígues-Navarro, A.B. & Harper, E.M. (2014). Crystalline organization of the fibrous prismatic calcitic layer of the Mediterranean mussel *Mytilus galloprovincialis*. European Journal of Mineralogy 26: 495-505. |
| "p. 6 line 13-14: explain better this statement. The examples that follows are not strictly related to it." | | Inorganic calcite contents were determined in altered specimens using XRD. Those initially aragonitic specimens differed in their microstructure (nacreous, prisms, needle-like, fine-grained). It was observed that calcite formation in fine-grained *A. islandica* was fastest compared to the needle-like *Porites* sp. coral skeleton. Slowest replacement kinetics was observed for *H. ovina* containing aragonite prisms and nacre. The latter is most resistant to dissolution-reprecipitation reactions. |
| "p. 6 line 29: How long does it take for organic fibrils to be destroyed? What is the relationship between this processdecay and the "dormant" interval reported at p. 7?" | | The degradation of organic matrix is depending on the temperature applied, and its chemical components. In a previous study, our experiments showed that the organic matrix of brachiopds was destroyed after 2 days of thermal alteration at 400 |

| | | °C (cf. Casella et al., 2018a-b). |
|---|---|---|
| "p. 6 line 30 and p. 14: the microstructure of A islandica is not prismatic (Dunca et 2009; Schone et al 2013)" | "…At these conditions aragonite prisms in the shell…" | Schöne et al. (2013) describe the microstructure as „simple prismatic crystal fabric". We changed the text passage as follows according to our previous publication (Casella et al., 2017). "…At these conditions aragonite mineral units in the shell…." |
| "p. 7 line 25: again it is very important for this statement that the microstructure of the two taxa is described in great details, which is not at the moment." | "…However, it should be noted that even though there is a resemblance in basic mineral unit morphology and size, the existence of primary porosity, and the fabric of occluded biopolymers between the prismatic shell parts of *H. ovina* and *A. islandica*, the kinetics of carbonate phase replacement is distinct for the two microstructures (Figs. 2A, 2C). While in *A. islandica* shell replacement between carbonate phases is 25 rapid and extensive, it is slow and patchy in the prismatic shell layer of *H. ovina*…." | "…In the *A. islandica* shell, in which small irregularly shaped aragonite mineral units comprise the shell microstructure, replacement between carbonate phases is rapid and extensive, while replacement in the outer shell layer of *H. ovina,* which microstructure consists of aragonite prisms, is slow and patchy…." |
| "p. 8 line 13: "for both microstructures", is it true also for both mineralogical phases in M edulis?" | | Yes, calcite fibres and nacreous aragonite increase in grain size due to amalgamation. |
| "p. 8 line 22-25: the description of the "rise in porosity" is very important but it is not described enough clearly. It should be stated more clearly that pores are present in the biogenic carbonates." | "…A further characteristic caused by hydrothermal alteration is the significant rise in porosity within individual basic mineral units (Fig. 6). Even though the latter grow together at their perimeters (Fig. 7) a multitude of nanopores develop within them due to decomposition of biopolymer fibrils, which were present in the pristine hard tissue (e.g., Griesshaber et al., 2013; Casella et al., 2018a, 2018b)…." | "…A further characteristic caused by hydrothermal alteration is the significant rise in porosity within individual basic mineral units (Fig. 6). that grew together at their perimeters (Fig. 7). A multitude of nanopores developed within each biocarbonate crystal due to decomposition of biopolymer fibrils. The latter were located in primary pores within each crystallite of the pristine hard tissue (e.g., Griesshaber et al., 2013; Casella et al., 2018a, 2018b)…." |
| "p. 9 line 12-14. Sentence not clear." | "…Based on Mg-contents, in addition to the 'final' calcite, two high-Mg-calcite phases can be distinguished (Figs. 10, 11, A15), which seperate | "…In addition to secondary calcite, , two high-Mg-calcite phases can be distinguished (Figs. 10, 11, A15) based on Mg-content measurements. Both |

| | the 'final' calcite (calcite with a low Mg-contents) from the overprinted aragonite that was not yet replaced by calcite (Figs. 11, A15)...." | high-Mg calcites seperate the secondary calcite (calcite with low Mg-content) from the altered aragonite that was not yet replaced by calcite (Figs. 11, A15)...." |
|---|---|---|
| "p. 12, lines 3-4 and p. 15, line 3-4. Prismatic and nacre microstructures are among the shell microstructures, the ones having the higher amount of organic content, more than the homogeneous/fine complex crossed lamellar fabric in A. islandica. Having a high organic content they should have also a high primary porosity. Or is it a matter of pore size?" | "...Stacks of calcite fibres in *Mytilus edulis* and the nacreous tablet arrangements in *M. edulis* and *H. ovina* are the most compact microstructures investigated in this study. These materials lack primary porosities. Nonetheless, when the shells are altered, the extent of alteration-induced secondary porosity is high in the nacreous tablets, as the occluded intra-tablet membranes and inter-tablet fibrils decompose and create space for fluid circulation....."

"...Our study clearly shows that of the investigated aragonite microstructures the nacreous tablets are the most resistant to replacement by calcite, irrespective of the assembly pattern of the tablets in columns or sheets. Porosity closure and basic mineral unit (nacre tablet), amalgamation recasts at first completely the original microstructure, however, with the 5 preservation of the original phase (Figs. 9A, A17A, A17B)...." | In the pristine shells, biopolymer matrices are surrounding each mineral unit and may also be located within each crystal as fibrils or network located within primary pores. Due to decomposition of the organic matter caused by alteration these pores become visible. Additionally, secondary porosity concomitantly is formed during dissolution-reprecipitation reactions when alteration is applied.

Changed to:
"...These materials scarcely contain primary porosities...." |
| "p. 12, line 11 Regenberg et al. 2007, comma missing after et al." | | Changed accordingly |
| "p. 12, line 29-35. This part is not very clear and not very well fitted into the paragraph.
Also should not it be placed in the results?" | "...The least difference in grain area change between pristine and most altered states was observed for *A. islandica* aragonite (Fig. 12A), while the most significant difference occurred for *M. edulis* fibrous calcite (Fig. 12E). For *Porites* sp. acicular aragonite and *H. ovina* prismatic and nacreous aragonite, we find a perceivable, but small difference in grain area size between the | "...The least difference in grain-areas between the pristine and most altered states was observed for *A. islandica* aragonite (Fig. 12A), while the most significant difference occurred for *M. edulis* fibrous calcite (Fig. 12E). For *Porites* sp. acicular aragonite. and *H. ovina* prismatic and nacreous aragonite we find a perceivable, but small difference in grain areas between the pristine and the most altered |

| | | |
|---|---|---|
| | pristine and the most altered states. For *M. edulis* nacre the majority of grain area data overlap for this microstructure, as well for some large grains formed in the altered shell (Fig. A16). …" | states. For pristine *M. edulis* nacre the majority of grain-area data overlap for this microstructure, as well for amalgamated nacre after applied hydrothermal alteration (Fig. A16). …" |
| "p. 14 line 5-13. Very important process, to be described more clearly. It is nor clear why "Carbonate phase alteration kinetics in A. islandica shell is sluggish at first" and why porosity "explains the little difference in mineral grain area"." | "…The large number of small basic mineral units gives rise to exceedingly large surface areas where the fluid can get into contact with the mineral. Carbonate phase alteration kinetics in *A. islandica* shell is sluggish at first. However, once the nucleation barrier is overcome and the alteration process is started, it proceeds very rapidly (Figs. 2A, A4A; Casella et al., 2017). Thus, overgrowth of inorganic aragonite in voids and basic mineral unit amalgamation might well be masked by the almost instantaneous replacement of biogenic aragonite by inorganic calcite in the microstructure of *A. islandica* shells. The high volume of interconnected porosity in *A. islandica* explains why alteration becomes active after only a short time in contact with diagenetic fluids. Moreover, the topological characteristics of porosity facilitate the coupling between the rate of aragonite dissolution and calcite reprecipitation. This, in turn, explains the little difference in mineral grain-area found in the hard tissue of *A. islandica* between the pristine and the most altered states. …" | The sluggish alteration kinetics is described in detail by Casella et al. (2017). In the present manuscript we refer to the publication above as data on *A. islandica* completes the presented research on hydrothermal alteration of mainly biogenic aragonites. "…The large number of small basic mineral units gives rise to exceedingly large surface areas where the fluid can get into contact with the mineral at grain boundaries and nanopores found within each mineral unit. Carbonate phase alteration kinetics in *A. islandica* shell is sluggish at first. However, once the nucleation barrier is overcome and the alteration process is started, it proceeds very rapidly (Figs. 2A, A4A; Casella et al., 2017). Thus, overgrowth of inorganic aragonite in voids and basic mineral unit amalgamation might well be masked by the almost instantaneous replacement of biogenic aragonite by inorganic calcite in the microstructure of *A. islandica* shells. The high volume of interconnected porosity in *A. islandica* and the presence of thermodynamically less stable biogenic aragonite explain why alteration becomes active after only a short time in contact with diagenetic fluids. Moreover, the topological characteristics of porosity facilitate the coupling between the rate of aragonite dissolution and calcite reprecipitation. This, in turn, explains the |

| | | little difference in mineral grain-area found in the hard tissue of *A. islandica* between the pristine and the most altered states...." |
|---|---|---|
| "p. 14 line 22-23. "the increased prevalence of the nacreous shell layer of M. edulis relative to calcitic shell layers in seashore sediments". This statement should be better explaned and supported." | "...The nacreous shell part grows into a compact entity and becomes sealed and protected against fluid infiltration. This explains the observation of remnants of nacreous shell areas surrounded by calcite (Brand, 1994) as well as the increased prevalence of the nacreous shell layer of *M. edulis* relative to calcitic shell layers in seashore sediments...." | We explain the statement in more detail. |
| "p. 15 line 5-6: sentence not clear" | "...Porosity closure and basic mineral unit (nacre tablet) amalgamation at first completely recasts the original microstructure, but with the retention of the original phase (Figs. 9A, A17A, A17B)...." | "...Reprecipitation processes and amalgamation of neighbouring nacre tablets at first completely recasts the original microstructure, but with the retention of the original phase (Figs. 9A, A17A, A17B)...." |
| "p. 15, "It has been further demonstrated that in Palaeozoic marine faunae taxa with calcitic skeletons prevail". The authors have to add fossil before marine fauna" | | Changed accordingly |
| "p. 16 line 26: tissue forms or tissues form" | "...Biogenic carbonate hard tissue form the basis of studies of past climate and environmental change...." | "...Biogenic carbonate hard tissues form the basis of studies of past climate and environmental change...." |
| "p. 17 line 26: "Thus, in the case of aragonitic tissue the survival of biogenic aragonite" better to correct into "Thus, in the case of aragonitic tissue the survival of biogenic aragonite"?" | "...Thus, in the case of aragonitic tissue the survival of biogenic aragonite cannot be used as a distinct indicator for pristine elemental and isotope signals...." | The comment of the reviewer corresponds to the text passage given in our manuscript. → no further changes needed |
| "References: Crippa & Raineri (2015) is in the text but it missing from the ref list" | "...to mark the former Pliocene–Pleistocene boundary (e.g., Crippa and Raineri, 2015;..." | Reference added
Crippa, G. and Raineri, G.: The genera Glycymeris, Aequipecten and Arctica, and associated mollusk fauna of the Lower Pleistocene Arda River section (Northern Italy), Riv. Ital. Paleontol. Stratigr., 121, 61-101, 2015. |

---

## Author Comment (AC2) · 15 Oct 2018

**Reviewer 2**

| Comments of the reviewer | Reviewed manuscript | Author comments / revised manuscript |
|---|---|---|
| " The authors present an interesting study of the process of alteration of six selected microstructures when submitted to the action of hydrothermal fluids. They assess the process of transformation of either biogenic aragonite or calcite into inorganic calcite with time. Although the simulated diagenetic alteration conditions are probably a small portion of the whole range existing in ambient conditions, the study is highly meritorious. Particularly, the application of EBSD analysis provides a wealth of information. Although I am not expert on diagenesis, I would be surprised if such a picture of the progression of a simulated diagenetic alteration ever existed. The study is significant in its field, and the conclusions are backed by data. It is technically very sound. The statistics is consistent. The Ms is well written and profusely illustrated. The selected microstructures are representative of the range present in molluscs. I am only concerned about the dissonance between the references in text and in the reference list. Many in-text references are not found at the end and the other way round (see at the end). This has to be amended before publication." | | It is double checked that all references that are stated in the text are in the reference list and vice versa. |
| **"Minor comments:** Abstract.- I do not know if the journal allows for the presence of references within the abstract. | | In the revised version of the manuscript we are not citing any references in the abstract, even though this would be possible. |

| | | |
|---|---|---|
| Please check." | | |
| "Page 2, line 8.- "Despite ongoing and extensive research", do you mean 'previous' extensive research." | "...Despite ongoing and extensive research, carbonate diagenesis remains only partly understood...." | "...Despite previous extensive research, carbonate diagenesis remains only partly understood...." |
| "Page 3, line 2.- delete "s" after "M."" | | "s" deleted |
| "Page 3, line 8.- 'were carried out' should not be italicized; "tissue" should be plural." | | Changed accordingly |
| "Page 5, line 13.- Correct spelling is Carvajal." | | Changed accordingly |
| "Page 5, lines 25-26.- "consists", "consisting"; avoid redundancy." | "...The skeleton of the modern stony coral *Porites* sp. consists of an assemblage of spherulites consisting of aragonitic needles and fibrils..." | "...The skeleton of the modern stony coral *Porites* sp. consists of an assemblage of spherulitic aragonite needles and fibrils..." |
| "Page 5, line 28.- "When sectioned in 2D", section, by definition, provides a 2D view; 'in 2D' can be deleted." | | Changed accordingly |
| "Page 7, line 7.- "sp.", here and elsewhere, should not be italicized." | | Changed accordingly |
| "Page 7. line 31.- "compare Fig. 4A with right hand part, framed in green with Fig. 4D", I would suggest 'compare Fig. 4A with Fig. 4D, right hand part, framed in green'." | | Changed accordingly |
| "Page 9, line 8.- Comma after "shows"." | "...However, as the phase map in Fig. 9E shows a phase..." | "...However, as the phase map in Fig. 9E shows, a phase replacement of biogenic..." |
| "Page 10, line 4.-Comma after "sediments"." | "...with the death of the organism and burial in sediments biomineralised..." | "...Accordingly, with the death of the organism and burial in sediments, biomineralised hard..." |
| "Page 10, line 20.- Correct spelling is 'Etschmann'. Jonas et al. 2015 appears as 2017 in References." | | Changed accordingly and Jonas et al. (2017) replaced by Jonas et al. (2014) in the list of references. |
| "Page 11, lines 13-14.- The sentence contained there is either incomplete or the initial "In" needs to be removed." | "...In the absence of primary porosity and/or secondary porosity that should have been generated at early stages of alteration is attributed to the positive molar volume change involved in the aragonite by calcite replacement...." | "...The absence of primary porosity and/or secondary porosity that should have been generated at early stages of alteration is attributed to the positive molar volume change involved in the aragonite by calcite replacement..." |
| "Page 13, line 21.- Correct spelling is "Fernández- | | Changed accordingly |

| | | |
|---|---|---|
| Díaz".” | | |
| "Page 14, lines 22-23.- "as well as the increased prevalence of the nacreous shell layer of M. edulis relative to calcitic shell layers in seashore sediments." Do the experimental conditions really simulate the diagenetic conditions in nearshore sediments? If so, please provide additional data and/or references." | | Hydrothermal experiments applied do not simulate realistic diagenetic conditions in sediments. |
| "Page 15, line 3.- Replace "the nacreous tablets are" by 'nacre is'; the microstructure is nacre." | | Changed accordingly |
| "Page 15, line 20.- Delete "of"." | | Changed accordingly |
| "Page 15, line 31.- "Biogenic aragonite was dissolved for the reprecipitation of low-Mg calcite", this is odd; please rewrite." | | This corresponds to a basic dissolution-precipitation reaction by which the thermodynamically less stable $CaCO_3$ phase (biogenic aragonite) dissolves. Thus, the solution becomes supersaturated with respect to the thermodynamically more favourable phase (calcite) which precipitates and replaces the former aragonite. |
| "Page 16, line 9.- Sorauf (1980) appears as 1981 in References." | | Changed accordingly |
| "Page 16, line 28.- Comma after "immediately"." | | Changed accordingly |
| "age 17, line 11.- 'replacement' instead of "replaced"." | | Changed accordingly |
| "Page 18, line 14.- "Their" should be lower case." | | Changed accordingly |
| "Page 19, line 33.- "tuberculate" should be 'tuberculata'." | | Changed accordingly |
| "Page 21, line 13.- "cabonates", 'carbonates'." | | Changed accordingly |
| "THE following in-text references have not been found in the References list (per order of appearance):
 - Patterson and Walter, 1994
 - Ku et al. 1999 | | References added:
 Patterson, W. P. and Walter, L. M.: Sydepositional diagenesis of modern platform carbonates: evidence from isotopic and minor element data, Geology, 22, 127-130, 1994. |

| | | |
|---|---|---|
| - Brad et al. 2004
- Zazzo et al. 2004
- Immerhauser et al. 2015
- Ridgway and Richardson 2011
 - Krause-Nehring et al. 2012
- Rodgway et al. 2012
- Crippa and Raineri 2015
- Blanchon et al. 2009
- Hahn et al. 2012
- Nidiyasari et al. 2015
- Brown et al. 1962
- Cardew and Davey 1985
- Regenberg et al. 2007
- Hover et al. 2001
- Cherns et al. 2008
 - Wright et al. 2003
- James et al. 2005
- Harper 1998
- Harper 2000
- Kidwell 2005
- Land 1967 " | | Ku, T. C. W., Walter, L. M., Coleman, M. L., Blake, R. E., and Martini, A. M.: Coupling between sulfur recycling and syndepositional carbonate dissolution: evidence from oxygen and sulfur isotope composition of pore water sulfate, South Florida Platform, U. S. A., Geochim. Cosmochim. Acta, 63(17), 2529-2546, 1991.
Brand, U.: Carbon, oxygen and strontium isotopes in Paleozoic carbonate components: an evaluation of original seawater-chemistry proxies, Chem. Geol., 204(1-2), 23-44, 2004.
Zazzo, A., Lécuyer, C., Sheppard, S. M., Grandjean, P., and Mariotti, A.: Diagenesis and the reconstruction of paleoenvironments: a method to restore original $\delta^{18}$O values of carbonate and phosphate from fossil tooth enamel, Geochim. Cosmochim. Acta, 68(10), 2245-2258, 2004.
Immenhauser, A., Schöne, B. R., Hoffmann, R., and Niedermayr, A.: Mollusc and brachiopod skeletal hard parts: intricate archives of their marine environment, Sedimentology, 63(1), 1-59, 2015.
Ridgway, I. D., Richardson, C. A., and Austad, S. N.: Maximum shell size, growth rate, and maturation age correlate with longevity in bivalve molluscs, J. Gerontol. A Biol. Sci. Med. Sci., 66(2), 183-90, 2011.
Krause-Nehring, J., Brey, T., and Thorrold, S. R.: Centennial records of lead contamination in northern Atlantic bivalves (*Arctica* |

|  |  | *islandica*), Mar. Pollut. Bull., 64(2), 233-40, 2012.

Blanchon, P., Eisenhauer, A., Fietzke, J., and Liebetrau, V.: Rapid sea-level rise and reef back-stepping at the close of the last interglacial highstand, Nature, 458, 881-884, 2009.

Hahn, S., Rodolfo-Metalpa, R., Griesshaber, E., Schmahl, W. W., Buhl D., Hall-Spencer, J. M., Baggini, C., Fehr, K. T., and Immenhauser, A.: Marine bivalve shell geochemistry and ultrastructure from modern low pH environments: environmental effect versus experimental bias, Biogeosciences, 9, 1897-1914, 2012.

Nindiyasari, F., Ziegler, A., Griesshaber, E., Fernández-Díaz, L., Huber, J., Walther, P., and Schmahl, W. W. (2015) Effect of hydrogel matrices on calcite crystal growth morphology, aggregate formation, and co-orientation in biomimetic experiments and biomineralization environments, Cryst. Growth Des., 15(6), 2667-2685, 2015.

Brown, W. H., Fyfe, W. S., and Turner, F. J.: Aragonite in California glaucophane schists, and the kinetics of the aragonite-calcite transformation, J. Petrol., 3, 566-582, 1962.

Cardew, P. T. and Davey, R. J.: The kinetics of solvent-mediated phase transformations, Proc. Roy. Soc. London, Ser. A Math. Phys. Eng. Sci., 398, 415-428, 1985.

Regenberg, M., Nürnberg, D., Schönfeld, J., and |
| --- | --- | --- |

Reichart, G. J.: Early diagenetic overprint in Caribbean sediment cores and its effect on the geochemical composition of planktonic foraminifera, Biogeosciences, 4, 957-973, 2007.

Hover, V. C., Walter, L. M., and Peacor, D. R.: Early marine diagenesis of biogenic aragonite and Mg-calcite: new constraints from high-resolution STEM and AEM analyses of modern platform carbonates, Chem. Geol., 175, 221–248, 2001.

Cherns, L., Wheeley, J. R., and Wright, V. P.: Taphonomic windows and molluscan preservation, Palaeogeogr. Palaeoclimatol. Palaeoecol., 270, 220-229, 2008.

Wright, V. P., Cherns, L., and Hodges, P.: Missing molluscs: field testing taphonomic loss in the Mesozoic through early largescale aragonite dissolution, Geology, 31, 211-214, 2003.

James, N. P., Bone, Y., and Kyser, K. T.: Where has all the aragonite gone? Mineralogy of Holocene neritic cool-water carbonates, Southern Australia, J. Sediment. Res., 75(3), 454-463, 2005.

Harper, E. M.: The fossil record of bivalve molluscs, in: Donovan S. K. and Paul C. R. C. (eds.) The adequacy of the fossil record, John Wiley and Sons, Chichester, 243-267, 1998.

Kidwell, S. M.: Shell composition has no net impact on large-scale evolutionary patterns in mollusks, Science, 307, 914-917, 2005.

Land, L. S.: Diagenesis of skeletal carbonates, J.

| | | |
|---|---|---|
| | | Sediment. Res., 37(3), 914-930, 1967.

General comments:
- Brad et al, 2004 → Brand, 2004
- Ridgway and Richardson, 2011 → Ridgway et al., 2011
- Rodgway et al., 2012 → Ridgway et al., 2011
- Crippa and Raineri, 2015 → already added
- Harper, 2000 → deleted |
| "THE following references in the References list have not been found in the main text:
- Addadi et al. 2006
- Allison et al. 2007
- Altree-Williams et al. 2017
- Barthelat and Spinosa 2007
- Bathurst 1975
- Böhm et al. 2006
- Brahmi et al. 2012
- Brocas et al. 2013
- Butler et al. 2009
- Cartwright and Checa 2007
- Checa et al. 2006
 - Checa et al. 2009
- Checa et al. 2011
- Cohen et al. 2001
- Currey et al. 2001
- Dauphin et al. 1989
- Elliot et al. 2003
- Gries et al. 2009
- Grossman et al. 1993
- Heiss 1994
- Hippler et al. 2009 | | References included into the manuscript:
- Addadi et al., 2006
- Allison et al., 2007
- Butler et al., 2009
- Checa et al., 2006
 - Checa et al., 2009
- Checa et al., 2011
- Korte et al., 2005
- Marchitto et al., 2000
- McGregor and Gagan, 2002
- Barthelat and Spinosa, 2007
- Bathurst, 1975
- Böhm et al., 2006
- Cohen et al., 2001
- Elliot et al., 2003
- Heiss, 1994
- Hippler et al., 2009
- Cartwright and Checa, 2007
- Gries et al., 2009
- Jackson et al., 1988
- Levi-Kalisman et al., 2001
- Li et al., 2006
- Marin and Luquet, 2004
- Mayer, 2005 |

| | | |
|---|---|---|
| - Hubbard et al. 1990
- Jackson et al. 1988
 - Korte et al. 2005
 - Levi-Kalisman et al. 2001
- Li et al. 2006
- Marchitto et al. 2000
- Marin and Luquet 2004
- Mayer 2005
- McGregor and Gagan 2002
- Metzler et al. 2007
- Morton 2011
- Nudelman et al. 2006
- Nudelman et al. 2008
- Oeschger and Storey 1993
- Parkinson et al. 2005
- Putnis et al. 2005
- Raffi 1986
- Richardson 2001
- Rüggeberg et al. 2008
- Sanchez et al. 2005
- Schöne et al. 2004
- Schöne et al. 2005a
 - Schöne et al. 2005b
- Schöne and Surge 2012
- Taylor 1976
- Wanamaker et al. 2008
- Wang and Gupta 2011
- Wang et al. 2011 " | | - Metzler et al., 2007
- Morton, 2011
- Raffi ,1986
- Richardson, 2001
- Rüggeberg et al., 2008
- Schöne et al., 2004
- Schöne et al., 2005a
 - Schöne et al., 2005b
- Schöne and Surge, 2012
- Wanamaker et al., 2008

References deleted:
- Altree-Williams et al., 2017
- Brahmi et al., 2012
- Brocas et al., 2013
- Currey et al., 2001
- Dauphin et al., 1989
- Grossman et al., 1993
- Hubbard et al., 1990
- Nudelman et al., 2006
- Nudelman et al., 2008
- Oeschger and Storey, 1993
- Parkinson et al., 2005
- Putnis et al., 2005
- Sanchez et al., 2005
- Taylor, 1979
- Wang and Gupta, 2011
- Wang et al., 2011 |
| "Figure A16.- Panel A is simply the sum of B and C. It can be simplified." | | Figure A16 was designed to be divided into three panels. Thus, the reader is able to directly compare the grain size scattering for both nacre containing specimens: *M. edulis* and *H. ovina* (panel A). Panels |

|  |  | B and C, however, each show data obtained only for one specimen so that no data points obtained for species A are hidden by data points of species B. |
|---|---|---|

---

## Author Comment (AC3) · 15 Oct 2018

**Reviewer 3**

| Comments of the reviewer | Reviewed manuscript | Author comments / revised manuscript |
|---|---|---|
| " Through a series of well planned experiments followed by analyses at the micrometer and nanometer scale, the authors have documented the effect of medium-temperature hydrothermal solutions on the calcium carbonate shells of six different species displaying five different types of microstructural units. They have demonstrated the (expected) importance of the amount of time the materials spend in the hydrothermal solutions on the degree of shell alteration. What may be less expected, however, are the documented differences in degree of alteration under the same experimental conditions, but in different species and in different microstructures. The authors focused on observable structural changes, but also investigated some aspects of compositional changes due to interaction between the calcite/aragonite shells and the Mg-Na-Cl solutions at 175 C. Changes in grain size and orientation within selected areas of the shells were monitored through EBSD and FE-SEM. Changes in grain size were found to be the most sensitive indicators of hydrothermal alteration. The paper is replete with excellent photomicrographs and EBSD color-coded images showing grain orientation and phase identification. The results of this extensive work have important implications for our ability to recognize which biocarbonate fossil materials are most pristine and therefore most appropriate for | | |

| | | |
|---|---|---|
| chemical and isotopic analysis that may be used to reconstruct the details of past environments." | | |
| "In summary, this is a very good paper that provides well-documented, quantitative experimental observations on the species-specific, microstructure-specific aqueous alteration of biocarbonate shells. I believe that this work marks a large step forward in the understanding of the preservation/fossilization process of shells. As the authors point out, their work also sheds light on how differential alteration effects can account for differential preservation of shell material. The implications of this result alone are extremely important for the interpretation of the fossil record. The results of this study also will guide the evaluation of fossil materials for isotopic and geochemical analysis with the goal of environmental reconstruction." | | |
| "The current document requires some minor re-writing, as well as the addition of materials indicated in the comments below and on the attached annotated copy of the manuscript. I believe this paper will be of strong interest to the readership of this journal." | | |
| "There are some improvements that could be made in the paper. Immediately below are more general suggestions. Below them are more specific comments, questions, and suggestions keyed to identified lines in the manuscript. A scanned copy of the annotated manuscript is also provided with additional detailed suggestions." | | |
| "The abstract is not appropriate as written. The first paragraph provides too much background | | We corrected our abstract and it includes now all suggestions of reviewer 3. |

| | | |
|---|---|---|
| information that instead should be in the introduction. The abstract, however, does not include necessary details about the experimental procedure and the types of analyses performed." | | |
| "This paper covers a huge amount of information on several morphologically different parts of the shells of six species, undergoing alteration for up to 35 days. The authors have included some figures and tables to guide the reader through this information so that the discussion and conclusions will be convincing. Any additional techniques to help the reader keep track of, organize, and compare these data/observations would further enhance the impact of this work." | | We don't understand this comment. To our opinion we did organize the manuscript well. **Results** 3.1 We describe microstructural characteristics of pristine samples. 3.2 We describe microstructural characteristics of hydrothermally altered samples 3.3 We describe alteration pathways **Discussion** 4.1 We describe the alteration process for non-biological and biological samples 4.2 We describe the effect of the microstructure at alteration 4.3 Implications for the preservation of carbonate hard tissues in the fossil record For a better understanding we rephrased large parts of the manuscript. Our descriptions and statements are now more target-oriented and easier to understand. |
| "As much as I appreciate the level of detail presented on the individual samples and their respective microstructures, I suggest that the authors take a fresh look at the paper to determine if they reasonably can condense it further. At times, the density of information approaches overwhelming." | | We condensed the text of the manuscript, but cannot omit any Figures. |

| | | |
|---|---|---|
| "A distinction that should be addressed in the paper is the difference intended by the authors between the terms "overprinted" and "replaced." This is an important distinction especially for (future?) work on chemical and isotopic signatures that may or may not be transferred to the "new" solids formed after hydrothermal alteration." | | For a better understanding we give a definition of the subsequently described terms in the methods section:
(i)    Throughout the manuscript we use the term 'overprinted' for altered, in this work hydrothermally, in the laboratory.

(ii)    We use the term 'replaced' only when the original carbonate phase is replaced by a new carbonate phase.

(iii)    If we describe alteration of fossil samples we use the term 'diagenetically altered' |
| "The tracking of Mg and its importance in the alteration products are not well explained; nor is the addition of only that one element to the NaCl hydrothermal solution. Given the so-called "Mg-poisoning" of calcite, some more discussion in needed. Does all the Mg that occurs in the secondary Mg-calcites come from that introduced by the hydrothermal solutions, or is there Mg in the pristine aragonite?" | | We checked for *Arctica islandica* the Mg-content in the pristine and the hydrothermally altered equivalent (Casella et al. Biogeosciences 2017). There is no Mg in the pristine aragonite, the Mg content that we detect in the hydrothermally altered skeletons comes from the alteration solution that was enriched in Mg. |
| "Question that is more appropriate for on-line discussion:
What should be the next steps in experimental and analytical studies to evaluate how well texturally altered samples (as shown by changes in grain-area evaluation) of bio genic carbonate hard tissues retain their chemical (incl. trace-element) and isotopic signatures?" | | Further important studies should include:
(i)    Longer alteration times
(ii)    Lower alteration temperatures
(iii)    Alteration at higher pressures
(iv)    Stable isotope measurements with a high spatial resolution performed on transects through the skeletons at different stages of alteration |

| | | |
|---|---|---|
| | | (v)    Performance of the same study with chemically different fluids
For a better understanding of diagenesis some of the experiments are currently carried out in our laboratory. |
| "Attached here is a scanned copy of the annotated manuscript, which has numerous detailed suggested changes. In the right-hand margins are circled numbers that are linked to comments and questions below." | | All annotated suggestions are changed. |
| "Point 1, page 2: The information in the first paragraph of the abstract would be better placed in the introduction. One usually avoids including reference citations in an abstract." | | It is changed. |
| "Point 2, page 2: The second paragraph in the abstract is a good overview of the results and implications of the work. However, more detail is needed on the experimental procedure and analytical techniques used." | | It is changed. |
| "Point 3, page 4, line 111: The term "basic mineral unit" should be defined here, where it is first introduced, or the placement of the definition should be stated here." | | It is changed. |
| "Point 4, page 4, lines 115-117: The parts of this statement that are not currently in the abstract would be very useful there." | | It is changed. |
| "Point 5, page 6, lines 169-171: It would seem that the authors wish to provide an overview for or give a "heads-up" to the reader regarding the upcoming experiments and their results. However, this sentence is very compact and uses several not-yet defined terms, making it unsuccessful in this inferred purpose." | | The used statistical approach for grain area evaluation is described in more detail. |

| | | |
|---|---|---|
| "Point 6, page 6, lines 175-176: The term "burial fluid" should be explained and its chosen composition justified, especially given the importance of the Mg component, as revealed by the experimental results." | "...Laboratory-based hydrothermal alteration experiments mimicked burial diagenetic conditions. In all experiments pieces of shells or skeletons up to 2 cm x 1 cm of modern *A. islandica*, modern *M. edulis*, modern *Porites* sp., and modern *H. ovina* were placed inside a polytetrafluoroethylene (PTFE) vessel together with 10 mL of simulated burial fluid (100 mM NaCl + 10 mM MgCl$_2$ aqueous solution) and sealed with a PTFE lid...." | "...Laboratory-based hydrothermal alteration experiments mimicked burial diagenetic conditions in terms of fluid composition. The latter is based on previously conducted hydrothermal experiments with a defined concentration of Mg that is comparable to Mg contents found in natural diagenetic environments. However, pressure conditions could not be adjusted to those of natural burial diagenesis due to the experimental design (closed system), and thus, corresponded to vapour pressure of water at the given temperature. In all experiments pieces of shells or skeletons up to 2 cm x 1 cm of modern *A. islandica*, modern *M. edulis*, modern *Porites* sp., and modern *H. ovina* were placed inside a polytetrafluoroethylene (PTFE) vessel together with 10 mL of simulated burial fluid (100 mM NaCl + 10 mM MgCl$_2$ aqueous solution) and sealed with a PTFE lid...." |
| "Point 7, page 9, line 232: Although the term "microstructure" was defined earlier, it would be useful to remind the reader how that term is being applied in this paper. It is not a general term, but rather a stand-in for several types of mineralogical-structural morphologies focused on in the paper. Perhaps the parenthetical interjection "(e.g., calcite fibers, prismatic aragonite, nacreous aragonite)" would be useful here." | | We implemented this suggestion of reviewer 3 wherever we could in the manuscript. |
| "Point 8, page 10, lines 261-262: It would helpful to the reader to be told how long these intervals of time are that give rise to the changes observed." | "...With progressively longer alteration large and randomly oriented calcite crystals develop in the coral skeleton (Figs. 3B, 3C, 3D, A5D)...." | "...With progressively longer alteration times up to 35 days large and randomly oriented calcite crystals develop in the coral skeleton (Figs. 3B, 3C, 3D, A5D)...." |

| | | |
|---|---|---|
| "Point 9a, page 10, line 278: Does "other shell regions" refer to other parts of the prismatic area or to regions other than in the prismatic area?" | | "Other shell regions" refers to the prismatic aragonite. |
| "Point 9b, page 13, lines 355-356: The excellent question is posed here of what determines the preservation potential of a fossil archive. This question should be re-stated and succinctly answered toward the end of the paper." | | We changed our manuscript according to the highly valuable comment of reviewer 3.

We re-state this question and give some answer in the very last paragraph of chapter 4.3. |
| "Point 10, page 15, lines 408-410: This is an important discussion of a complicated set of experiments. It is difficult for the uninitiated reader to understand what the focal points are. Perhaps some additional details about the original experiments would clarify the discussion." | | We rephrased this section of the manuscript so that it becomes easier for the reader. |
| "Point 11, page 16, lines 446-447: I agree with this statement. At some point in the paper, though, it should be acknowledged that the present work does not directly address the issue of the geochemical fidelity of the recognizably/quantifiably altered carbonates." | | This is changed accordingly. |
| "Point 12, pages 20 and 21, lines 563 and 566: There seems to be a self-contradiction between these two lines with regard to the (lack of) fidelity in the geochemical signature." | | Figure 9D shows that some regions of the altered nacreous aragonite contain different concentrations of Mg. Furthermore, nacre tables amalgamated into larger units.
Both features cannot be observed in pristine *H. ovina* nacre. |
| "Please also note the supplement to this comment: https://www.biogeosciences-discuss.net/bg-2018-249/bg-2018-249-RC3-supplement.pdf" | | Changed accordingly |

---

## Author Comment (AC6) · 15 Oct 2018

M. Sc. Laura Antonella Casella Department of Earth and Environmental Sciences Ludwig-Maximilians-University Munich Theresienstr. 41/II 80333 München, Germany Telephone: +49 89 2180 4354 Email:Laura.Casella@lrz.uni-muenchen.de

Munich, October 15th 2018

Dear Dr. de Nooijer,

[Figure]

Please find attached our reply to all referee's comments and the revised manuscript entitled: "Hydrothermal alteration of aragonitic biocarbonates: assessment of micro- and nanostructural dissolution-reprecipitation and constraints of diagenetic overprint from quantitative statistical grain-area analysis" by Laura A. Casella, Sixin He, Erika Griesshaber, Lourdes Fernández-Díaz, Martina Greiner, Elizabeth M. Harper, Daniel J. Jackson, Andreas Ziegler, Vasileios Mavromatis, Martin Dietzel, Anton Eisenhauer, Sabino Veintemillas-Verdaguer, Uwe Brand and Wolfgang W. Schmahl.

We are happy about the positive replies of all three referees and express our acknowledgements to the reviewers for their comments which helped to improve the manuscript.

We attached a detailed reply to all reviewers' comments and a description of all changes made in the revised manuscript. The manuscript has been consequently amended. Acopy of the manuscript showing the changes is attached.

Due to additional data on TGA measurements presented in the revised manuscript, we extended our list of co-authors as follows: Martina Greiner and Sabino Veintemillas-Verdaguer. Both co-authors contributed to the preparation, conduction, and data evaluation of TGA data during the revision of our manuscript.

We hope that the manuscript in its revised version will be acceptable for publication in Biogeosciences. Thank you for your time handling our manuscript.

Yours sincerely, Laura Casella

Please also note the supplement to this comment:
https://www.biogeosciences-discuss.net/bg-2018-249/bg-2018-249-AC6-supplement.pdf

[Figure]

**Supplement:**

[revised manuscript text omitted]

---

## Author Response (AR1)

Laura A. Casella, M. Sc.

Ludwig-Maximilians-Universität München

Dept. für Geo- und Umweltwissenschaften

Theresienstr. 41

80333 München

Germany

Munich, October 8th, 2018

Dear Dr. de Nooijer,

Thank you very much for your positive comments and decision on our manuscript "Assessment of hydrothermal alteration on micro- and nanostructures of biocarbonates: quantitative statistical grain-area analysis of diagenetic overprint"
by Laura A. Casella, Sixin He, Erika Griesshaber, Lourdes Fernández-Díaz, Elizabeth M. Harper, Daniel J. Jackson, Andreas Ziegler, Vasileios Mavromatis, Martin Dietzel, Anton Eisenhauer, Uwe Brand, and Wolfgang W. Schmahl.

Please find the requested corrections of our manuscript below. A second revised version with changes tracked was uploaded as separate author comment.

Thank you very much for your time handling our manuscript!

Yours sincerely,

Laura Casella

**2nd revison of bg-2018-249**

| Comments of the associate editor | Submitted and revised manuscript | Newly revised manuscript |
|---|---|---|
| page 2, line 7: replace the comma after 'Zazzo et al., 2004' by a semicolon. I have also noticed this at various places in the text: please check. | " ... Brand, 2004; Zazzo et al., 2004, Casella et al…." "…al., 2012; Butler et al., 2009 2013; Schöne…" "…(Casella et al., 2017, Greiner et al., 2018)…" "… Figs. 2A, A4A; Casella et al. 2017)…" "…and texture (see also Casella et al. 2017)…." | "…Brand, 2004; Zazzo et al., 2004; Casella et al. ..." "…Butler et al., 2009, 2013; Schöne…" "… (Casella et al., 2017; Greiner et al., 2018)…." "…Figs. 2A, A4A; Casella et al., 2017…" "…and texture (see also Casella et al., 2017)…" |
| page 3, line 24: replace 'plane' by 'planar'. | "…with glass knifes to obtain plane surfaces.…" | "…with glass knifes to obtain planar surfaces…" |
| Throughout the manuscript and particularly figures: make sure 'sp.' in e.g. 'Porites sp.' | "…modern stony coral (*Porites sp.*) and…" "…modern stony coral *Porites sp.* consists…" "… (B): *Porites sp.*, (C, D): *Haliotis...*" | "… stony coral (*Porites* sp.) and..." "…stony coral *Porites* sp. consists..." "…(B): *Porites* sp., (C, D): *Haliotis…*"  Figures changed: - Fig. 1 - Fig. 2 - Fig. 6 - Fig. 12 - Fig. A3 - Fig. A4 - Fig. A5 - Fig. A9 |
| In many of the b/w figures (e.g. figure A11), the text is difficult to read. Consider placing the text in a black or white box. | | Figures changed: - Fig. 1 - Fig. 11 - Fig. A11 |

[revised manuscript text omitted]